# Learning a Neural Solver for Parametric PDEs to Enhance Physics-Informed Methods

Lise Le Boudec [1] *        Emmanuel de Bezenac [2]        Louis Serrano [1]

Ramon Daniel Regueiro-Espino [1]        Yuan Yin [3] †        Patrick Gallinari [1, 4]

[1] Sorbonne Université, CNRS, ISIR, F-75005 Paris, France
[2] INRIA Paris, France
[3] Valeo.ai, Paris, France
[4] Criteo AI Lab, Paris, France

## Abstract

Physics-informed deep learning often faces optimization challenges due to the complexity of solving partial differential equations (PDEs), which involve exploring large solution spaces, require numerous iterations, and can lead to unstable training. These challenges arise particularly from the ill-conditioning of the optimization problem caused by the differential terms in the loss function. To address these issues, we propose learning a solver, i.e., solving PDEs using a physics-informed iterative algorithm trained on data. Our method learns to condition a gradient descent algorithm that automatically adapts to each PDE instance, significantly accelerating and stabilizing the optimization process and enabling faster convergence of physics-aware models. Furthermore, while traditional physics-informed methods solve for a single PDE instance, our approach extends to parametric PDEs. Specifically, we integrate the physical loss gradient with PDE parameters, allowing our method to solve over a distribution of PDE parameters, including coefficients, initial conditions, and boundary conditions. We demonstrate the effectiveness of our approach through empirical experiments on multiple datasets, comparing both training and test-time optimization performance. The code is available at https://github.com/2ailesB/neural-parametric-solver.

## 1 Introduction

Partial Differential Equations (PDEs) are ubiquitous as mathematical models of dynamical phenomena in science and engineering. Solving PDEs is of crucial interest to researchers and engineers, leading to a huge literature on this subject (Evans, 2010; Salsa, 2015). Traditional approaches to solving PDEs such as finite difference, finite element analysis, or spectral methods (Zienkiewicz et al., 2005; LeVeque, 2007) often come with stability and convergence guarantees but suffer from a high computational cost. Improving numerical PDE solvers through faster and more accurate algorithms remains an active research topic (Zienkiewicz et al., 2005).

PDE solvers usually rely on discretization and/or linearization of the problem through various techniques to simplify the computations. Iterative methods such as Jacobi, Gauss-Seidel, Conjugate Gradient, and Krylov subspace methods can then be used to solve the resulting systems. Unfortunately, many PDEs have an ill-conditioned nature, and these iterative processes can demand extensive computational resources. Preconditioning techniques are often essential to mitigate this, though they require precise customization to the specific PDE problem, making the development of effective solvers a significant research endeavor in itself. Yet, the computational demands, time, and expertise required to develop these algorithms sometimes make them impractical or sub-optimal for specific

---

*Corresponding author: lise.leboudec@isir.upmc.fr.
†Work done during post-doc at Sorbonne University.

classes of problems. Instead of relying on hand-designed algorithms, researchers have investigated, as an alternative, the use of machine learning to train iterative PDE solvers Hsieh et al. (2019); Li et al. (2023a); Rudikov et al. (2024); Kopaničáková et al. (2023). These approaches usually parallel the classical numerical methods by solving a linear system resulting from the discretization of a PDE, for example, using finite differences or finite elements. A preconditioner is learned from data by optimizing a residual loss computed w.r.t. a ground truth solution obtained with a PDE solver. This preconditioner is used on top of a baseline iterative solver and aims at accelerating its convergence. Examples of baseline solvers are the conjugate gradient (Li et al., 2023a; Rudikov et al., 2024) or the Jacobi method Hsieh et al. (2019).

Another recent research direction investigates the use of neural networks for building surrogate models in order to accelerate the computations traditionally handled by numerical techniques. These methods fall into two main categories: *supervised* and *unsupervised*. The *supervised* methodology consists of first solving the PDE using numerical methods to generate input and target data and then regressing the solution using neural networks in the hope that this surrogate could solve new instances of the PDE. Many models, such as Neural Operators, lie within this class (Li et al., 2020; Raonić et al., 2023; Bartolucci et al., 2023) and focus on learning the solution operator directly through a single neural network pass. *Unsupervised* approaches, involve considering a neural network as a solution of the PDE. The neural network parameters are found by minimizing the PDE residual with gradient descent. Methods such as Physics-Informed Neural Networks (PINNs) (Raissi et al., 2019), or DeepRitz (E & Yu, 2018) fall under this category. This family of methods is attractive as it does not rely on any form of data, but only on information from the PDE residual. However, they exhibit severe difficulties during training (Krishnapriyan et al., 2021; Ryck et al., 2023), often requiring many optimization steps and sophisticated training schemes (Krishnapriyan et al., 2021; Rathore et al., 2024). The ill-conditioned nature of PDE residual loss appears again in this context, making standard optimizers such as Adam inappropriate (see appendix E.6 and appendix E.7 for a visualization of the ill conditioning of this loss landscape). A detailed review of the existing literature is described in appendix A.

In this work, we consider having access to the PDE as in unsupervised approaches and also to some data for training our neural solver. Our objective is to solve the optimization issues mentioned above by *learning an iterative algorithm that solves the PDE* from its residual, defined as in the PINNs framework (see fig. 1). This *neural solver* is trained from data, either simulations or observations. Different from the classical ML training problem which aims at learning the parameters of a statistical model from examples, the problem we handle is learning to learn, Andrychowicz et al. (2016), i.e. learning an iterative algorithm that will allow us to solve a learning problem. When vanilla PINNs handle a single PDE instance, requiring retraining for each new instance, we consider the more complex setting of solving parametric PDEs, the parameters may include boundary or initial conditions, forcing terms, and PDE coefficients. Each specific instance of the PDE, sampled from the PDE parameter distribution, will then be considered as a training example. The objective is then to learn a solver from a sample of the parametric PDE distribution in order to accelerate inference on new instances of the equation. With respect to unsupervised approaches, our model implements an improved optimization algorithm, tailored to the parametric PDE problem at hand instead of using a hand-defined method such as stochastic gradient descent (SGD) or Adam. As demonstrated in the experimental section, this approach proves highly effective for the ill-posed problem of optimizing the PINNs objective, enabling convergence in just a few steps. This is further illustrated through gradient trajectory visualizations in appendix E.6. In the proposed methodology, the neural solver will make use of the gradient information computed by a baseline gradient method to accelerate its convergence. In our instantiation, we will use SGD as our baseline algorithm, but the method could be easily extended to other baselines. Our model deviates from the traditional preconditioning methods by directly optimizing the non-linear PDE residual loss (Raissi et al., 2019) without going through the discretization steps. Our contribution includes :

- Setting an optimization framework for learning to iteratively solve parametric PDEs from physics-informed loss functions. We develop an instantiation of this idea using an SGD baseline formulation. We detail the different components of the framework as well as training and inference procedures.
- Evaluating this method on challenging PDEs for physics-informed method, including failure cases of classical PINNs and showing that it solves the associated optimization issues and accelerates the convergence.

- Extending the comparison to several parametric PDEs with varying parameters from 1d static to 2d+time problems. We perform a comparison with baselines demonstrating a significant acceleration of the convergence w.r.t. baselines.

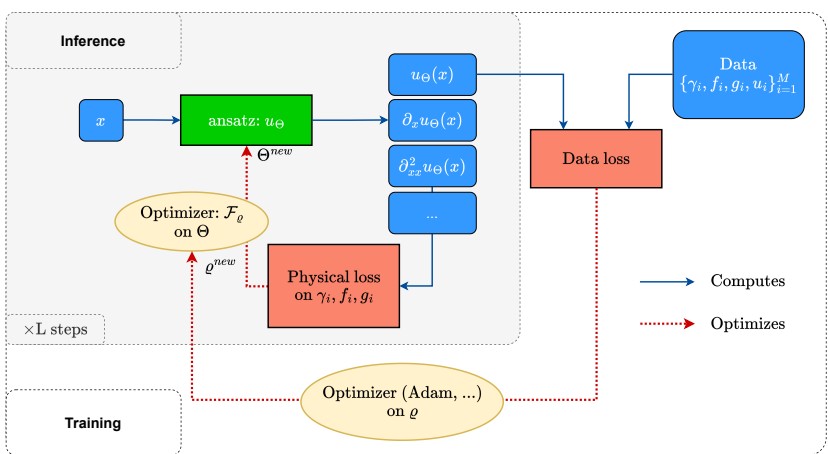

Figure 1: Optimization scheme of a physics-informed method with our framework.

## 2 MOTIVATION

Let us first motivate our objective with a simple example. Physics-informed neural networks (PINNs) are a promising tool for optimizing neural networks in an unsupervised way to solve partial differential equations (PDEs). However, these methods are notoriously difficult to train (Krishnapriyan et al., 2021; Ryck et al., 2023). As an illustrative example of this challenge, let us solve the zero-boundary Poisson equation in 1d on $\Omega = [-\pi, \pi]$. Note that this section is intentionally informal, we provide rigorous statements and proofs in Appendix B.

**Poisson equation, 1d.** The solution is given by $u(x) = \sin(kx)$.

$$u''(x) = -k^2 \sin(kx),$$
$$u(-\pi) = 0,\ u(\pi) = 0. \tag{1}$$

Physics-informed machine learning relies on an *ansatz space* of *parametric functions*, e.g. neural networks $u_\Theta : \Omega \mapsto \mathbb{R}$, minimizing the following loss in order to satisfy the constraints in equation 1:

$$\mathcal{L}_{\mathrm{PDE}} = \mathcal{L}_{\mathrm{Res}} + \lambda \mathcal{L}_{\mathrm{BC}}, \quad \mathcal{L}_{\mathrm{Res}} = \int_\Omega |u''_\Theta(x) - f(x)|^2 \, dx,$$

$$\mathcal{L}_{\mathrm{BC}} = \frac{1}{2} \left[ u_\Theta(-\pi)^2 + u_\Theta(\pi)^2 \right].$$

As a simple example, consider the parametrization given by considering a linear combination of Fourier features widely used (Tancik et al., 2020)[1], $u_\Theta(x) = \sum_{k=-K}^{K} \theta_k \phi_k(x)$, with $\phi_0(x) = \frac{1}{\sqrt{2\pi}}$, $\phi_{-k}(x) = \frac{1}{\sqrt{\pi}} \cos(kx)$ and $\phi_k(x) = \frac{1}{\sqrt{\pi}} \sin(kx)$ for $1 \le k \le K$.

This simple but informative example yields a tractable gradient descent algorithm, as the associated updates are linear in the parameters, governed by a matrix $A$ and constant $b$:

$$\begin{aligned} \Theta_{l+1} &= \Theta_l - \eta \nabla \mathcal{L}_{\mathrm{PDE}}(\Theta_l) \\ &= (I - \eta A)\Theta_l + b \end{aligned} \tag{2}$$

---

[1]Note that even though the ansatz is linear in $\Theta$, it is not linear in $x$.

with $A$ whose condition number is $\kappa(A) := \lambda_{\max}(A)/\lambda_{\min}(A) \geq K^4$:

$$A = \begin{bmatrix} 0^4 & 0 & \cdots & 0 \\ 0 & 1^4 & \cdots & 0 \\ \vdots & \vdots & \ddots & \vdots \\ 0 & 0 & \cdots & K^4 \end{bmatrix} + \lambda \begin{bmatrix} \phi_1(\pi) \\ \phi_2(\pi) \\ \vdots \\ \phi_K(\pi) \end{bmatrix} \begin{bmatrix} \phi_1(\pi) & \phi_2(\pi) & \cdots & \phi_K(\pi) \end{bmatrix}. \tag{3}$$

This implies that the condition number of $A$ increases extremely rapidly in the ratio between the highest and lowest frequencies of the network. Given that the rate of convergence to the optimum $\Theta^* = \Theta_0 + A^{-1}b$ can be bounded as

$$\|\Theta_l - \Theta^*\|_2 \leq (1 - c/\kappa(A))^l \|\Theta_0 - \Theta^*\|_2, \tag{4}$$

the number of steps $N(\varepsilon)$ required to obtain an error of size at most $\varepsilon$, i.e., $\|\Theta_l - \Theta^*\|_2 \leq \varepsilon$ increases linearly in the condition number, i.e. as the fourth power of the maximal frequency $K$:

$$N(\varepsilon) = O\left(\kappa(A)\ln\frac{1}{\epsilon}\right) = O\left(K^4 \ln\frac{1}{\epsilon}\right). \tag{5}$$

We believe that this simple example clearly illustrates and highlights the fact that PINNs–even when considering a linear basis, and when the PDE is linear–suffer heavily from ill-conditioning: if 500 steps are required in order to achieve a given error when $K = 5$, roughly speaking, $312\,500$ steps are required for only $K = 25$. This result extends to more general linear systems of equations and linear ansatz, as explained in appendix B.

Our objective in the following will be to accelerate the convergence of such systems in this context–as well as extend them to the non-linear setting. To do so, in the following section, we will learn how to transform the optimization problem in such a way that the number of gradient descent iterations is small. The resulting method can be seen as a standalone, iterative solver as it is not only applicable to different PDEs but can handle a wide range of initial/boundary conditions and parameters.

## 3 APPROACH

In order to optimize PDE-based losses, we propose to learn a physics-based optimizer that will fulfill two objectives: (i) allowing a fast test-time optimization given a new PDE and (ii) solving without retraining parametric PDEs, with varying PDE coefficients $\gamma$ [2], forcing terms $f$, and initial/boundary conditions $g$ using the same model. We present the general framework below and propose an instantiation that leverages a linear combination of basis functions as the ansatz.

### 3.1 PROBLEM STATEMENT

Let us consider the following family of boundary value problems parameterized by $\gamma$ with domain $\Omega$, representing both space and time, with $\mathcal{N}$ a potentially nonlinear differential operator, $\mathcal{B}$ the boundary operator, $g$ the initial/boundary conditions, and source term $f$:

$$\mathcal{N}(u; \gamma) = f \quad \text{in } \Omega, \tag{6}$$
$$\mathcal{B}(u) = g \quad \text{on } \partial\Omega. \tag{7}$$

Note that different PDEs can be represented in this form, amounting to changing the parameters $\gamma$. The goal here is to develop a generic algorithm that is able to solve the above problem, yielding an approximate solution $u$ given the PDE and different sets of inputs $(\gamma, f, g)$.

For training, we assume access to a dataset of $M$ problem instances, represented by the PDE parameters $(\gamma_i, f_i, g_i)_{i=1}^M$ and to associated target solution $(u_i)_{i=1}^M$ given on a $m$ point grid $(x_j)_{j=1}^m$. The solutions $(u_i)_{i=1}^M$ will be used to train the neural solver. At inference, for a new PDE instance, only the PDE parameters are provided and we do not have access to solution points $(u_i)_{i=1}^M$.

---

[2]Note that PDE coefficients can be functions, an example is the Darcy PDE in section 4.1

## 3.2 METHODOLOGY

Physics-informed neural networks consider an ansatz $u_\Theta$ parametrized by some finite-dimensional $\Theta$. The parameters $\Theta$ are iteratively updated by minimizing a criterion $\mathcal{L}_{\mathrm{PDE}}$ (e.g. the PDE residual), which assesses how well the ansatz $u_\Theta$ meets the conditions specified in equations 6 and 7. As introduced for example for PINNs (Raissi et al., 2019) or Deep Galerkin method (Sirignano & Spiliopoulos, 2018), we consider $\mathcal{L}_{\mathrm{PDE}}$ to be given by the strong formulation of the residual $\mathcal{L}_{\mathrm{Res}}$, plus a boundary discrepancy term[3] $\mathcal{L}_{\mathrm{BC}}$: $\mathcal{L}_{\mathrm{PDE}} = \mathcal{L}_{\mathrm{Res}} + \lambda \mathcal{L}_{\mathrm{BC}}, \quad , \lambda > 0$.

$$\mathcal{L}_{\mathrm{Res}} = \sum_{x_j \in \Omega} |\mathcal{N}(u_\Theta; \gamma)(x_j) - f(x_j)|^2, \quad \mathcal{L}_{\mathrm{BC}} = \sum_{x_j \in \partial\Omega} |\mathcal{B}(u_\Theta)(x_j) - g(x_j)|^2 \qquad (8)$$

As illustrated in section 2, performing gradient descent, or alternatives such as Adam and L-BFGS on such a highly ill-conditioned loss $\mathcal{L}_{\mathrm{PDE}}$ leads to severe training difficulties (Krishnapriyan et al., 2021). The key idea in our work is to improve a baseline gradient descent algorithms with the neural solver. More precisely starting from a baseline gradient algorithm, SGD in our instantiation, instead of considering the classical update, we first *transform* the gradient using a neural network $\mathcal{F}_\varrho$ with parameters $\varrho$, depending on the values of the PDE parameters as well as on other inputs such as the residual gradient provided by SGD: $\nabla_\Theta \mathcal{L}_{\mathrm{PDE}}$. The objective is to transform, through the neural solver $\mathcal{F}_\varrho$, the ill-conditioned problem into a new, simpler problem that requires fewer steps in order to achieve a given error.

Once the neural solver $\mathcal{F}_\varrho$ has been learned, inference can be performed on any new PDE as follows (see inference algorithm 1). Starting from an initial ansatz parameter $\Theta_0$, it is iteratively updated by this solver. At iteration $l$, the steepest direction of the loss $\mathcal{L}_{\mathrm{PDE}}$ is first computed with autograd. Then, the gradient is transformed, in a PDE parameter dependant way, with $\mathcal{F}_\varrho$:

$$\Theta_{l+1} = \Theta_l - \eta \mathcal{F}_\varrho(\nabla_\Theta \mathcal{L}_{\mathrm{PDE}}(\Theta_l),\ \gamma, f, g) \qquad (9)$$

The objective is to iteratively refine the ansatz to closely approximate the true solution after a series of $L$ iterations, *ideally small* for efficiency.

This approach can be seen as learning the iterates in a PDE solver to achieve a low loss, similar to the residual minimization methods in PDEs (Elman et al., 2014). By design, this solver is intended to be applicable to different PDEs, as well as various sources, boundary conditions, and initial conditions. This flexibility allows for a broad range of applications, making it a versatile tool in solving complex PDEs with varying characteristics.

Designed as a *parametric PDE solver*, $\mathcal{F}_\varrho$ [4] is trained with input target data from different sets of PDE parameters, as outlined in section 3.1. Once trained, it will be used without retraining on new instances of the PDE, i.e. with new values of the PDE parameters. The underlying hypothesis is that even though the solutions may be different for different inputs and parameters, the solution methodology remains relatively consistent. This consistency is expected to enhance the algorithm's ability to generalize across novel scenarios effectively.

## 3.3 TRAINING OF A PHYSICS-INFORMED SOLVER

**Choice of Ansatz $u_\Theta$.** A very common choice (Shen et al., 2011) is to consider a family of basis functions $\Psi(x) = \{\psi_i(x)\}_{i=1}^N$ and consider the ansatz to be given by its linear span $u_\Theta(x) = \sum_{i=0}^N \theta_i \psi_i(x)$. In the following, we consider this linear reconstruction, although our formulation is generic in the sense that it can also accommodate nonlinear variants. [5]

As indicated in eq. (10), the solver $\mathcal{F}_\varrho$ will be trained from samples of the PDE parameter distribution $(\gamma, f, g)$ and from the associated samples of the solution $u$. We first describe below the inference step aiming at iteratively updating the parameters $\Theta$ of the solution function $u_\Theta$ while $\mathcal{F}_\varrho$ is held fixed. We then describe how the $\varrho$ parameters of the solver are trained. Please refer to fig. 1 that illustrates the interaction between the two steps.

---

[3]Note that other formulations of the loss may also be considered in a straightforward manner.

[4]In the following we use "solver" with $\mathcal{F}_\varrho$ as a short-hand to refer to our proposed method.

[5]Although we have found this may further complicate training.

**Inference** The inference step is performed at fixed values of the solver parameters $\mathcal{F}_\varrho$. It consists, for a given instance of the PDE with parameters $(\gamma_i, f_i, g_i)$, in finding the best $\Theta$ with a few steps of the solver using eq. (9). It is illustrated in fig. 1 - grey box and formalized in algorithm 1: starting from initial parameters $\Theta_0$, we compute the Physical loss $\mathcal{L}_{\text{PDE}}$ using the ansatz $u_{\Theta_0}$. The PDE derivatives in $\mathcal{L}_{\text{PDE}}$, can be computed by hand or automatic differentiation depending on the application [6]. Then, eq. (9) is used to update the parameters $\Theta$ for a given number of steps $L$. The final solution is reconstructed using the linear combination, $u_\Theta(x) = \sum_{i=0}^N \theta_i \psi_i(x)$, introduced in section 3.2 with the computed coefficients $\Theta_L$. Note that inference does not make use of the sampled target solutions $(u_i)_{i=1}^M$ computed on the grid points $(x_j)_{j=1}^m$. These targets are used exclusively for training the neural solver.

**Training the neural solver** Training amounts to learning the parameters of the solver $\mathcal{F}_\varrho$ and is performed with a training set of PDE parameters and simulation data considered as ground truth $(\gamma_i, f_i, g_i, u_i)_{i=1}^M$, corresponding to PDE instances (*i.e.* with **different** parameters

---

**Algorithm 1:** Inference using the neural PDE solver.

Data: $\Theta_0 \in \mathbb{R}^n$, PDE $(\gamma, f, g)$
Result: $\Theta_L \in \mathbb{R}^n$
for l = 0...L-1 do
$\quad \mid \quad \Theta_{l+1} = \Theta_l - \eta \mathcal{F}_\varrho(\nabla \mathcal{L}_{\text{PDE}}(\Theta_l), \gamma, f, g)$
end
return $\Theta_L$

---

**Algorithm 2:** Training algorithm for learning to optimize physics-informed losses.

Data: $\Theta_0 \in \mathbb{R}^n$, PDE $(\gamma, f, g)$, sample values $u(x)$
Result: $\mathcal{F}_\varrho$
for e = 1... epochs do
$\quad \mid \quad$ for (PDE, x, u) in dataset do
$\quad \mid \quad \quad \mid \quad$ Initialize $\Theta_0$
$\quad \mid \quad \quad \mid \quad$ Estimate $\Theta_L$ from $\Theta_0, (\gamma, f, g)$ using algorithm 1
$\quad \mid \quad \quad \mid \quad$ Reconstruct $u_{\Theta_L}(x)$
$\quad \mid \quad \quad \mid \quad$ Update $\varrho$ with gradient descent from the data loss in eq. (10)
$\quad \mid \quad$ end
end
return $\mathcal{F}_\varrho$

---

$\gamma$ and/or forcing terms $f$ and/or initial/boundary conditions $g$). See fig. 1 - white box and algorithm 2. The objective is to learn a solver $\mathcal{F}_\varrho$ able, at inference, to converge to a target solution in a small (2 to 5 in our experiments) number of steps. For that, an optimizer (Adam in our experiments) is used to update the $\mathcal{F}_\varrho$ parameters. The training algorithm makes use of the data associated with the different PDE instances by sampling PDEs in batches and running algorithm 1 on several PDE instances. For each PDE instance, one starts from an initial parameter value $\Theta_0$ and then performs two optimization steps (see algorithm 2): (i) one consists in solving in the ansatz parameters $\Theta$ using the neural solver using algorithm 1, leading to $\Theta_L$; (ii) the second one is the optimization of the solver parameters $\varrho$. We train the outputs directly to match the associated ground truth $(u_i(x_j))_{j=1}^m$ using the data loss:

$$\mathcal{L}_{\text{DATA}} = \mathbb{E}_{\gamma, f, g} \left[ ||u_{\Theta_L} - u_{\gamma, f, g}|| \right]. \tag{10}$$

The expectation is computed on the distribution of the PDE parameters $(\gamma, f, g)$. The solution $u$ is entirely determined by these parameters as indicated by the notation $u_{\gamma, f, g}$. $||u_{\Theta_L} - u||$ denotes a distance between the target $(u(x_j))_{j=1}^m$ and the forecast $(u_{\Theta_L}(x_j))_{j=1}^m$ with $m$ the trajectory size [7]. In practice, one samples a set of PDE instances $(\gamma_i, f_i, g_i)$ and for each instance a corresponding sample $u_i$.

**Theoretical analysis and relation to preconditioning** Analyzing the behavior of the inference algorithm is challenging due to the non linear nature of the solver. We however could get some intuition using simplifying assumptions. We build on the ideas introduced in section 2 for the simple case of the Poisson equation, for which an explicit analytical solution could be derived. We provide in appendix B, a proof for a more general case and give below in theorem 1 our main result. This shows that the number of steps induced by $\mathcal{F}$ for the proposed algorithm is significantly less than the number of steps required by the baseline PINNs algorithm. This results is obtained under two main assumptions: (i) $\mathcal{F}$ behaves like its linearization and (ii) the descent operator $\mathcal{F}$ used in our algorithm, allows us to reach the optimum of $\mathcal{L}_{\text{DATA}}$.

---

[6] In our experiments, we computed the derivative by hand when possible since it fastens computations.

[7] To simplify the notation, we used a fixed grid size $m$. However, this framework can be used with different grid sizes, as well as irregular grids. See Ablation in appendix E, table 13

**Theorem 1.** *(Convergence rate in the linear case). Given a linear ansatz $u_\Theta(x) = \sum_{i=1}^{N} \theta_i \phi_i(x)$, assume the conditioner $\mathcal{F}$ behaves like its linearization $P = Jacobian(\mathcal{F})$, meaning that $\mathcal{F}$ can be replaced by $P$ at any point. Let $A$ be the matrix derived from the PDE loss as eq. (3) for the Poisson equation or eq. (15) in the more general case. Denote by $\kappa(A)$ the condition number of the matrix $A$. The number of steps $N'(\varepsilon)$ required to achieve an error $\|\Theta_l - \Theta^*\|_2 \leq \varepsilon$ satisfies:*

$$N'(\varepsilon) = O\left(\kappa(PA)\ln\left(\tfrac{1}{\varepsilon}\right)\right), \tag{11}$$

*Moreover, if $\mathcal{F}$ minimizes $\mathcal{L}_{DATA}$ this necessarily implies $\kappa(PA) = 1 \leq \kappa(A)$. Consequently, the number of steps is effectively reduced, i.e., $N'(\varepsilon) \ll N(\varepsilon)$ with $N(\varepsilon)$ the number of steps of the vanilla PINNs.*

*Proof.* We sketch the main insights here and refer to appendix B for the proof and a detailed analysis.

- Using a linearization of the neural solver, it can be shown that the solver performs as a pre-conditioner on the linear system.

- Assuming that solution $u_L$ provided by the solver reaches the optimum $u^*$, and that the training set is such that the learned parameter $\Theta$ vectors span the whole parameter space of the model, then the convergence of the solver is guaranteed at an optimal rate.

- In practice, and as shown in the experiments (section 4), the convergence rate is significantly improved w.r.t. the reference baseline gradient algorithm.

□

## 4 EXPERIMENTS

We present the datasets used in the experiments in section 4.1, a comparison with selected baselines in section 4.2, and a test-time comparison with different optimizers demonstrating the remarkable effectiveness of the proposed method in section 4.3. Finally, we make a comparison of the training and inference time in section 4.4. Experimental details and additional experiments can be found in the appendices: ablations are in appendix E and additional results and visualization are in appendix E.6, appendix E.7, and appendix F.

### 4.1 DATASETS

We consider several representative parametric equations for our evaluation. More details about the data generation are presented in appendix C. Our objective is to learn a neural solver able to solve quickly and accurately a new instance of a PDE, given its parametric form, and the values of the parameters $\gamma$, forcing terms $f$ and initial/boundary conditions $g$, *i.e.* $(\gamma, f, g) \mapsto u$. Solving is performed with a few iterations of the neural solver (algorithm 1). For that, one trains the neural solver on a sample of the PDE parameter instances, see table 1 for the parameter distributions used for each parametric PDE. **Evaluation is performed on unseen sets of parameters within the same PDE family.** **Helmholtz**: We generate a dataset following the $1d$ static Helmholtz equation $u''(x) + \omega^2 u(x) = 0$ with

Table 1: Parameters changed between each trajectory in the considered datasets.

| Dataset | Parameters | Distribution |
|---|---|---|
| Helmholtz | $\omega$ | $\mathcal{U}[0.5, 50]$ |
| | $u_0$ | $\mathcal{N}(0,1)$ |
| | $v_0$ | $\mathcal{N}(0,1)$ |
| Poisson | $A_i$ | $\mathcal{U}[-100, 100]$ |
| | $u_0$ | $\mathcal{N}(0,1)$ |
| | $v_0$ | $\mathcal{N}(0,1)$ |
| NLRD | $\nu$ | $\mathcal{U}[1, 5]$ |
| | $\rho$ | $\mathcal{U}[-5, 5]$ |
| Darcy | $a(x)$ | $\psi_\# \mathcal{N}(0, (-\Delta + 9I)^{-2})$ with $\psi = 12 * \mathbb{1}_{\mathbb{R}_+} + 3 * \mathbb{1}_{\mathbb{R}_+}$ |
| Heat | $\nu$ | $\mathcal{U}[2 \times 10^{-3}, 2 \times 10^{-2}]$ |
| | $J_{max}$ | $\{1, 2, 3, 4, 5\}$ |
| | $A$ | $\mathcal{U}[0.5, -0.5]$ |
| | $K_x, K_y$ | $\{1, 2, 3\}$ |
| | $\phi$ | $\mathcal{U}[0, 2\pi]$ |

boundary conditions $u(0) = u_0$ and $u'(0) = v_0$. We generate $1,024$ trajectories with varying $\omega, u_0$,

and $v_0$ with a spatial resolution of 256. **Poisson**: We generate a dataset following the $1d$ static Poisson equation with forcing term: $-u''(x) = f(x)$ with $u(0) = u_0$ and $u'(0) = v_0$. The forcing term $f$ is a periodic function, $f(x) = \frac{\pi}{K} \sum_{i=1}^{K} a_i i^{2r} \sin(\pi x)$, with $K = 16$ and $r = -0.5$. We generate $1,000$ trajectories with varying $u_0, v_0$, and $f$ (through changing $a_i$) with a spatial resolution of 64. **Reaction-Diffusion**: In Krishnapriyan et al. (2021); Toloubidokhti et al. (2024), the authors propose a non-linear reaction-diffusion (*NLRD*). This PDE has been shown to be a failure case for PINNs (Krishnapriyan et al., 2021). We generate $1,000$ trajectories by varying the parameters of the PDE: $\nu$ and $\rho$ (see table 1). Spatial resolution is 256 and temporal resolution is 100. The PDE is solved on $[0, 1]^2$. **Darcy Flow**: The $2d$ Darcy Flow dataset is taken from (Li et al., 2020) and is commonly used in the operator learning literature (Li et al., 2023b; Goswami et al., 2022). For this dataset, the forcing term $f$ is kept constant $f = 1$, and $a(x)$ is a piece-wise constant diffusion coefficient taken from (Li et al., 2020). We kept $1,000$ trajectories (on the $5,000$ available) with a spatial resolution is $64 \times 64$. **Heat**: The $2d + t$ Heat equation is simulated as proposed in (Zhou & Farimani, 2024). For this dataset, the parameter $\nu$ is sampled from $\mathcal{U}[2 \times 10^{-3}, 2 \times 10^{-2}]$ and initial conditions are a combination of sine functions with a varying number of terms, amplitude, and phase. A summary of the datasets and the varying parameters for each PDE are presented in table 1 and more details on the dataset are provided in appendix C. Experiments have been conducted on NVIDIA TITAN V (12 Go) for $1d$ datasets to NVIDIA RTX A6000 GPU with 49Go for $1d$ + time or $2d$ datasets. For all datasets, 800 PDEs are considered during training and 200 for testing. All metrics reported are evaluated on test samples (*i.e.* **PDEs not seen during training**. Coefficients as well as initial and/or boundary conditions can vary from training).

## 4.2 COMPARISON WITH BASELINES

We performed comparisons with several baselines including fully data-driven supervised approaches trained from a data-loss only, unsupervised methods relying only on a PDE loss, and hybrid techniques trained from PDE + DATA losses. Network size and training details are described in appendix D. In this experiment, we considered training the models using the training sets (physical losses or MSE when possible) unless stated otherwise.

**Fully supervised** We train a standard MLP to learn the mapping $(\gamma, f, g) \mapsto \Theta$, using as loss function $\mathcal{L}_{\text{DATA}} = \mathbb{E}_{\gamma, f, g, u} [||u_{\Theta_L} - u_{\gamma, f, g}||]$ with $u_\Theta(x) = \sum_{i=0}^{N} \theta_i \psi_i(x)$, the $\psi_i(.)$ being fixed B-Spline basis functions (see appendix D). We denote this baseline as *MLP+basis*. **Unsupervised** We compare our approach with unsupervised physics-informed models (Raissi et al., 2019). While the initial version of PINNs solves only one PDE instance at a time and requires retraining for each new instance, we developed here a parametric version of PINNs (*PPINNs*) where the PDE parameters are fed to the network (similarly to (Zhang et al., 2023)). Finally, we used (Cho et al., 2024)'s (*P2INNs*) method as a physics-informed baseline specifically designed for parametric PDEs. In addition to PINNs-methods, we also compare our solver to the Physic-informed DeepONet (*PO-DeepONet* for Physics-Only DeepONet) (Wang et al., 2021b), which is designed to learn an operator for *function-to-function* mappings from physical losses and handles parametric PDEs. The mapping learned for the two unsupervised baselines is $(x, \gamma, f, g) \mapsto u_{\gamma, f, g}(x)$. In order to provide a fair comparison with our optimization method, we fine-tuned the unsupervised baselines for each specific PDE instance for a few steps (10 or 20). **Comparison to preconditioning** We compare our approach with vanilla PINNs (Raissi et al., 2019), *i.e.* by fitting one PINN per PDE in the test set and averaging the final errors. We optimize the PDE losses using L-BFGS (Liu & Nocedal, 1989) and refer to this baseline as *PINNs+L-BFGS*. As discussed in (Rathore et al., 2024), L-BFGS can be considered as a nonlinear preconditioning method for Physics-Informed methods and fastens convergence. Finally, we use the training strategy proposed by (Rathore et al., 2024) *i.e.* trained PINNs using successive optimizer (Adam + L-BFGS). This baseline is denoted as *PINNs-multi-opt*. For these baselines, one model is trained and evaluated for each PDE in the **test** set. We report the reader to appendix D for more details on the training procedure. **Hybrid** Finally, we compare our proposed method with neural operators, *i.e.*, models trained to learn mappings $(x, \gamma, f, g) \mapsto u_{\gamma, f, g}(x)$ using a combination of physical and data loss: $\mathcal{L}_{\text{DATA}} + \mathcal{L}_{\text{PDE}}$. We use as baselines Physics-Informed Neural Operator (*PINO*) (Li et al., 2023b) and Physics-Informed DeepONet (*PI-DeepONet*) (Goswami et al., 2022). As already indicated, for a fair comparison, the *Unsupervised* and *Hybrid* baselines are fine-tuned on each specific PDE instance for a few steps (10 on all datasets except for Heat for which 20 steps are made). **Ours** We represent the solution $u_\Theta$ with a linear combination of B-Spline functions for $\Psi$ (Piegl & Tiller, 1996). This was motivated by the nature of B-Splines which allows to capture local

phenomena. However, other bases could be used such as Fourier, Wavelet or Chebychev Polynomials. The neural solver $\mathcal{F}_\varrho$ is composed of Fourier Layers (FNO) (Li et al., 2020) that allow us to capture the range of frequencies present in the phenomenon. We refer the reader to appendix D for more details about the construction of the B-Spline basis and the training hyper-parameters.

Table 2: Results of trained models - metrics in Relative MSE on the test set. Best performances are highlighted in **bold**, and second best are underlined.

|  |  | 1d | | 1d+time | 2d | 2d+time |
| --- | --- | --- | --- | --- | --- | --- |
|  | Baseline | Helmholtz | Poisson | NLRD | Darcy-Flow | Heat |
| Supervised | *MLP + basis* | 4.66e-2 | 1.50e-1 | **2.85e-4** | 3.56e-2 | 6.00e-1 |
| Unsupervised | *PINNs+L-BFGS* | 9.86e-1 | 8.83e-1 | 6.13e-1 | 9.99e-1 | 9.56e-1 |
|  | *PINNS-multi-opt* | 8.47e-1 | 1.18e-1 | 7.57e-1 | 8.38e-1 | 6.10e-1 |
|  | *PPINNs* | 9.89e-1 | 4.30e-2 | 3.94e-1 | 8.47e-1 | 1.27e-1 |
|  | *P2INNs* | 9.90e-1 | 1.50e-1 | 5.69e-1 | 8,38e-1 | 1.78e-1 |
|  | *PO-DeepONet* | 9.83e-1 | 1.43e-1 | 4.10e-1 | 8.33e-1 | 4.43e-1 |
| Hybrid | *PI-DeepONet* | 9.79e-1 | 1.20e-1 | 7.90e-2 | 2.76e-1 | 9.18e-1 |
|  | *PINO* | 9.99e-1 | 2.80e-3 | 4.21e-4 | 1.01e-1 | 9.09e-3 |
| Neural Solver | *Ours* | **2.41e-2** | **5.56e-5** | 2.91e-4 | **1.87e-2** | **2.31e-3** |

**Results:** Table 2 presents the comparison with the baselines. We recall that the evaluation set is composed of several PDE instances sampled from unseen PDE parameters $(\gamma, f, g)$. The proposed method is ranked first or second on all the evaluations. The most comparable baselines are the unsupervised methods, since at inference they leverage only the PDE residual loss, as our method does. Therefore our method should be primarily compared to these baselines. Supervised and hybrid methods both incorporate data loss and make different assumptions while solving a different optimization problem.

Table 2 clearly illustrates that unsupervised Physics-informed baselines all suffer from ill-conditioning and do not capture the dynamics. Compared to these baselines, the proposed method improves at least by one order of magnitude in all cases. PINNs baseline performs poorly on these datasets because of the ill-conditioning nature of the PDE, requiring numerous optimization steps to achieve accurate solutions (appendix D). This is observed on PINNs models for parametric PDEs (PPINNs and P2INNs) as well as on PINNs fitted on one equation only (PINNs+L-BFGS and PINNs-multi-opt). We observe that our neural solver has better convergence properties than other Physics-Informed methods. As will be seen later it also converges much faster.

The supervised baseline performs well on all the PDEs except *Poisson* and *Heat*. The data loss used for training this model is the mean square error which is well-behaved and does not suffer from optimization problems as the PDE loss does. We note that our method reaches similar or better performances on every datasets, while relying only on physical information at inference (algorithm 1) and solving a more complex optimization problem.

The hybrid approaches, do not perform well despite taking benefits from the PDE+DATA loss and from adaptation steps at test time. Again, the proposed method is often one order of magnitude better than the hybrids except on NLRD, where it has comparable performances. This shows that the combination of physics and data losses is also hard to optimize, and suffers from ill-conditioning.

### 4.3 OPTIMIZATION FOR SOLVING NEW EQUATIONS

The main motivation for our learned PDE solver is to accelerate the convergence to a solution, w.r.t. predefined solvers, for a new equation. In order to assess this property, we compare the convergence speed at test time inference with classical solvers, PINNs, and pre-trained PINO as detailed below. Results are presented in fig. 2 for the *Poisson* equation with performance averaged on 20 new instances of the *Poisson* equation. This experiment is also performed on the other datasets in appendix F.

**Baseline optimizers** As for the classical optimizer baselines, we used SGD, Adam (Kingma & Ba, 2015), and L-BFGS (Krishnapriyan et al., 2021). These optimizers are used to learn the coefficient of

the B-Spline basis expansion in the model $u_\Theta(x) = \sum_{i=0}^{N} \theta_i \psi_i(x)$. This provides a direct comparison to our iterative neural solver. **PINNs** - We also compare to the standard PINNs (Raissi et al., 2019), *i.e.* by fitting one Neural Network (NN) per equation. Note that this requires full training from scratch for each new equation instance and this is considerably more computationally demanding than solving directly the parametric setting. This baseline is similar to the Adam optimizer mentioned above, except that the ansatz for this experience is a multilayer perceptron instead of a linear combination of a B-Spline basis. **Hybrid pre-training strategies** - Finally we compare against the hybrid *PINO* pre-trained on a set of several parametric PDE instances and then fine-tuned on a new PDE instance using only the PDE loss associated to this instance. **Ours** - We train our model as explained in algorithm 2 and show here the optimization process at **test time**. In order to perform its optimization, our model leverages the gradient of the physical loss and the PDE parameters (coefficients, initial/boundary conditions). In this experiment, we use $L = 5$ steps for a better visualization (whereas, we used $L = 2$ in table 2).

**Result**: fig. 2 compares the number of optimization steps required for the different methods. Our neural solver converges very fast in only a few steps (5 here) to a good approximation of the solution, while all the other methods require thousands of iterations - we stopped here at $10,000$ steps. The classical optimizers (SGD, Adam, L-BFGS) do not converge for a new equation. The baseline PINNs trained here from scratch on each new equation show an erratic convergence behavior. Pretrained PINO behaves better than the other baselines but still did not converged after $10,000$ steps. This clearly demonstrates the potential of our learned solver to deal with physical losses w.r.t. alternative pre-defined solvers.

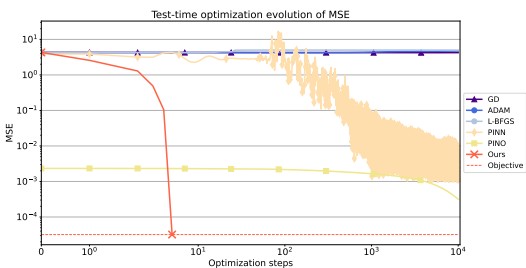

Figure 2: Test-time optimization based on the physical residual loss $\mathcal{L}_{\text{PDE}}$ for new PDE on *Poisson*.

## 4.4 Computational time

An important aspect of solving PDEs is the computational time required for each solution. Methods should find a trade-off between achieving high performance and maintaining reasonable computational costs. In appendix E.3, we provide the training (table 16) and inference (table 17) times for our method compared to various baselines. Our results show that while our method has comparable training and inference times to other approaches, it demonstrates substantially better precision (table 2).

## 5 Conclusion

We have presented a PDE solver learned from data that allows fast test-time optimization of physical losses. Our method succeeds to considerably accelerate the optimization process for the complex problem of minimizing physical losses and is several orders of magnitude faster than classical hand-defined optimization methods such as Adam or L-BFGS. **Limitations and Future Work** While efficient, the proposed method can be further improved. First, training our iterative algorithms requires more memory than standard machine learning models due to the complexity of backpropagation through iterations, which becomes challenging in higher-dimensional bases. Second, we have focused on solution approximations expressed as a linear expansion in predefined bases. More expressive representations, such as neural networks, could be explored; however, our preliminary experiments indicate increased ill-conditioning due to the compositional nature of neural networks. More sophisticated training schemes could enhance the optimization process. Future work will investigate these directions to improve scalability and broaden the applicability of the proposed method. **Reproducibility Statement** Hyper-parameters, baselines configurations and training details are detailed in appendix D, tables 4 and 5, and algorithms 1 and 2. The creation of the datasets is explained in appendix C and tables 1 and 3. Finally, we provide a theoretical analysis of the model, under an ideal scenario in appendix B. Code is available at `https://github.com/2ailesB/neural-parametric-solver`.

ETHICS STATEMENT

Solving PDE is of crucial interest in many applications of science and engineering. While we do not directly target such real-world applications in this paper, one should acknowledge that solvers can be used in various ranges of scenarios including weather, climate, medical, aerodynamics, industry, and military applications.

ACKNOWLEDGMENTS

We acknowledge the financial support provided by DL4CLIM (ANR-19-CHIA-0018-01), DEEPNUM (ANR-21-CE23-0017-02), PHLUSIM (ANR-23-CE23-0025-02), and PEPR Sharp (ANR-23-PEIA-0008", "ANR", "FRANCE 2030").

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

## A  DETAILED RELATED WORK

**PDE solvers:**   Many tools for numerically solving PDEs have been developed for years. The standard methods for PDE include Finite Differences (FD), Finite Volume (FV), Finite Element Method (FEM), spectral and multigrid methods, and many others (S. H, 2012; Liu, 2009). While these methods are widely used, they often suffer from a high computational cost for complex problems or high-precision simulations. To address these challenges, integrating deep learning (DL) into solvers has emerged as a promising approach. Current solutions include incorporating correction terms into mathematical solvers to reduce numerical errors (Um et al., 2021). Some work such as Hsieh et al. (2019); Li et al. (2023a); Rudikov et al. (2024); Kopaničáková et al. (2023) build a method to directly enhance the convergence of numerical solvers through preconditioner learning. As an example, Rudikov et al. (2024); Li et al. (2023a) use a neural operator to approximate conditioner for the flexible conjugate gradient method or Hsieh et al. (2019) for the Jacobi method. Another example of preconditioner learning lies in Li et al. (2023a), where the author uses GNN to assess symmetry and positive definiteness.

**Unsupervised training:**   Physics-Informed Neural Networks (PINNs) (Raissi et al., 2019) have been a pioneering work in the development of DL method for physics. In these models, the solution is a neural network that is optimized using the residual loss of the PDE being solved. However, this method suffers from several drawbacks. First, as formulated in Raissi et al. (2019), PINNs can solve one instance of an equation at a time. Any small change in the parameters of the PDE involves a full retraining of the network. Efforts such as Beltrán-Pulido et al. (2022); Zhang et al. (2023); Cho et al. (2024) have attempted to address this limitation by introducing parametric versions of PINNs capable of handling parametric equations, while Huang et al. (2022); Qin et al. (2022) explores meta-learning approaches. Other approaches to improve PINNs generalization include using neural operators (Wang et al., 2021b), or hyper-network (de Avila Belbute-Peres et al., 2021). Moreover, PINNs have shown convergence difficulties: Krishnapriyan et al. (2021) show that PINNs' losses have complex optimization landscapes, complicating training despite adequate neural network expressiveness. Approaches like those detailed in Wang et al. (2022) adopt a Neural Tangent Kernel (NTK) perspective to identify reasons for failure and suggest using adaptive weights during training to enhance performance. Additionally, studies such as Ryck et al. (2023) demonstrate that PINNs suffer from ill-conditioned losses, resulting in slow convergence of gradient descent algorithms. Recently, (Rathore et al., 2024) shows how training strategies can improve the convergence of Physics-Informed Neural Networks and show that specific optimizers such as L-BFGS act as conditioners on the physical losses.

**Supervised training:**   In contrast to the unsupervised training of Physics-informed Neural Networks, purely data-driven models have demonstrated remarkable capabilities for PDE simulation and forecasting. In most of the existing literature, the entire solver is replaced by a DL architecture and focuses on directly computing the solution from a given input data. A widely studied setting is operator learning which learns mappings between function spaces (Li et al., 2020; Kovachki et al., 2023; Lu et al., 2021). This method is very efficient, with the downside of relying on quite large quantities of data for training in order to ensure adequate generalization. Additionally, the neural network does not have access to the PDE in itself, only indirectly through the data. To ensure physical constraints in purely data-driven training, hybrid models have been proposed. The latter relies on both the available physical knowledge and some data. Some examples include the Aphinity model (Yin et al., 2021) (where the authors assume partial knowledge of the physics and learned the remaining dynamics from data), Physics-informed Deep Operator Networks (PIDON) (Wang et al., 2021b; Goswami et al., 2022), Physics-informed Neural Operator (PINO) (Li et al., 2023b) (DeepONet architecture (Lu et al., 2021) or Neural Operator models (Kovachki et al., 2023; Li et al., 2020) respectively with a combination of data and physical losses).

**Learning to solve:**   Improving the learning scheme and optimizers through data-driven training has been studied since Li & Malik (2016) and Andrychowicz et al. (2016). These works propose to learn the optimizer of neural networks, which are classically optimized through gradient-based algorithms such as Adam. They focus on improving training strategies for neural networks, which do not suffer from the optimization issues and ill-conditioning properties of physics-informed losses. We refer the reader to the survey of Chen et al. (2021) for a complete overview. The closer work to ours is the very

recent work of Bihlo (2024) in which the author assesses the capabilities of learned optimizers for physics-informed neural networks. The main difference with our work relies on the problem setting. Bihlo (2024) considers learning an optimizer on a single equation, and for different neural networks initialization, while we focus on efficiently solving several instances of parametric PDE with varying PDE parameters $\gamma, f, g$.

# B   THEORETICAL ANALYSIS OF OUR METHOD AND PINNS

**Setting.**   We consider the following linear PDE:

$$
\begin{aligned}
\mathcal{D}u(x) &= f(x), & x \in \Omega, \\
u(x) &= g(x), & x \in \partial\Omega,
\end{aligned}
\tag{12}
$$

where $\Omega \subset \mathbb{R}^d$ is an open bounded domain, $\mathcal{D}$ is a linear differential operator, $f(x)$ is a given function in $\Omega$, and $g(x)$ is a given function on the boundary $\partial\Omega$.

## B.1   THEORETICAL ANALYSIS OF PINNS

Our aim is to find an approximate solution $u_\Theta(x)$, parameterized by $\Theta \in \mathbb{R}^N$, $\Theta = \{\theta_i\}_{i=1}^N$ that minimizes the loss function:

$$
\mathcal{L}_{\text{PDE}}(\Theta) = \mathcal{L}_{\text{Res}}(\Theta) + \lambda \mathcal{L}_{\text{BC}}(\Theta),
\tag{13}
$$

where:

$$
\mathcal{L}_{\text{Res}}(\Theta) = \frac{1}{2}\int_\Omega \left(\mathcal{D}u_\Theta(x) - f(x)\right)^2 dx, \quad \mathcal{L}_{\text{BC}}(\Theta) = \frac{1}{2}\int_{\partial\Omega}\left(u_\Theta(x) - g(x)\right)^2 dx,
$$

and $\lambda > 0$ is a regularization parameter balancing the PDE residual and boundary conditions.

We perform gradient descent updates with step size $\eta$. At step $k > 0$, updates write as:

$$
\Theta_{k+1} = \Theta_k - \eta\nabla_\Theta\mathcal{L}_{\text{PDE}}(\Theta_k).
$$

We establish the following theorem regarding the convergence rate of gradient descent.

**Theorem 2 (Convergence rate of PINNs).** *Given a linear ansatz $u_\Theta(x) = \sum_{i=1}^N \theta_i\phi_i(x)$, the number of steps $N(\varepsilon)$ required to achieve an error $\|\Theta_k - \Theta^*\|_2 \le \varepsilon$ satisfies:*

$$
N(\varepsilon) = O\left(\kappa(A)\ln\left(\tfrac{1}{\varepsilon}\right)\right),
\tag{14}
$$

*where $\kappa(A)$ is the condition number of the matrix $A \in \mathbb{R}^{n\times n}$ defined by:*

$$
A_{i,j} = \int_\Omega \left(\mathcal{D}\phi_i(x)\right)\left(\mathcal{D}\phi_j(x)\right) dx + \lambda\int_{\partial\Omega}\phi_i(x)\phi_j(x)dx.
\tag{15}
$$

*Proof.*   Since $u_\Theta(x) = \sum_{i=1}^N \theta_i\phi_i(x)$, we have:

$$
\frac{\partial u_\Theta(x)}{\partial\theta_i} = \phi_i(x), \quad \frac{\partial(\mathcal{D}u_\Theta(x))}{\partial\theta_i} = \mathcal{D}\phi_i(x).
$$

The gradient of the residual loss is:

$$
\nabla_\Theta\mathcal{L}_{\text{Res}}(\Theta) = \int_\Omega \left(\mathcal{D}u_\Theta(x) - f(x)\right)\mathcal{D}\phi(x)\, dx,
$$

where $\mathcal{D}\phi(x)$ is the vector with components $\mathcal{D}\phi_i(x)$. Similarly, the gradient of the boundary loss is:

$$\nabla_\Theta \mathcal{L}_{\text{BC}}(\Theta) = \int_{\partial\Omega} (u_\Theta(x) - g(x)) \, \phi(x) \, dx,$$

where $\phi(x)$ is the vector of basis functions evaluated at $x$. Therefore, the total gradient is:

$$\nabla_\Theta \mathcal{L}_{\text{PDE}}(\Theta) = A\Theta - b,$$

where the (positive, semi-definite) matrix $A$ and vector $b$ are defined as:

$$\begin{aligned}
A_{i,j} &= \int_\Omega (\mathcal{D}\phi_i(x)) \, (\mathcal{D}\phi_j(x)) \, dx + \lambda \int_{\partial\Omega} \phi_i(x)\phi_j(x) dx, \\
b_i &= \int_\Omega f(x)\mathcal{D}\phi_i(x) \, dx + \lambda \int_{\partial\Omega} g(x)\phi_i(x) \, dx.
\end{aligned} \tag{16}$$

Thus, the gradient descent update becomes:

$$\Theta_{k+1} = \Theta_k - \eta(A\Theta_k - b).$$

Subtracting $\Theta^*$ (the optimal parameter vector satisfying $A\Theta^* = b$) from both sides:

$$\Theta_{k+1} - \Theta^* = \Theta_k - \Theta^* - \eta A(\Theta_k - \Theta^*).$$

Simplifying:
$$\Theta_{k+1} - \Theta^* = (\text{Id} - \eta A)(\Theta_k - \Theta^*).$$

By recursively applying the update rule, we obtain:

$$\Theta_k - \Theta^* = (\text{Id} - \eta A)^k (\Theta_0 - \Theta^*).$$

Since $A$ is symmetric positive definite, it has eigenvalues $\lambda_1 \le \lambda_2 \le \cdots \le \lambda_n$ with $\lambda_i > 0$. To ensure convergence, we require $0 < \eta < \frac{2}{\lambda_{\max}(A)}$. Choosing $\eta = \frac{c}{\lambda_{\max}(A)}$ with $0 < c < 2$, we have:

$$1 - \eta\lambda_i = 1 - c\frac{\lambda_i}{\lambda_{\max}(A)}.$$

The spectral radius $\rho$ of $\text{Id} - \eta A$ is:

$$\rho = \max\left\{\left|1 - c\frac{\lambda_{\min}(A)}{\lambda_{\max}(A)}\right|, |1 - c|\right\} = \max\left\{1 - \frac{c}{\kappa(A)}, |1 - c|\right\},$$

where $\kappa(A) = \frac{\lambda_{\max}(A)}{\lambda_{\min}(A)}$ is the condition number of $A$. By choosing $0 < c < 1$, we ensure $|1 - c| < 1$, and since $\kappa(A) \ge 1$, we have $1 - \frac{c}{\kappa(A)} < 1$. Thus, the convergence factor is:

$$\rho = 1 - \frac{c}{\kappa(A)}.$$

Therefore:

$$\|\Theta_k - \Theta^*\|_2 \le \left(1 - \frac{c}{\kappa(A)}\right)^k \|\Theta_0 - \Theta^*\|_2.$$

To achieve $\|\Theta_k - \Theta^*\|_2 \le \varepsilon$, the number of iterations $N(\varepsilon)$ satisfies:

$$N(\varepsilon) \geq \frac{\ln\left(\varepsilon/\|\Theta_0 - \Theta^*\|_2\right)}{\ln\left(1 - \frac{c}{\kappa(A)}\right)}.$$

Using the inequality $\ln(1 - x) \leq -x$ for $0 < x < 1$, we get:

$$N(\varepsilon) \leq \frac{\kappa(A)}{c} \ln\left(\frac{\|\Theta_0 - \Theta^*\|_2}{\varepsilon}\right).$$

Thus:

$$N(\varepsilon) = O\left(\kappa(A) \ln\left(\frac{1}{\varepsilon}\right)\right).$$

$\square$

We have shown that for a linear ansatz $u_\Theta(x) = \sum_{i=1}^{N} \theta_i \phi_i(x)$, the convergence rate of gradient descent depends linearly on the condition number $\kappa(A)$ of the system matrix $A$. A large condition number impedes convergence, requiring more iterations to achieve a desired accuracy $\varepsilon$.

$\square$

### B.2 THEORETICAL ANALYSIS OF OUR METHOD

In practice, we often work with multiple data points. For each data point, there is an associated parameter vector $\Theta \in \mathbb{R}^N$. We are interested in the iterative update where the gradient is transformed by a neural network $\mathcal{F}$:

$$\Theta_{l+1} = \Theta_l - \eta \mathcal{F}\left(\nabla_\Theta \mathcal{L}_{\text{PDE}}(\Theta_l)\right), \tag{17}$$

where $\Theta_l$ represents the parameter vector at iteration $l$. Recall that $\mathcal{F}$ is trained to minimize the loss after $L$ iteration steps for $M$ data points:

$$\mathcal{L}_{\text{DATA}} = \frac{1}{m} \sum_{k=1}^{M} \left\| u_{\Theta_L^{(k)}} - u_k^* \right\|_2^2, \tag{18}$$

**Theorem 3.** *(**Convergence rate of our method**). Given a linear ansatz $u_\Theta(x) = \sum_{i=1}^{N} \theta_i \phi_i(x)$, assume $\mathcal{F}$ behaves like its linearization $P = \left.\frac{\partial \mathcal{F}}{\partial v}\right|_{v=0}$. The number of steps $N'(\varepsilon)$ required to achieve an error $\|\Theta_l - \Theta^*\|_2 \leq \varepsilon$ satisfies:*

$$N'(\varepsilon) = O\left(\kappa(PA) \ln\left(\frac{1}{\varepsilon}\right)\right), \tag{19}$$

*Moreover, if $\mathcal{F}$ minimizes $\mathcal{L}_{DATA}$ this necessarily implies $\kappa(PA) = 1 \leq \kappa(A)$. Consequently, the number of steps is effectively reduced, i.e., $N'(\varepsilon) \ll N(\varepsilon)$.*

*Proof.* Since $\mathcal{F}$ behaves like its linearization $P$, the gradient descent update becomes (refer to proof of Theorem 2 for steps):

$$\Theta_{l+1} = \Theta_l - \eta P(A\Theta_l - b).$$

Let $\Theta^*$ be the optimal parameter vector minimizing $\mathcal{L}_{\text{PDE}}$. Then, the difference between the parameter vector at iteration $l$ and the optimal parameter vector is:

$$\Theta_{l+1} - \Theta^* = \Theta_l - \Theta^* - \eta PA(\Theta_l - \Theta^*) = (\text{Id} - \eta PA)(\Theta_l - \Theta^*).$$

By recursively applying this update until the final step $L$, we obtain:

$$\Theta_L - \Theta^* = (\mathrm{Id} - \eta P A)^L (\Theta_0 - \Theta^*).$$

Since we have multiple data points, each with its own parameter vector, we consider the concatenation when necessary. Let's introduce $\Xi_l$ as the matrix whose columns are the parameter vectors:

$$\Xi_L = [\Theta_L^{(1)}, \Theta_L^{(2)}, \ldots, \Theta_L^{(m)}].$$

Similarly, $\Xi^*$ contains the optimal parameter vectors for each data point. The update for all data points can be written collectively:

$$\Xi_L - \Xi^* = (\mathrm{Id} - \eta P A)^L (\Xi_0 - \Xi^*).$$

Since $\mathcal{F}$ minimizes $\mathcal{L}_{\mathrm{DATA}}$, we have $\Xi_L = \Xi^*$, implying:

$$(\mathrm{Id} - \eta P A)^L (\Xi_0 - \Xi^*) = 0.$$

Given that the values of $\Xi_0$ are iid and sampled randomly from a continuous distribution, because the set of singular matrices has measure zero, the square matrix $(\Xi_0 - \Xi^*)(\Xi_0 - \Xi^*)^\top$ is full rank (i.e., invertible), with probability 1. Thus, the only way for the above equality to hold is if:

$$(\mathrm{Id} - \eta P A)^L = 0.$$

This means $\mathrm{Id} - \eta P A$ is nilpotent of index $L$. Consequently, all eigenvalues of $\mathrm{Id} - \eta P A$ are zero, implying that all eigenvalues of $PA$ are equal to $\dfrac{1}{\eta}$, leading to $\kappa(PA) = \lambda_{\max}(PA)/\lambda_{\min}(PA) = 1$, which is the optimal condition number. Referring to the convergence analysis in Theorem 2, we have:

$$N'(\varepsilon) \leq \frac{\kappa(PA)}{c} \ln\left( \frac{\|\Xi_0 - \Xi^*\|_2}{\varepsilon} \right).$$

Which directly implies

$$N'(\varepsilon) = O\left( \kappa(PA) \ln\left( \tfrac{1}{\varepsilon} \right) \right), \tag{20}$$

With $\kappa(PA) = 1$, this leads us to the desired result:

$$N'(\varepsilon) = O\left( \ln\left( \tfrac{1}{\varepsilon} \right) \right) \ll O\left( \kappa(A) \ln\left( \tfrac{1}{\varepsilon} \right) \right) =: N(\epsilon),$$

Thus, the number of iterations required is significantly reduced compared to the case without the neural network preconditioner.

$\square$

**Discussion**   The convergence proofs for our method fundamentally rely on the assumption of linearity in the underlying problems. It is important to note that the theoretical analysis does not extend to non-linear cases. Consequently, for non-linear scenarios, the theory should be viewed primarily as a tool for building intuition or providing motivation, rather than a definitive proof. This is due to the lack of established methods for rigorously studying the non-linear regime, as no known results currently address such cases.

Under these conditions:

- This optimal condition number implies that **convergence is not only guaranteed but also optimal, requiring fewer iterations.**

- **Guaranteed Convergence**: The method reliably achieves convergence to the optimal solution due to the reduced condition number.

- **Optimal Convergence Speed**: With $\kappa(PA) = 1$, the neural network provides an enhanced convergence rate, resulting in fewer required iterations compared to the original system without the neural network.

$\square$

## C    DATASET DETAILS

For all datasets, we kept $800$ samples for training and $200$ as testing examples (except otherwise stated in the experiments).

### C.1    HELMHOLTZ

We generate a dataset following the $1d$ static Helmholtz equation eq. (21). For $x \in [0, 1[$,

$$\begin{cases} u(x)'' + \omega^2 u(x) & = 0, \\ u(0) & = u_0, \\ u'(0) & = v_0. \end{cases} \tag{21}$$

The solution can be analytically derived: $u(x) = \alpha \cos(\omega x + \beta)$, with $\beta = \arctan(\frac{-v_0}{\omega u_0})$, $\alpha = \frac{u_0}{\cos(\beta)}$ and directly computed from the PDE data. We generate $1,024$ trajectories for training and $256$ for testing with $u_0, v_0 \sim \mathcal{N}(0, 1)$, and $\omega \sim \mathcal{U}(0.5, 50)$ and compute the solution on $[0, 1]$ with a spatial resolution of $256$. For training, we keep $800$ samples and use the complete dataset for the additional experiments presented in figs. 12a and 12b. Moreover, we sub-sample the spatial resolution by $4$ and keep $64$ points for training.

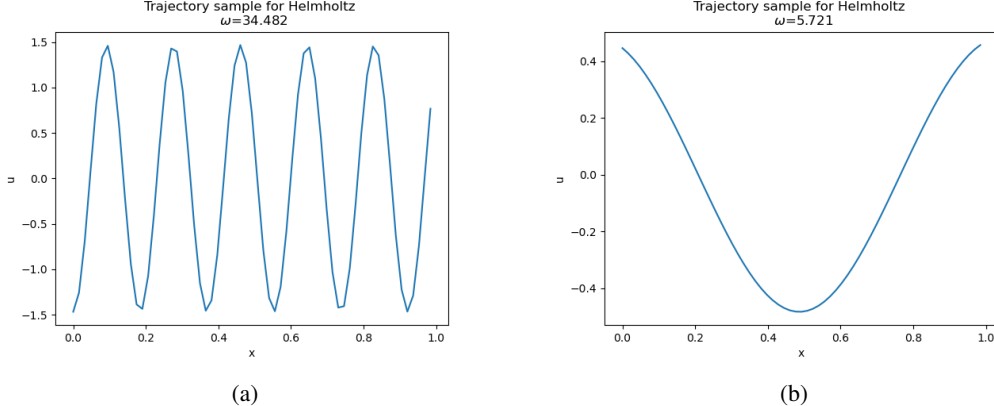

Figure 3: Samples from the Helmholtz Dataset.

### C.2    POISSON

We generate a dataset following the $1d$ static Poisson equation eq. (22) with forcing term. For $x \in [0, 1[$,

$$\begin{cases} -u''(x) & = f(x), \\ u(0) & = u_0, \\ u'(0) & = v_0. \end{cases} \tag{22}$$

We chose $f$ to be a non-linear forcing term: $f(x) = \frac{\pi}{K} \sum_{i=1}^{K} a_i i^{2r} \sin(\pi x)$, with $a_i \sim \mathcal{U}(-100, 100)$, we used $K = 16$, $r = -0.5$, and solve the equation using a backward finite difference scheme. We generate $1,000$ trajectories with $u_0, v_0 \sim \mathcal{N}(0, 1)$ and compute the solution on $[0, 1]$ with a spatial resolution of $64$.

**Reaction-Diffusion**    We use a non-linear reaction-diffusion used in (Krishnapriyan et al., 2021; Toloubidokhti et al., 2024). This PDE has been shown to be a failure case for PINNs (Krishnapriyan et al., 2021). The PDE states as follows:

$$\frac{\partial u(t,x)}{\partial t} - \nu \frac{\partial^2 u(t,x)}{\partial x^2} - \rho u(t,x)(1 - u(t,x)) = 0, \tag{23}$$

$$u(0,x) = e^{-32(x-1/2)^2}. \tag{24}$$

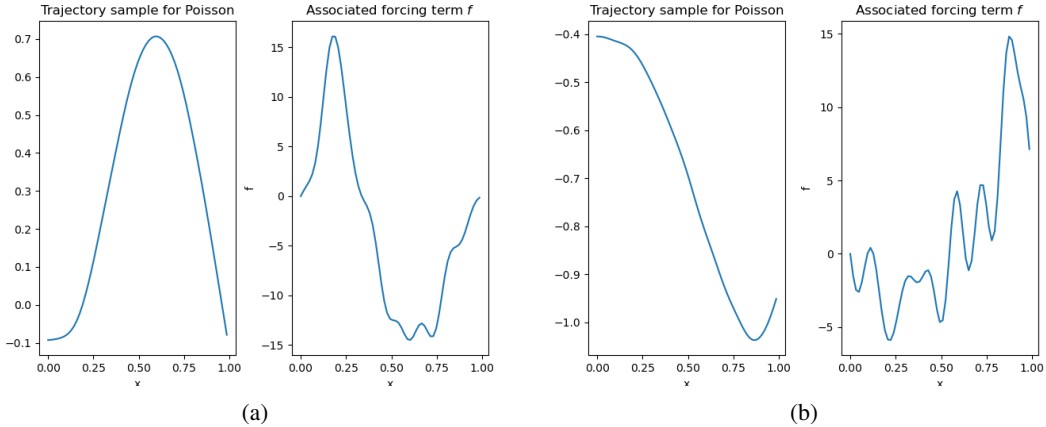

Figure 4: Samples from the Poisson Dataset.

We generate $800$ trajectories by varying $\nu$ in $[1, 5]$ and $\rho$ in $[-5, 5]$. Spatial resolution is $256$ and temporal resolution is $100$, which we sub-sample by $4$ for training, leading to a spatial resolution of $64 \times 25$. The PDE is solved on $[0, 1]^2$ as in (Toloubidokhti et al., 2024).

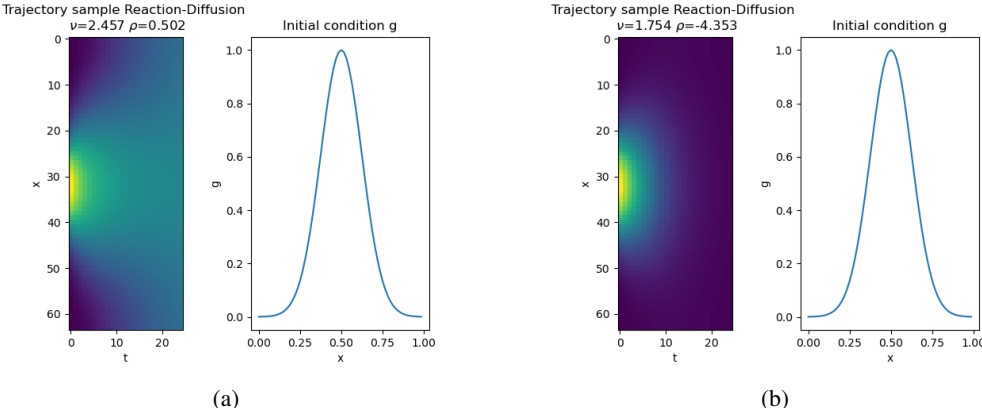

Figure 5: Samples from the Reaction-Diffusion Dataset.

**Reaction-Diffusion with initial conditions:**  To complexify the setting, we also change the initial condition of the problem (*NLRDIC* in the following). The initial condition is expressed as follows:

$$u(x, 0) = \sum_{i=1}^{3} a_i e^{-\frac{\left(\frac{x - h/4}{h}\right)^2}{4}}. \tag{25}$$

Where $a_i$ are randomly chosen in $[0, 1]$ and $h = 1$ is the spatial resolution.

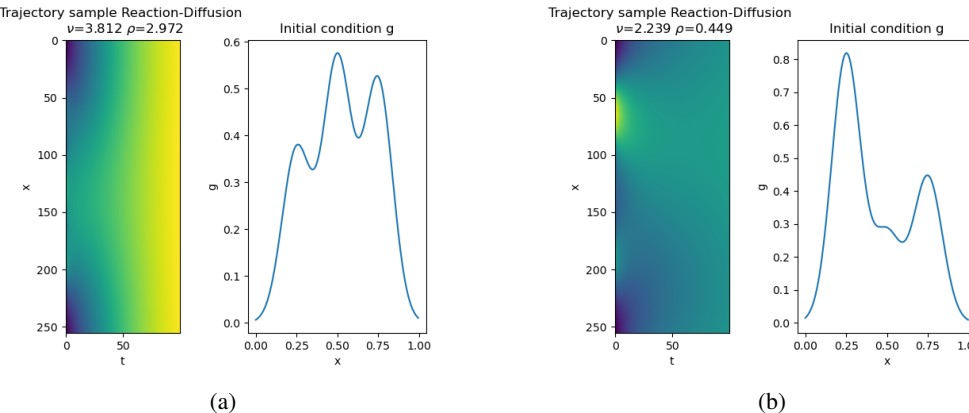

Figure 6: Samples from the Reaction-Diffusion Dataset with initial conditions changed.

## C.3 DARCY FLOW

The $2d$ Darcy Flow dataset is taken from (Li et al., 2020) and is commonly used in the operator learning literature (Li et al., 2023b; Goswami et al., 2022).

$$-\nabla.(a(x)\nabla u(x)) = f(x) \quad x \in (0,1)^2, \tag{26}$$

$$u(x) = 0 \quad x \in \partial(0,1)^2. \tag{27}$$

For this dataset, the forcing term $f$ is kept constant $f = 1$, and $a(x)$ is a piece-wise constant diffusion coefficient taken from (Li et al., 2020). We kept $1,000$ trajectories (on the $5,000$ available) with a spatial resolution of $64 \times 64$.

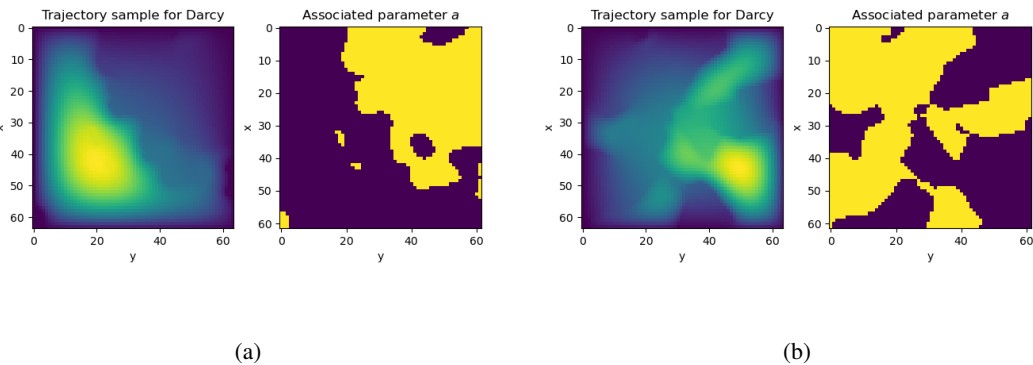

Figure 7: Samples from the Darcy Dataset.

## C.4 HEAT

As proof that our method can handle 2d + time, we consider the dataset proposed by (Zhou & Farimani, 2024).

$$\frac{\partial u(x,y,t)}{\partial t} - \nu\nabla^2 u(x,y,t) = 0, \tag{28}$$

$$u(x,y,0) = \sum_{j=1}^{J} A_j \sin(\frac{2\pi l_{xj}x}{L} + \frac{2\pi l_{yj}y}{L} + \phi_i). \tag{29}$$

Where $L = 2$, $\nu$ is randomly chosen between $[2 \times 10^{-3}, 2 \times 10^{-2}]$, $A_j$ in $[-0.5, 0.5]$, $l_{xj}, l_{xy}$ are integers in $\{1, 2, 3\}$ and $\phi$ is in $[0, 2\pi]$. As a difference with (Zhou & Farimani, 2024), we randomly

chose $J$ between $1$ and $J_{max} = 5$ to have more diversity in the represented frequencies in the data. The PDEs are sampled with spatial resolution $64$ in x and y and temporal resolution $100$. However, during training the spatial resolution are subsampled by $4$ and the coordinates are re scaled between $0$ and $1$. As for other PDEs, we use $800$ trajectories for training and $200$ for testing.

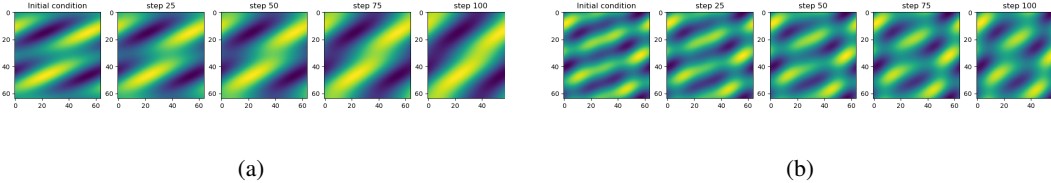

(a)                                          (b)

Figure 8: Samples from the Heat dataset.

## C.5 ADDITIONAL DATASET: ADVECTION

The dataset is taken from (Takamoto et al., 2023).

$$\frac{\partial u(t,x)}{\partial t} + \beta \frac{\partial u(t,x)}{\partial x} = 0, \quad x \in (0,1), t \in (0,2], \tag{30}$$

$$u(0,x) = u_0(x), \quad x \in (0,1). \tag{31}$$

Where $\beta$ is a constant advection speed, and the initial condition is $u_0(x) = \sum_{k_i = k_1 \ldots k_N} A_i \sin(k_i x + \phi_i)$, with $k_i = \frac{2\pi n_i}{L_x}$ and $n_i$ are randomly selected in $[1,8]$. The author used $N = 2$ for this PDE. Moreover, $A_i$ and $\phi_i$ are randomly selected in $[0,1]$ and $(0,2\pi)$ respectively. Finally, $L_x$ is the size of the domain (Takamoto et al., 2023).

The PDEBench's Advection dataset is composed of several configurations of the parameter $\beta$ ($\{0.1, 0.2, 0.4, 0.7, 1, 2, 4, 7\}$), each of them is composed of $10,000$ trajectories with varying initial conditions. From these datasets, we sampled a total of $1,000$ trajectories for $\beta \in \{0.2, 0.4, 0.7, 1, 2, 4\}$ (which gives about $130$ trajectories for each $\beta$). This gives a dataset with different initial conditions and parameters. Moreover, during training, we sub-sampled the trajectories by $4$, leading to a grid of resolution $25$ for the t-coordinate and $256$ for the x-coordinate.

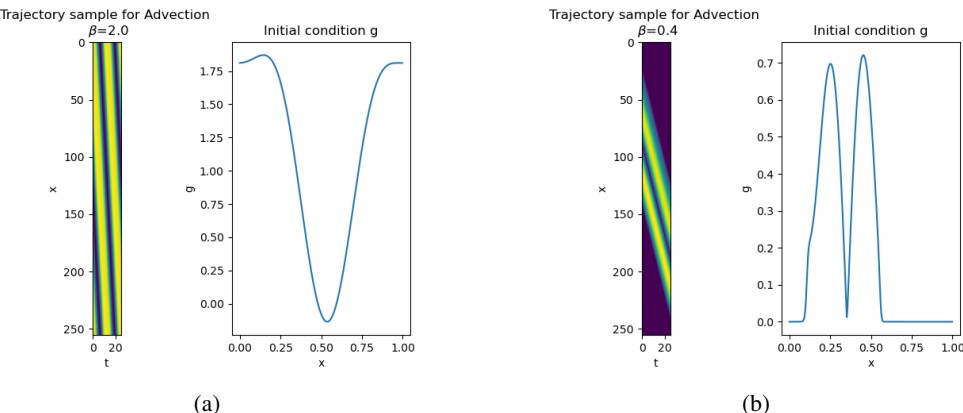

(a)                                          (b)

Figure 9: Samples from the Advections Dataset.

## C.6 SUMMARY OF PROBLEM SETTINGS CONSIDERED

A summary of the datasets and parameters changing between 2 trajectories is presented in table 3.

Table 3: Parameters changed between each trajectory in the considered datasets in the main part of the paper as well as additional datasets (Advections and NLRDIC).

| Dataset | Changing PDE data | Range / Generation |
|---|---|---|
| Helmholtz | $\omega$ | $[0.5, 50]$ |
| | $u_0$ | $\mathcal{N}(0, 1)$ |
| | $v_0$ | $\mathcal{N}(0, 1)$ |
| Poisson | $A_i$ | $[-100, 100]$ |
| | $u_0$ | $\mathcal{N}(0, 1)$ |
| | $v_0$ | $\mathcal{N}(0, 1)$ |
| Reaction-diffusion | $\nu$ | $[1, 5]$ |
| | $\rho$ | $[-5, 5]$ |
| Darcy | $a(x)$ | $\psi_{\#}\mathcal{N}(0, (-\Delta + 9I)^{-2})$ with $\psi = 12 * \mathbb{1}_{\mathbb{R}_+} + 3 * \mathbb{1}_{\mathbb{R}_+}$ |
| Heat | $\nu$ | $[2 \times 10^{-3}, 2 \times 10^{-2}]$ |
| | $J_{max}$ | $\{1, 2, 3, 4, 5\}$ |
| | $A$ | $[0.5, -0.5]$ |
| | $K_x, K_y$ | $\{1, 2, 3\}$ |
| | $\phi$ | $[0, 2\pi]$ |
| Advection | $\beta$ | $\{0.2, 0.4, 0.7, 1, 2, 4\}$ |
| | $A_i$ | $[0, 1]$ |
| | $\phi_i$ | $[0, 2\pi]$ |
| | $k_i$ | $\{2k\pi\}_{k=1}^{8}$ |
| NLRDIC | $\nu$ | $[1, 5]$ |
| | $\rho$ | $[-5, 5]$ |
| | $a_i$ | $[0, 1]$ |

# D   IMPLEMENTATION DETAILS

We add here more details about the implementation and experiments presented in section 4.

We implemented all experiments with PyTorch (Ansel et al., 2024). We estimate the computation time needed for development and the different experiments to be approximately 300 days.

## D.1   B-SPLINE BASIS

We chose to use a B-Spline basis to construct the solution. We manually build the spline and compute its derivatives thanks to the formulation and algorithms proposed in (Piegl & Tiller, 1996). We used Splines of degree $d = 3$ and constructed the Splines with 2 different configurations:

- Take $N + d + 1$ equispaced nodes of multiplicity 1 from $\frac{d}{N}$ to $1 + \frac{d}{N}$. This gives a smooth local basis with no discontinuities (see fig. 10a) represented by a shifted spline along the x-axis (denoted as `shifted` in the following).

- Use $N + 1 - d$ nodes of multiplicity 1 and 2 nodes of multiplicity $d$ (typically on the boundary nodes: 0 and 1). This means that nodes 0 and 1 are not differentiable (see fig. 10b). We call this set-up `equispaced`.

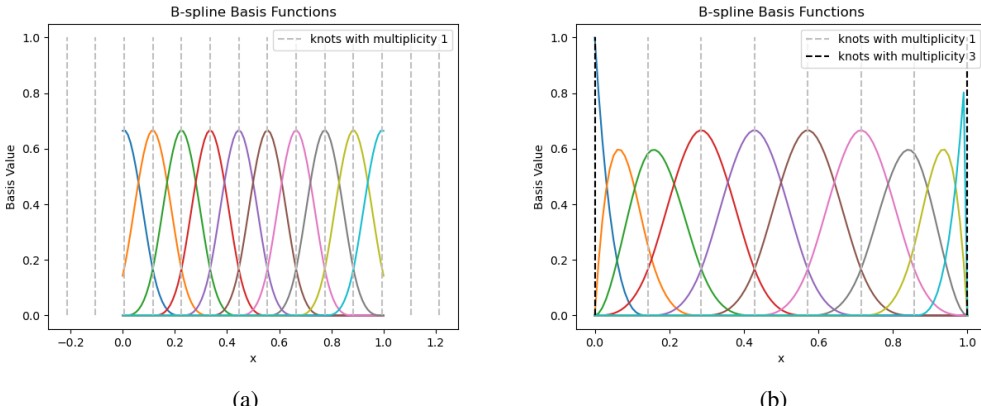

(a)                                                                                          (b)

Figure 10: B-spline basis with $N = 10$ terms with shifted spline (Left) and higher multiplicity nodes (Right). Dashed lines represent nodes' position with color the darker, the higher the multiplicity.

**Higer-dimension basis**   For 1d + time, 2d dataset, and 2d + time we build a 2 (or 3) dimensional B-spline basis, *i.e.*, we treat the time coordinates as spatial ones. To build such bases, we compute the Cartesian product between the 2 (or 3) 1d-bases, 1 per dimension. This means that for a 2d dataset, for which we chose to use bases of size $N1$ and $N2$ for each coordinate, the resulting basis will have $N1 + N2 + N1 * N2$ terms. This method makes the training more costly and several techniques to improve its scalability could be used. For 3d datasets, the number of terms in the basis is cubic.

## D.2   TRAINING DETAILS

In our experiments, neural networks are trained using the Adam optimizer. For network optimization, we employ a smooth $l1$-loss for our solver while for other baselines, we use MSE loss and/or physical losses. All models are trained for at least $1,500$ epochs on datasets composed of some sampling of $\gamma$ and/or $f$ and/or $g$. If not stated otherwise, we train our proposed method for 750 epochs and baselines for $1,500$ epochs. We use the Adam optimizer and an initial learning rate of $0.001$. We use an exponential learning rate scheduler that lowers the learning rate by $0.995$ every epoch. Experiments are conducted on NVIDIA TITAN V (12 Go) for 1d datasets and NVIDIA RTX A6000 GPU with 49Go for $1d$ + time, 2d datasets or 2d + time. We recall in algorithm 3 the pseudo-code for training our proposed method.

---

**Algorithm 3:** Training algorithm for learning to optimize physics-informed losses.

---

Data: $\Theta_0 \in \mathbb{R}^n$, PDE $(\gamma, f, g)$, sample values $u(x)$
Result: $\mathcal{F}_\varrho$
for $e = 1...$ epochs do
    for (PDE, x, u) in dataset do
        Initialize $\Theta_0$
        Estimate $\Theta_L$ from $\Theta_0, (\gamma, f, g)$ using Algorithm 1
        Reconstruct $u_{\Theta_L}(x)$
        Update the parameters $\varrho$ of $\mathcal{F}$ with gradient descent from the data loss in Equation 10.
    end
end
return $\mathcal{F}_\varrho$

---

We make use of two nested components: the solver for providing the approximate solution to the PDE and the optimizer that conditions the training of the solver. Both are using gradient descent but with different inputs and objectives. The former optimizes the PDE loss (inner loop), while the latter optimizes the gradient steps of the solver through conditioning (outer loop). Optimization of the two components proceeds with an alternate optimization scheme. In particular, this implies that the map $F_\varrho$ is kept fixed during the inner optimization process.

### D.3 MODELS

We present here the training details for our model and the baseline. The results can be found in table 2. In all our experiments, we use a GeLU (Hendrycks & Gimpel, 2016) activation function. The details of the model architecture on each dataset are presented in table 4.

**Hyperparameters for Helmholtz:** For the baselines, we empirically searched hyperparameters to allow the network to handle the high frequencies involved in the *Helmholtz* dataset. Unfortunately, other network architectures did not improve the results. We proceed similarly for other datasets.

**Training of PPINNs:** For the Helmholtz dataset, we trained our model for 2000 epochs with a plateau learning rate scheduler with patience of 400 epochs. For the Poisson dataset, we trained our parametric PINN model for 5000 epochs with a cosine annealing scheduler with a maximum number of iterations of 1000. For the Reaction-diffusion dataset, we consider an initial learning rate of 0.0001 instead of 0.001. Finally, *P2INNs* is trained for $5,000$ epochs using the Adam optimizer and an exponential scheduler with patience 50.

**PINNs baselines (PINNs + L-BFGS and PINNs-multi-opt):** The conditioning of the problem highly depends on the parameters of the PDE and initial/boundary conditions. This can lead to unstable training when optimizing PINNs, requiring specific parameter configurations for each PDE. To avoid extensive research of the best training strategy, we found a configuration that allowed a good fitting of **most of the testing dataset**. This means that in the values reported in table 2, we removed trainings for which the losses exploded (only a few hard PDEs were removed, typically between none to 20). This prevents us from extensive hyper-parameter tuning on each PDE. Please note that this lowers the reported relative MSE, thus advantaging the baseline. We detail in tables 5 and 6 the hyper-parameters for each dataset.

**Basis configuration:** For all datasets, we use Splines of degree 3, built with `shifted` nodes. We change the number of terms in the basis depending on the problem considered. For the $1d$-problem, we use 32 terms. For the $2d$-problem, 40 elements are in the basis of each dimension, except for *Reaction-Diffusion* where the variable t has 20 terms. Moreover, during training, we use the projection of the initial conditions and/or parameters and/or forcing terms function in the basis as input to the networks. Finally, for experiments using the *Heat* dataset (*i.e.* 3d basis), we used 15 terms for the $x$ and $y$ spatial coordinates and 10 terms for the $t$-coordinate.

Table 4: Architecture details of our model and baselines

| Model | Architecture | Dataset | | | | |
|---|---|---|---|---|---|---|
| | | 1d | | 1d + time | 2d | 2d + time |
| | | Helmholtz | Poisson | NLRD | Darcy | Heat |
| PINNs | MLP depth | 3 | 3 | 3 | 3 | 3 |
| | MLP width | 256 | 256 | 256 | 256 | 256 |
| PPINNs | MLP depth | 8 | 3 | 5 | 3 | 3 |
| | MLP width | 64 | 256 | 256 | 256 | 256 |
| PO-DeepONet | Branch Net depth | 3 | 2 | 5 | 5 | 5 |
| | Branch Net width | 256 | 256 | 256 | 256 | 256 |
| | Trunk Net depth | 3 | 2 | 5 | 5 | 5 |
| | Trunk Net width | 256 | 256 | 256 | 256 | 256 |
| P2INNs | Enc params depth | 4 | 4 | 4 | 4 | 4 |
| | Enc params width | 256 | 256 | 256 | 256 | 256 |
| | Emb params | 128 | 128 | 128 | 128 | 128 |
| | Enc coord depth | 3 | 3 | 3 | 3 | 3 |
| | Enc coord width | 256 | 256 | 256 | 256 | 256 |
| | Emb coord | 128 | 128 | 128 | 128 | 128 |
| | Dec depth | 6 | 6 | 6 | 6 | 6 |
| | Dec width | 256 | 256 | 256 | 256 | 256 |
| | Activations | GeLU | GeLU | GeLU | GeLU | GeLU |
| MLP + basis | MLP depth | 5 | 5 | 5 | 5 | 5 |
| | MLP width | 256 | 256 | 1,024 | 1,024 | 1,024 |
| PI-DeepONet | Branch Net depth | 3 | 2 | 5 | 5 | 5 |
| | Branch Net width | 256 | 256 | 256 | 256 | 5 |
| | Trunk Net depth | 3 | 2 | 5 | 5 | 5 |
| | Trunk Net width | 256 | 256 | 256 | 256 | 256 |
| PINO | FNO depth | 3 | 3 | 4 | 3 | 3 |
| | FNO width | 64 | 64 | 64 | 64 | 64 |
| | FNO modes 1 | 16 | 16 | 10 | 20 | 7 |
| | FNO modes 2 | - | - | 5 | 20 | 7 |
| | FNO modes 3 | - | - | - | - | 5 |
| | FNO fc dim | 64 | 64 | 64 | 64 | 64 |
| Ours | FNO depth | 3 | 3 | 3 | 3 | 3 |
| | FNO width | 64 | 64 | 64 | 64 | 64 |
| | FNO modes 1 | 16 | 16 | 10 | 20 | 7 |
| | FNO modes 2 | - | - | 5 | 20 | 7 |
| | FNO modes 3 | - | - | - | - | 5 |
| | FNO fc dim | 64 | 64 | 64 | 64 | 64 |

Table 5: Hyperparameters for *PINNs+L-BFGS* baseline.

| Hyper-parameter | Dataset | | | | |
|---|---|---|---|---|---|
| | 1d | | 1d + time | 2d | 2d + time |
| | Helmholtz | Poisson | NLRD | Darcy | Heat |
| epochs | 1,000 | 1,000 | 1,500 | 1,500 | 1,000 |
| learning rate | 1e-4 | 1e-5 | 1e-4 | 1e-3 | 1e-3 |

Table 6: Hyperparameters for *PINNs-multi-opt* baseline.

| Hyper-parameter | Dataset | | | | |
|---|---|---|---|---|---|
| | 1d | | 1d + time | 2d | 2d + time |
| | Helmholtz | Poisson | NLRD | Darcy | Heat |
| epochs Adams | 800 | 800 | 800 | $1,200$ | $1,200$ |
| epochs L-BFGS | 200 | 200 | 200 | 300 | 300 |
| epochs total | $1,000$ | $1,000$ | $1,000$ | $1,500$ | $1,500$ |
| learning rate Adam | 1e-4 | 1e-5 | 1e-4 | 1e-3 | 1e-3 |
| learning rate L-BFGS | 1 | 1 | 1e-3 | 1 | 1 |

# E ADDITIONAL EXPERIMENTS

## E.1 ABLATION

We experimentally show some properties of our iterative method. These evaluations are made on a test set composed of several instances of PDE with varying configurations $(\gamma, f, g)$ that **are unseen during training**. These experiments are performed on the *Helmholtz* equation which appeared as one of the most complex to optimize in our evaluation. We used $L = 5$ in our method for a better visualization, unless stated otherwise.

**Error w.r.t the number of steps L**  On fig. 11, we show that having more optimizer steps allows for a better generalization. However, we observed in our experiments that the generalization error stabilizes or even increases after the proposed 5 steps. This limitation of the solver should be investigated in future work in order to allow the model to make more iterations. Note that for this experiment, we lowered batch size (and adapted the learning rate accordingly) when the number of steps increased (fig. 11).

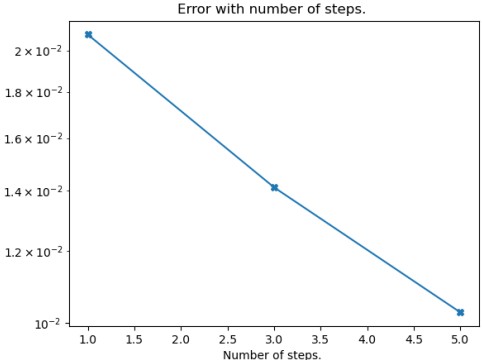

Figure 11: Error on the test set (*Helmholtz* equation) *w.r.t* the number of iterations

**Error w.r.t the number of training samples**  On figs. 12a and 12b, we show that compared to other physics-informed baselines, our solver requires less data to learn to solve PDE. Note that in this simple example, the *MLP+basis* baseline also performed well. However, as shown in table 2, this is not the case for all the datasets. The contribution of the iterative procedure clearly appears since with $2\times$ less data, our model performs better than this baseline.

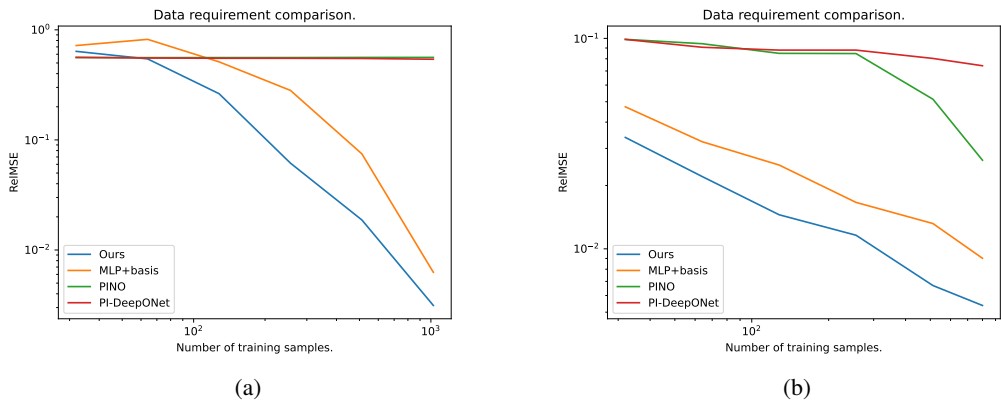

(a)                                                      (b)

Figure 12: Relative MSE on the test set *w.r.t* the number of training samples on *Helmholtz* (left) and *Darcy* (right) datasets.

**Optimization with $\mathcal{L}_{\text{PDE}}$**     Table 7 shows that optimizing our network with physical loss greatly complicates training. Indeed, adding an ill-conditioned loss to the standard MSE makes training ill-conditioned.

Table 7: How the physical loss complexifies training and lowers performances. Metrics in Relative MSE on the test set of *Helmholtz* equation.

| Input | Relative MSE |
|---|---|
| $\mathcal{L}_{\text{PDE}} + \mathcal{L}_{\text{DATA}}$ | 1.11 |
| $\mathcal{L}_{\text{DATA}}$ | **2.19e-2** |

**Iterative update & SGD-based update**     In table 8, we compare two optimizer configurations. "Direct" means that the network directly predicts the parameters for the next step (*i.e.* $\Theta_{k+1} = \mathcal{F}_\varrho(\nabla\mathcal{L}_{\text{PDE}}(\Theta_k), \gamma, f, g)$), while "GD", corresponds to the update rule described before and using SGD as the base algorithm $\Theta_{k+1} = \Theta_k - \eta\mathcal{F}_\varrho(\nabla\mathcal{L}_{\text{PDE}}(\Theta_k), \gamma, f, g)$. The latter clearly outperforms the direct approach and shows that learning increments is more efficient than learning a direct mapping between two updates. As shown in table 8 increasing the number of steps improves the performance (shown here for 1 and 5 update steps). However, the performance does not improve anymore after a few steps (not shown here).

Table 8: Comparison of different optimizer configurations for solving the *Helmholtz* equation. Metrics in Relative MSE on the test set.

| | Relative MSE | |
|---|---|---|
| N-steps | Direct | GD |
| 1-step | 1.08e-1 | 8.5e-2 |
| 5-step | 9.07e-2 | **2.19e-2** |

**Optimization with different inner learning rates**     In table 9, we study the performance of our proposed method with different inner learning rates $\eta$. As expected, a higher learning rate leads to better performances since the optimization is taking bigger steps.

Table 9: Ablation on the inner learning rate. Metrics in Relative MSE on the test set of *Helmholtz* equation.

| learning rate | Relative MSE |
|---|---|
| 0.01 | 7.32e-2 |
| 0.1 | 4.93e-2 |
| 1 | **2.19e-2** |

**Quantifying the importance of input feature for the learned solver**     As indicated in eq. (9) the inputs of our learned solver are $(\gamma, f, g, \nabla\mathcal{L}_{\text{PDE}}(\theta_L))$. We performed experiments by removing either $\gamma$ or $\nabla\mathcal{L}_{\text{PDE}}(\theta_l)$ from the input (For the *Helmholtz* equation, there is no forcing term $f$). The BC $g$ are kept since they are part of the PDE specification and are required to ensure the uniqueness of the solution. This experiment (table 10) illustrates that conditioning on the PDE parameters $\gamma$ is indeed required to solve the parametric setting. Without $\gamma$, the solver has no hint on which instance should be used. The addition of the gradient information, $\nabla$, which is at the core of our method, is also crucial for improving the convergence and validates our setting.

Table 10: Effect of using the gradient as input *w.r.t* the PDE parameters. Metrics in Relative MSE on the test set of *Helmholtz* equation.

| Input | Relative MSE |
|---|---|
| $\gamma + g$ | 3.75e-1 |
| $\nabla + g$ | 1.07e-1 |
| $\nabla + \gamma + g$ | **2.19e-2** |

We showed by a simple experiment that our model can handle nonlinear cases (see table 11). We propose to model the solution $u$ using a *non-linear combination of the basis terms* $\phi_i$. The relation between the $\phi_i$ is modeled using a simple NN with one hidden layer and a `tanh` activation function. This experiment is performed on the *Poisson* PDE.

Table 11: Nonlinear combination of the basis. Relative MSE on the test set for our proposed method and comparison with other non linear models and optimizers.

| baselines | Relative MSE |
|---|---|
| PINNs+L-BFGS | 8.83e-1 |
| PPINNs | 4.30e-2 |
| Ours | **3.44e-3** |

**Network architecture** We show in table 12, an ablation on the layer type used for $\mathcal{F}_\varrho$ in our experiments: MLP, Residual Network (ResNet), FNO and a modified version of a MLP (ModMLP) taken from (Wang et al., 2021a), inspired from attention mechanism. We conducted this experiment on the Helmholtz dataset.

Table 12: Ablation on different layer types. Metric on the test set of the *Helmholtz* equation.

| Layer type | Relative MSE |
|---|---|
| MLP | 8.25e-1 |
| ResNet | 6.87e-1 |
| ModMLP | 5.55e-2 |
| FNO | **2.41e-2** |

**Irregular grids** We show in table 13, an ablation on different types of grids: regular as in table 2, and irregular. The latter were created by sampling uniformly 75% of the points in the original grid. The sampled grids **are different between each trajectory both during the training and testing phases**. We conducted this experiment on the Helmholtz dataset. Finally, we show some reconstruction examples in figs. 13a and 13b.

Table 13: Comparison of the performances when training our solver using regular or irregular and different grids for each PDE. Metrics on the test set.

| Grid | Relative MSE |
|---|---|
| regular | 2.41e-2 |
| irregular | 3.38e-1 |

This experiment shows that our method is capable of handling irregular grids. We observe a decrease in the performances (table 13) and in the reconstruction quality figs. 13a and 13b. However, we also note that our method is still capable of reconstructing the dynamic of the PDE, where most of the considered baselines failed to capture the oscillation in the Helmholtz PDE solution.

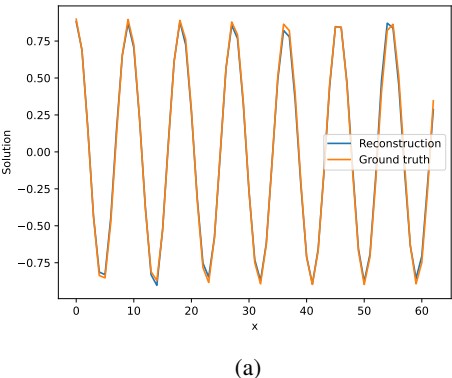 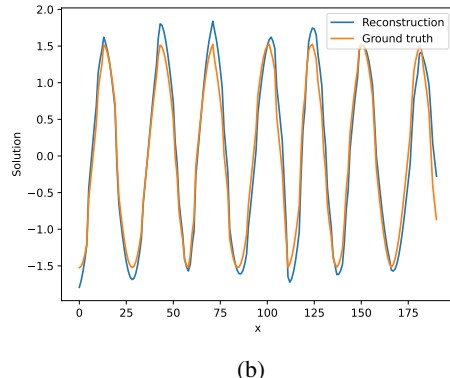

(a)                                                    (b)

Figure 13: Comparison of the reconstruction of the solution when models are trained on regular (left) and irregular grid (right).

**Error as a function of the PDE parameters values** We illustrate in fig. 14 the behavior of the reconstruction of the MSE varying the PDE parameter values. We conducted this experiment on the Helmholtz PDE and varied $\omega$ from $-5$ to $55$ *i.e.* extrapolation of $10\%$ beyond the parameter distribution (and kept fixed boundary conditions).

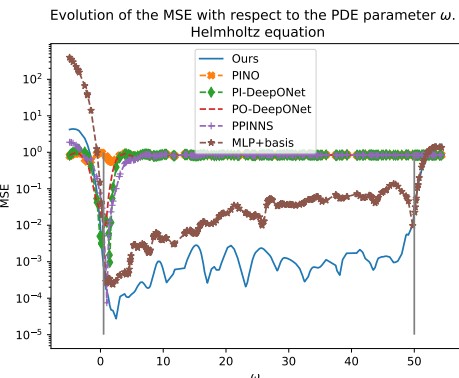

Figure 14: MSE comparison with PDE parameters $\omega$ of the Helmholtz dataset and extrapolation outside of the training distribution.

We observe in fig. 14 that, as expected, solving Helmholtz PDE outside the training distribution fails for our model as well as for all baselines. The proposed method performs well inside the training distribution (as already observed in table 2) and behaves similarly to fully supervised methods on Out-Of-Distribution (OOD) examples. Finally, other baselines, involving physical only or hybrid training, only predict a mean solution.

### E.2   STATISTICAL ANALYSIS

We provide a statistical analysis on some datasets and baselines. These experiments are conducted using $L = 5$ (instead of $L = 2$ in table 2). For computational cost reasons, we did not make this experiment on all datasets but kept 2 datasets in $1d$ (table 14), 1 dataset in 1d+time, and 1 dataset in 2d (table 15) so that several configurations and data sizes are represented.

This analysis shows the robustness of our proposed method *w.r.t.* initial seed.

Table 14: Results of trained models with error bars (std errors) on 1d datasets - metrics in Relative MSE on test set. Best performances are highlighted in **bold**, and second best are underlined.

| | | 1d | |
| --- | --- | --- | --- |
| | Baseline | Helmholtz | Poisson |
| Supervised | *MLP + basis* | 5.26e-2 $\pm$ 7.56e-3 | 1.58e-1 $\pm$ 7.98e-3 |
| Unsupervised | *PPINNs* | 8.33e-1$\pm$5.61e-3 | 3.59e-2$\pm$2.11e-2 |
| | *PO-DeepONet* | 9.84e-1$\pm$6.93e-4 | 1.79e-1$\pm$3.09e-2 |
| Hybrid | *PI-DeepONet* | 9.81e-1 $\pm$ 2.25e-3 | 1.25e-1 $\pm$1.04e-2 |
| | *PINO* | 9.95e-1 $\pm$ 3.30e-3 | 3.27e-3 $\pm$ 1.38e-3 |
| | *Ours* | **2.17e-2 $\pm$ 1.12e-3** | **4.07e-5 $\pm$ 2.65e-5** |

Table 15: Results of trained models with error bars (std errors) on 1d + time, and 2d datasets - metrics in Relative MSE on test set. Best performances are highlighted in **bold**, and second best are underlined.

| | | 1d + time | 2d |
| --- | --- | --- | --- |
| | Baseline | 1dnlrd | Darcy-Flow |
| Supervised | *MLP + basis* | 2.83e-5 $\pm$ 6.83e-7 | 3.78e-2$\pm$2.09e-3 |
| Unsupervised | *PPINNs* | 4.64e-1 $\pm$ 1.92e-2 | 9.99e-1$\pm$2.63e-2 |
| | *PO-DeepONet* | 4.18e-1 $\pm$ 1.04e-2 | 8.32e-1$\pm$2.51e-4 |
| Hybrid | *PI-DeepONet* | 7.88e-2 $\pm$ 1.96e-4 | 2.72e-1 $\pm$ 4.44e-3 |
| | *PINO* | 8.00e-5 $\pm$ 1.00e-5 | 1.17e-1 $\pm$ 1.42e-2 |
| | *Ours* | **2.61e-5 $\pm$ 2.53e-6** | **1.62e-2 $\pm$ 3.06e-4** |

### E.3 COMPUTATIONAL COST

Finally, we detail here the training and inference times of our method as well as baselines.

As we can see in table 16, our model takes longer to train due to the iterative process occurring at each epoch. However, note that this is training time; inference time is similar to other methods (see table 17). We detail a justification for this additional training time compared to PINNs variants below:

- Comparison with vanilla PINNs: Consider the following experiment. Suppose we train a classical PINN on a single instance of the Darcy PDE. Based on the training times shown in table 16, using this method, we performed $1,500$ steps, which took $420$ minutes for training (please note that the performances were less accurate than our model's performance). If we wanted to train a PINNs on each PDE of our entire test dataset for $15000$ epochs (sometimes even more steps are required), this would take $4200$ minutes or stated otherwise approximatly 3 days. Suppose now, that one wants to solve an additional equation. This will require an average of $0.226$ seconds (see table 17) with our method, while PINNs would require an entirely new training session of approximately $20$ minutes for $15,000$ steps. This makes our method $5,000$ times faster than traditional PINNs for solving any new equation.

- Comparison with PINNs parametric variants: Now let us consider two parametric variants of PINNs designed to handle multiple PDEs (PPINNs for parametric PINNs, P2INNs for the model proposed by (Cho et al., 2024)). In table 2, models was trained for $5,000$ epochs only. Let us consider training it further as suggested for vanilla PINNs, until $15,000$ epochs. First, this training would require approximately 19h30m. Then, the optimization problem is still ill-conditioned, training further would probably not significantly improve the performance. We can extend this reasoning to the *P2INNs* baseline, for which we observed similar performance and behaviors.

Table 16: Training time of the experiments shown in Table 2. of the paper on a single NVIDIA TITAN RTX (25 Go) GPU. d stands for days, h for hours, m for minutes.

| Dataset | Helmholtz | Poisson | 1dnlrd | Darcy | Heat |
|---|---|---|---|---|---|
| MLP + basis | 30m | 20m | 1h10m | 2h | 4h45 |
| PPINNs | 15m | 20m | 4h15m | 6h30m | 1d2h |
| P2INNs | 2h | 3h | 11h | 1d7h | 1d8h |
| PODON | 10m | 10m | 3h30m | 1d9h | 22h |
| PIDON | 10m | 10m | 3h30m | 1d10h | 22h |
| PINO | 15m | 10m | 1h10m | 45m | 2h40 |
| Ours | 30m | 30m | 4h30 | 10h15 | 1d 13h |

Table 17: Inference time, averaged (in seconds) on the test set. All experiments are conducted on a single NVIDIA RTX A6000 (48Go). We report the mean time computed on the test set to evaluate the baselines as performed in table 2 in the paper (*i.e.* with 10 test-time optimization steps when applicable and 20 steps on the Heat dataset). We consider as inference the solving of a PDE given its parameters and/or initial/boundary conditions.

| Dataset | Helmholtz | Poisson | 1dnlrd | Darcy | Heat |
|---|---|---|---|---|---|
| MLP + basis | 1.12e-2 | 1.18e-2 | 1.25e-2 | 1.19e-2 | 1.66e-2 |
| PINNs+L-BFGS | 274 | 136 | 369 | 126 | 234 |
| PINNs-multi-opt | 15.5 | 25.5 | 16.5 | 105 | 90 |
| PPINNs | 3.09e-1 | 2.03e-1 | 2.91e-1 | 3.22e-1 | 5.45e-1 |
| P2INNs | 2.84e-1 | 3.09e-1 | 6.76e-1 | 1.29 | 1.23 |
| PODON | 3.27e-1 | 2.71e-1 | 4.38e-1 | 6.32e-1 | 8.85e-1 |
| PIDON | 3.32e-1 | 2.96e-1 | 4.43e-1 | 6.35e-1 | 8.80e-1 |
| PINO | 3.14e-1 | 1.24e-1 | 5.19e-1 | 2.21e-1 | 8.08e-1 |
| Ours | 2.58e-1 | 2.16e-1 | 2.84e-1 | 2.26e-1 | 2.90e-1 |

## E.4 ADDITIONAL DATASETS

We provide additional experiments on 2 new datasets: Non-linear Reaction-Diffusion in a more complex setting and Advections. We refer the reader to appendix C for the details about the PDE setting. These datasets were not included in the main part of the paper due to a lack of space in the results table.

Table 18: Results of trained models on additional datasets - metrics in Relative MSE on the test set. Best performances are highlighted in **bold**, and second best are underlined.

| | Baseline | NLRDIC | Advections |
|---|---|---|---|
| Supervised | *MLP + basis* | **9.88e-4** | 6.90e-2 |
| Unsupervised | *PPINNs* | 3.71e-1 | 4.50e-1 |
| | *PO-DeepONet* | 4.36e-1 | 5.65e-1 |
| Hybrid | *PI-DeepONet* | 5.39e-2 | 4.26e-1 |
| | *PINO* | 3.79e-3 | **6.51e-4** |
| | *Ours* | 1.41e-3 | 5.39e-3 |

These additional datasets show cases where baselines are performing very well on the considered PDE. For *Advections*, PINO reached very good performances. We believe that the Fourier layers used in the model fit well to the phenomenon. Indeed, the solution is represented using a combination of sine moving with time. This simple dynamics is easily captured using Fourier transformations. Our B-Splines basis can be sub-optimal for this dataset. For the complexified version of *NLRD*, *NLRDIC*, it is the supervised baseline that performs best. This dataset does not present high frequencies, and the MLP looks sufficiently expressive to find the coefficient in the basis. However, this baseline had difficulties in reconstructing the higher frequencies in *Advections*. Even if our model does not

perform best on these datasets, it is ranked second. We believe that these additional results show the robustness of our method across different physical phenomena.

### E.5 TRAINING BEHAVIOR

We show in this section the evolution of the MSE as the training progresses (see fig. 15).

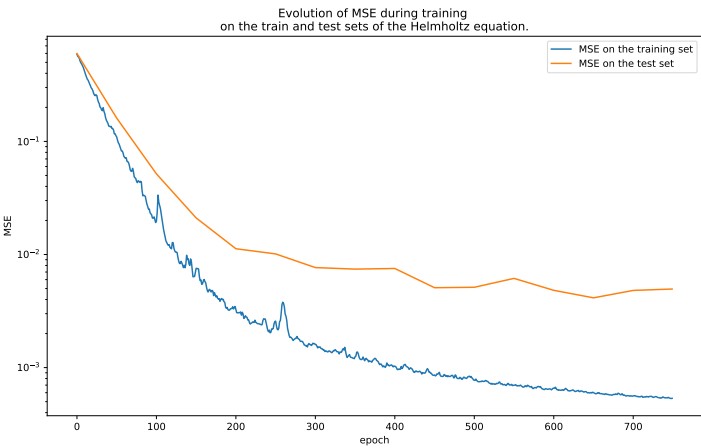

Figure 15: MSE during training on the training and testing sets. Example shown on the Helmholtz equation for results as presented in table 2.

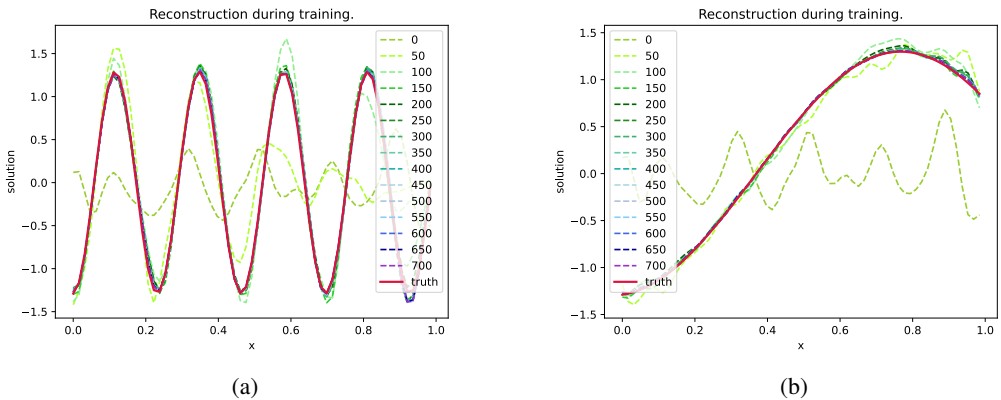

(a)  (b)

Figure 16: Reconstruction of the solution PDEs during training. Example drawn from the test set.

In fig. 15, we show the MSE evolution on both the train and test sets (evaluation every 50 epochs) and on figs. 16a and 16b some reconstruction (from the test set), with respect to the training epochs. The training set (800 PDE trajectories) corresponds to PDEs used to update the network parameters, while the test set (200 trajectories) are unseen PDEs for the network. **This means that the model has not been trained or optimized on these PDEs.** The test set is composed of PDEs with varying PDE parameter values ($\omega$) and boundary conditions ($u_0, v_0$). This illustrates that the generalization performance on **new** PDEs **within** the training distribution is rapidly achieved by the network.

### E.6 LOSS LANDSCAPES

In this section, we propose a visual representation of the optimization paths in the loss landscape. Figures 17 and 18 illustrates the behavior of the vanilla PINNs algorithm and of our "learning to optimize" method. The plots represent respectively a 2D visualization of the physical loss landscape $\mathcal{L}_{PDE}$ (fig. 17) and of the data loss landscape $\mathcal{L}_{\text{DATA}}$ (fig. 18), around the approximate solution (in our basis) of a given Helmholtz equation. The function basis used in this experiment has a size of 32. We used the technique in (Garipov et al., 2018) to visualize the solution coordinates in the basis $\Theta$ in the loss landscape (more details in appendix E.7.1). Superimposed on the loss background, we plot three trajectories obtained by starting from an initial random vector $\Theta_0$.

Figure 17 visualizes the sharp and ill conditioned landscape of the physical loss $\mathcal{L}_{PDE}$, while Figure 18 shows the better conditioned landscape of the MSE data loss $\mathcal{L}_{\text{DATA}}$. The figures provide intuition on how the proposed algorithm operates and improves the convergence. The optimizer $\mathcal{F}$ modifies the direction and magnitude of the stochastic gradient descent (SGD) updates in the physical loss landscape (Figure 17, left plot). It achieves this by utilizing solution values at various sample points to adjust the gradient, steering it toward the corresponding minimum in the mean squared error (MSE) landscape (Figure 18, central plot). While the descent remains within the residual physics error space, the learned optimizer provides an improved gradient direction, enhancing convergence efficiency (Figure 17, right plot). We describe below the different figures.

- Left Columns in Figures 17 and 18: these figures show the gradient path (100 steps) obtained by directly optimizing the physical loss $\mathcal{L}_{PDE}$, similar to the vanilla PINNs algorithm. This trajectory highlights the ill-conditioning of the optimization problem associated to $\mathcal{L}_{PDE}$.

- Center Column in Figures 17 and 18: these figures plot the trajectory of a gradient-based optimization algorithm trained with a mean squared error $\mathcal{L}_{\text{DATA}}$ data loss (100 steps), under the assumption that the solution values are known at collocation points. While these quantities are not available in our case, this visualization is included to illustrate the differences in convergence behavior between physics-based and MSE-based loss functions when both are accessible.

- Right Column in Figures 17 and 18: these plots illustrate the behavior of our algorithm (the solver is trained with 2 gradient steps for this example). It demonstrates the effect of our learned optimizer and the significant improvements achieved compared to a standard gradient descent algorithm on the physical loss.

. We describe in appendix E.7 the construction of the figures figs. 17 and 18.

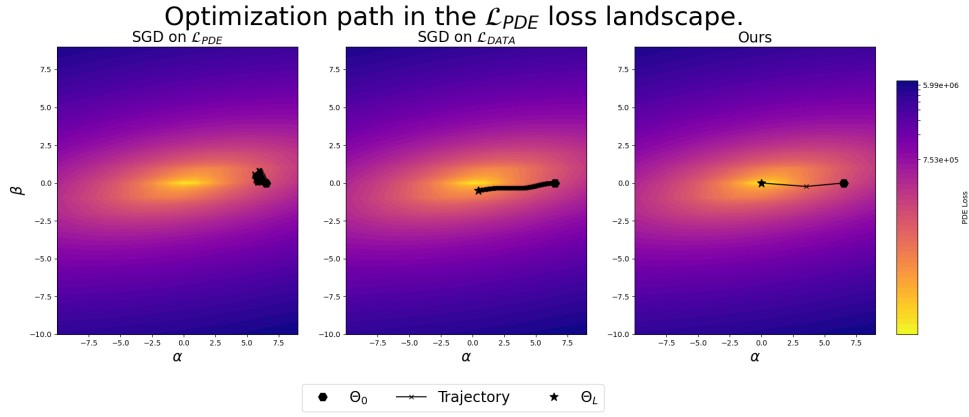

Figure 17: Loss landscapes and optimization trajectories on an instance of the Helmholtz PDE. The figure represents the PDE loss $\mathcal{L}_{\text{PDE}}$ landscape and superimposed are examples of optimization paths computed using SGD on the physical loss (left), the data loss (center), and our method (right).

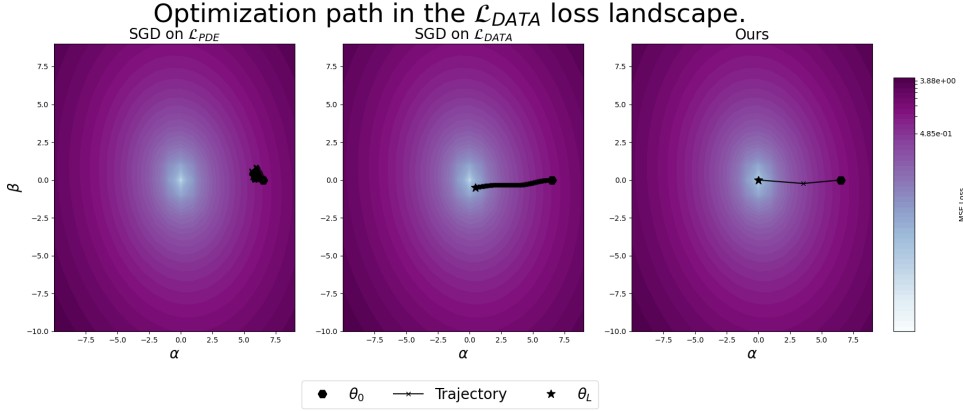

Figure 18: Loss landscapes and optimization trajectories on an instance of the Helmholtz PDE. The figures represents the DATA loss $\mathcal{L}_{\text{DATA}}$ and superimposed are examples of optimization paths computed using SGD on the physical loss (left), the data loss (center), and our method (right).

### E.7 PROJECTION OF THE BASIS FOR VISUALIZATION

Following the technique described in (Garipov et al., 2018), we plot a 2D slice of the loss function around the minimum solution. We explain below the method used for creating figs. 17 and 18.

#### E.7.1 CREATE A BASIS FOR PROJECTION

We want to visualize the loss landscapes around a parameter solution $\Theta^\star$ of a given PDE. Due to the high dimensionality of the parameter space, we use the method proposed in (Garipov et al., 2018), that involves creating a 2d slice of the loss landscape for visualization. This methods projects the multidimensional landscape onto a 2D basis $(u, v)$ so that any solution vector $\Theta$ could be written as $\Theta = \alpha u + \beta v$, with, $\alpha, \beta \in \mathbb{R}$ and $u, v \in \mathbb{R}^N$, 2 orthonormal vectors. Let $\Omega_{u,v} := \text{Vect}\{u, v\}$ denote this projection space. For the visualization, the basis must include the solution vectors found by our algorithm. For that:

1. We start by running our optimization algorithm from a random vector parameters $\Theta_0$ to find $\Theta_L \in \mathbb{R}^N$, our approximate PDE solution. We set $w_1 = \Theta_L$ and set a second vector $w_2 = \Theta_0$, chosen here as the initial vector of our optimization process.

2. We generate a third vector $w_3 \in \mathbb{R}^N$ randomly.

3. We then set: $u = w_2 - w_1$ and $v = (w_3 - w_1) - \frac{<w_3 - w_1, w_2 - w_1>}{||w_2 - w_1||^2}(w_2 - w_1)$

4. The resulting normalized vectors $\hat{u} = \frac{u}{||u||}$ and $\hat{v} = \frac{v}{||v||}$ form an orthogonal basis containing $w_1, w_2$ and $w_3$.

5. Using the solution parameters $\Theta_L$ as the origin for our newly created basis, we can compute a 2D slice of the loss landscape where each solution $\Theta$ could be represented in the $(u, v)$ basis with its projection coordinates $(\alpha, \beta)$ as $\Theta_{projection} = w_1 + \alpha\hat{u} + \beta\hat{v}$.

#### E.7.2 PLOT TRAJECTORIES

Finally, we run a baseline optimization procedures and get trajectories $\{\Theta_i\}_{i=0}^L$, with L being the number of optimization steps, e.g. a gradient descent, on the physical loss, the MSE loss or our proposed method. We project the $\Theta_i$'s in the created basis (see appendix E.7.1) and plot this projection of the optimization trajectory on the loss landscape $\Omega_{u,v}$ as visualized on figs. 17 and 18.

#### E.7.3 ILLUSTRATION OF THE ILL-CONDITIONING OF THE PDE LOSS

As an illustration of the ill-conditioning of the PDE loss, we replicate fig. 17 using a basis that further emphasizes this aspect. To build this basis, we use the procedure described in appendix E.7.1 by using 2 eigenvectors of the hessian of the PDE loss. First, we compute $\text{Hess}(\mathcal{L}_{\text{PDE}})$ and its

eigenvectors decomposition. We select the vectors associated to the highest and lowest eigenvalues, and respectively set them to $w_2$ and $w_3$. The resulting landscape visualization is shown in fig. 19.

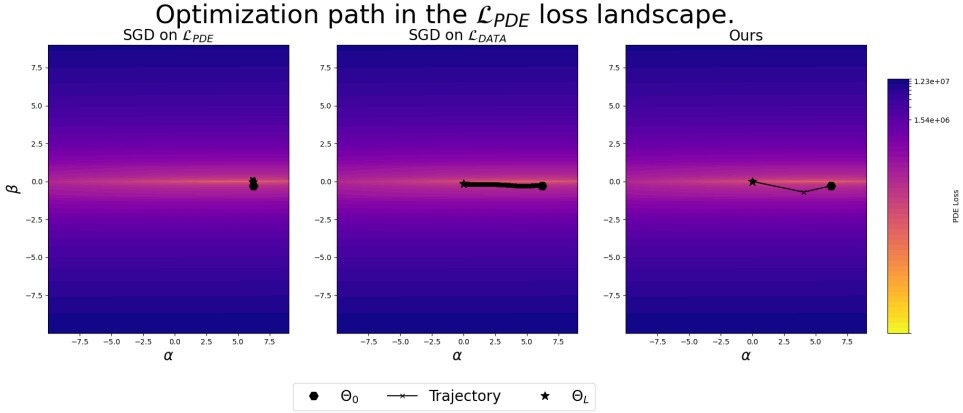

Figure 19: Loss landscapes and optimization trajectories on an instance of the Helmholtz PDE. These optimization trajectories are computed using SGD on the physical loss (left), the data loss (center), and our method (right). The background represents the PDE loss $\mathcal{L}_{\text{PDE}}$.

Figure 19, clearly illustrates the characteristic shape of the ill-conditioned function $\mathcal{L}_{\text{PDE}}$. The two directions, extracted from the highest and lowest eigenvalues, are clearly visible on this PDE loss landscape. This highlights the difficulty of this optimization problem for standard descent methods.

## F QUALITATIVE RESULTS

This section is dedicated to visualization of the results of our model, baselines and optimizers, presented in section 4. For each PDE considered, we chose 2 samples in the test sets and compute the solutions with our model and the different baselines. We provide visualization samples with $L = 5$ *i.e.* results proposed in tables 14 and 15 to detail more precisely the evolution of the solution at several steps of optimization. 3 datasets are shown with $L = 2$: *Heat* for computational reasons (training with $L = 5$ is much more expensive when the dimension of the problem increases) and the 2 additional datasets trained only with $L = 2$ (*Advections* and *NLRDIC*). Then, we show the evolution of the reconstruction of the solution with our method *i.e.* we plot the solution at each step of the optimization (figs. 20a, 20b, 23a, 23b, 26a, 26b, 29a, 29b, 32a, 32b, 35a, 35b, 38a and 38b) and we compare the final prediction with baselines' (figs. 21a, 21b, 24a, 24b, 27a, 27b, 30a, 30b, 33a, 33b, 36a, 36b, 39a and 39b). Finally, we chose 20 (6 for *Heat*) PDEs and we reproduce fig. 2 for every dataset. More precisely, we optimize one PINN per PDE using Adam, we fit our basis using several optimizers (GD, ADAM, L-BFGS and our learned optimization process) and we fine-tune the learned PINO for $10,000$ steps and visualize the evolution of the MSE (averaged at each step on the selected PDE). These figures show the relevance of learning the optimizer when using physics-informed losses (figs. 22, 25, 28, 31, 34, 37 and 40).

### F.1 HELMHOLTZ

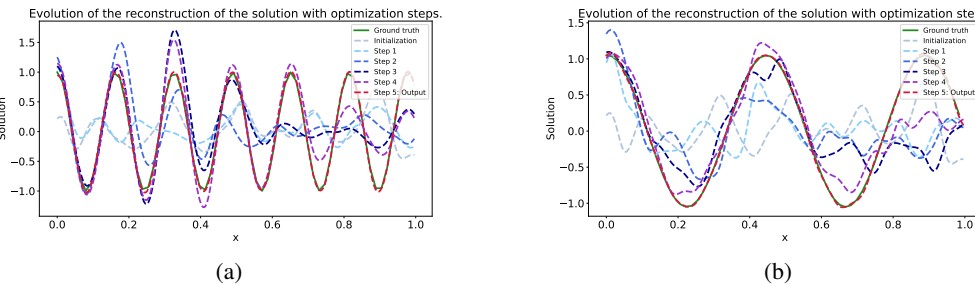

(a)                                                                      (b)

Figure 20: Reconstruction of the solution using our optimizer on the Helmholtz dataset.

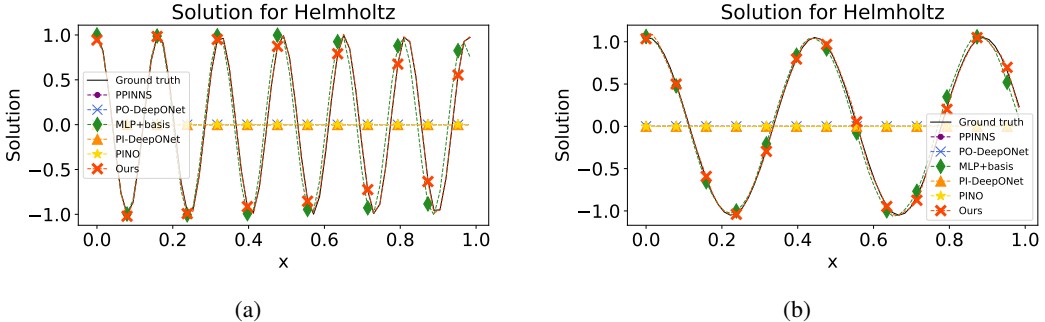

(a)                                                                      (b)

Figure 21: Visual comparison of the solutions for the Helmholtz equation.

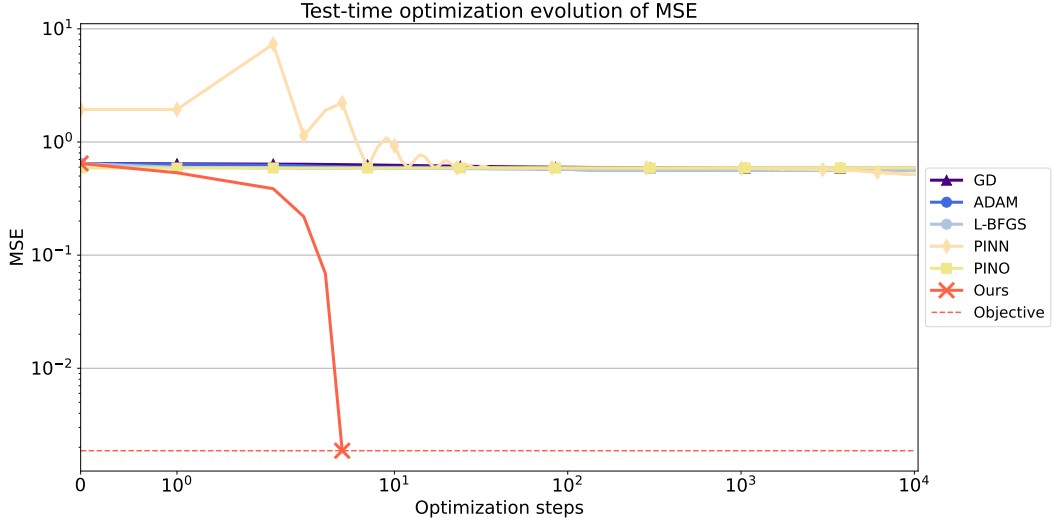

Figure 22: Test-time optimization based on the physical residual loss $\mathcal{L}_{\text{PDE}}$ on *Helmholtz*. Note that, even though hardly visible on this figure, the optimization is running very slowly and the PINN MSE (orange) decreases for the last steps. This dataset will probably need even more steps before convergence.

## F.2 POISSON

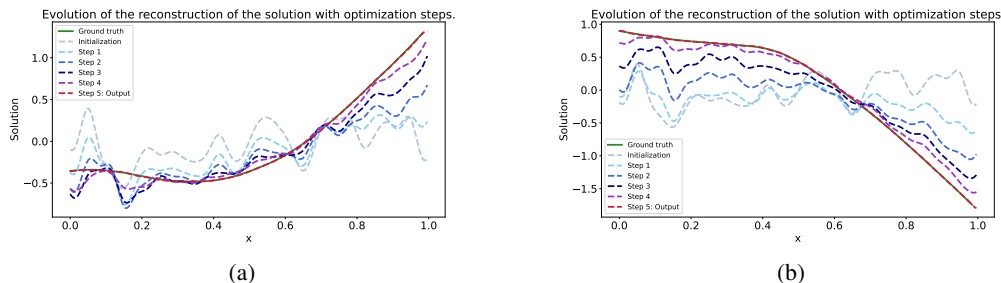

|     |     |
| --- | --- |
| (a) | (b) |

Figure 23: Reconstruction of the solution using our optimizer on the Poisson dataset.

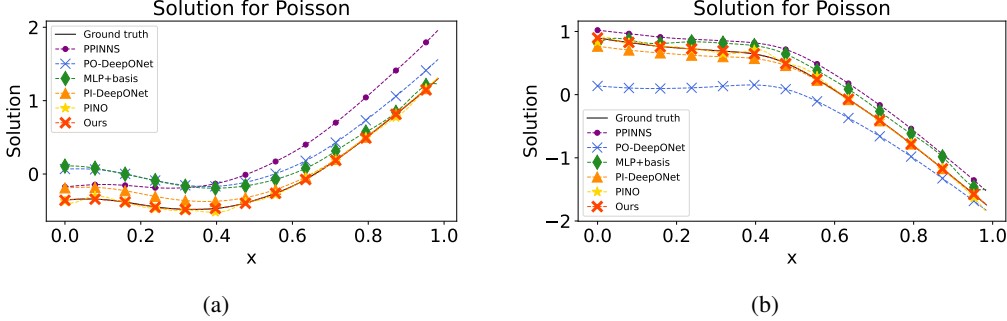

|     |     |
| --- | --- |
| (a) | (b) |

Figure 24: Visual comparison of the solutions for the Poisson equation.

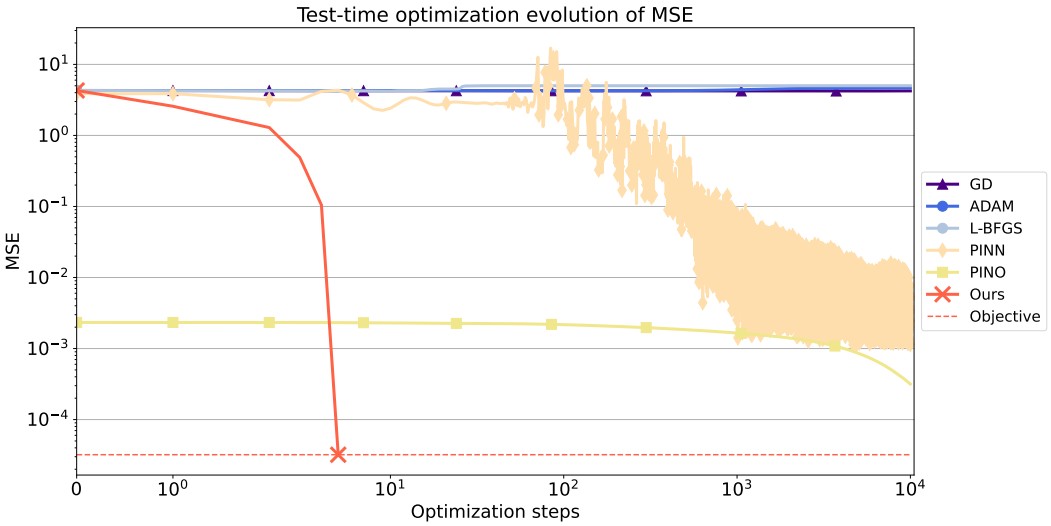

Figure 25: Test-time optimization based on the physical residual loss $\mathcal{L}_{\mathrm{PDE}}$ on *Poisson*.

## F.3 REACTION-DIFFUSION

Evolution of the reconstruction of the solution with optimization steps.

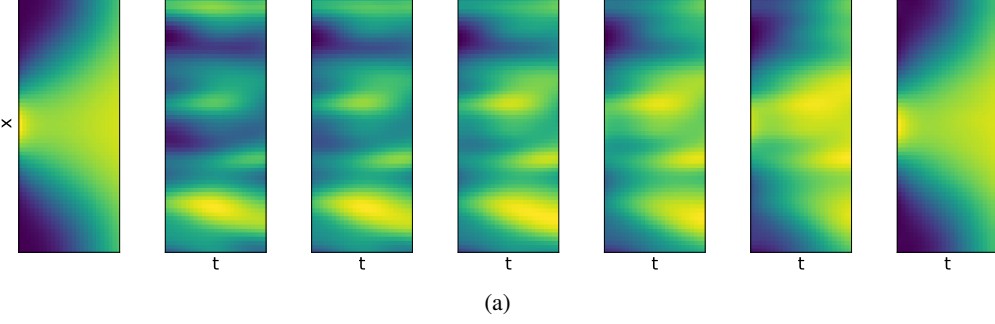

(a)

Evolution of the reconstruction of the solution with optimization steps.

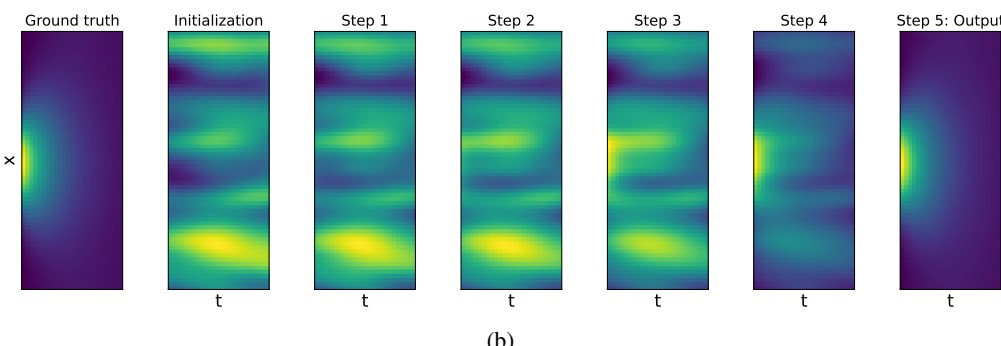

(b)

Figure 26: Reconstruction of the solution using our optimizer on the Reaction-Diffusion dataset.

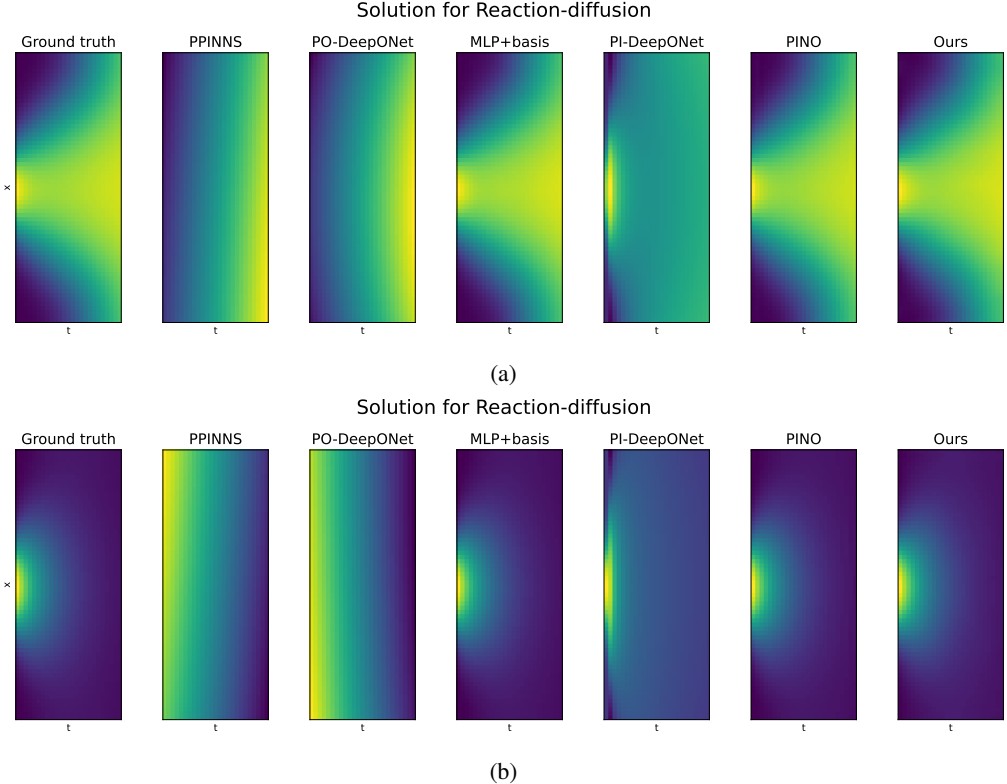

Figure 27: Visual comparison of the solutions for the Reaction-Diffusion equation.

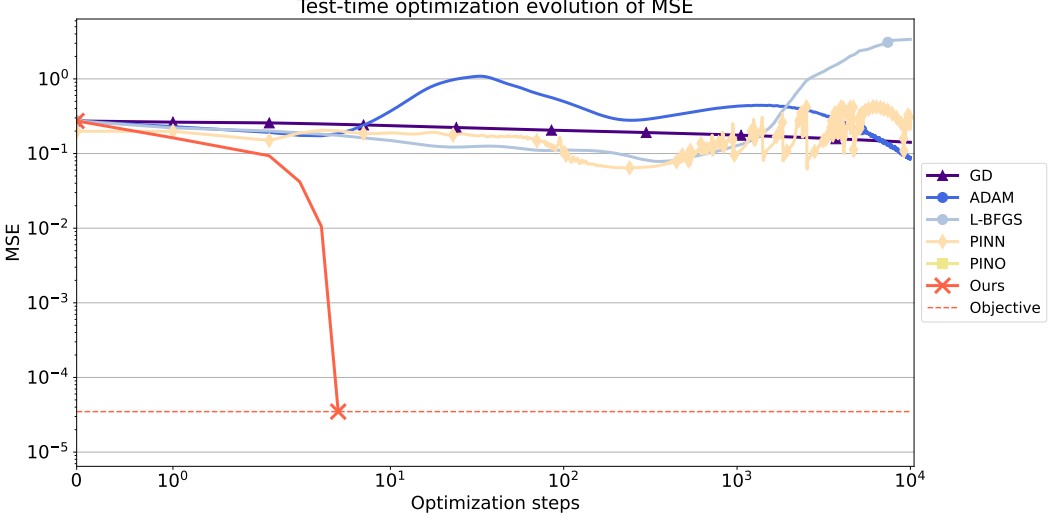

Figure 28: Test-time optimization based on the physical residual loss $\mathcal{L}_{\text{PDE}}$ on *NLRD*.

## F.4 DARCY

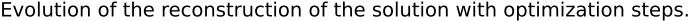

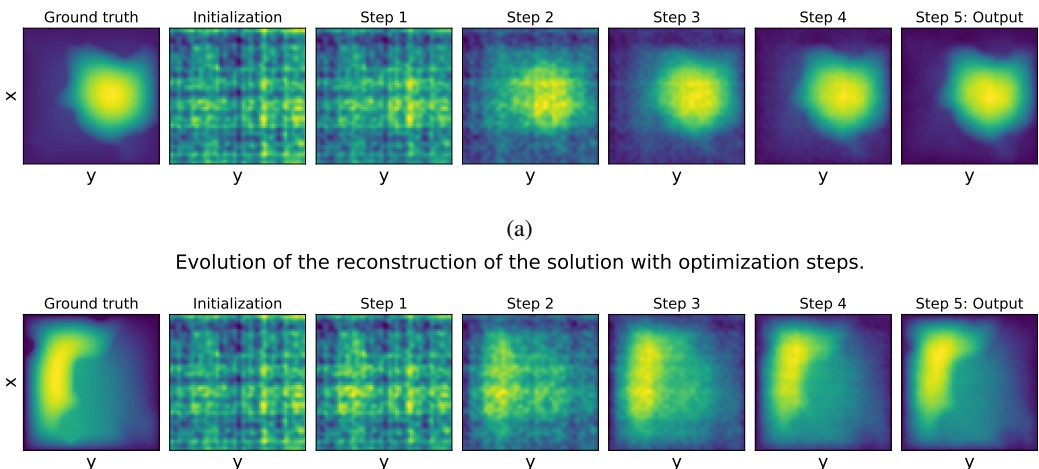

Figure 29: Reconstruction of the solution using our optimizer on the Darcy dataset.

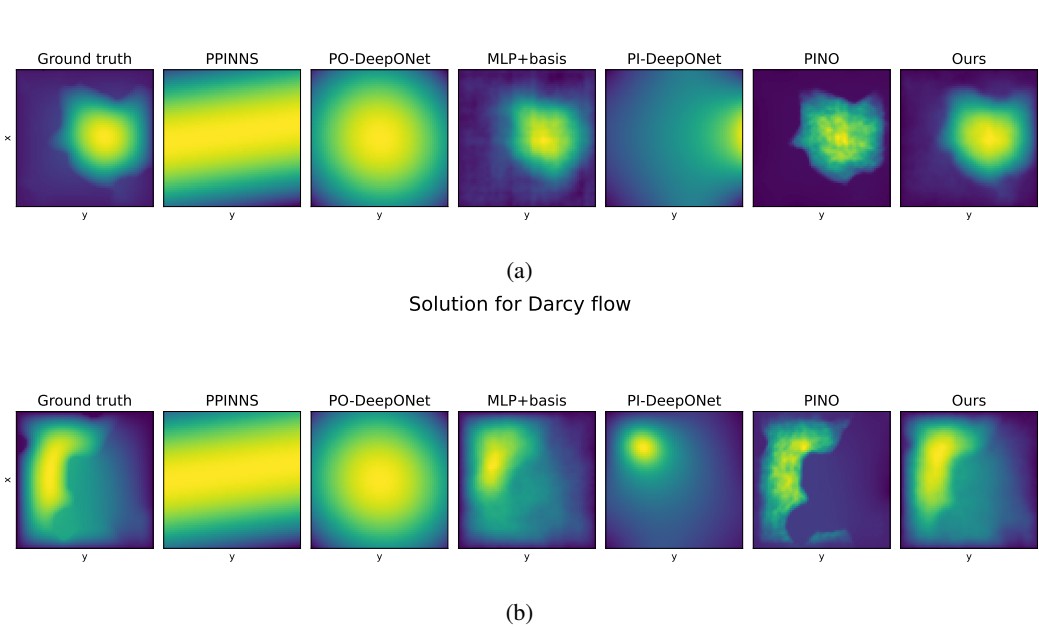

Figure 30: Visual comparison of the solutions for the Darcy equation.

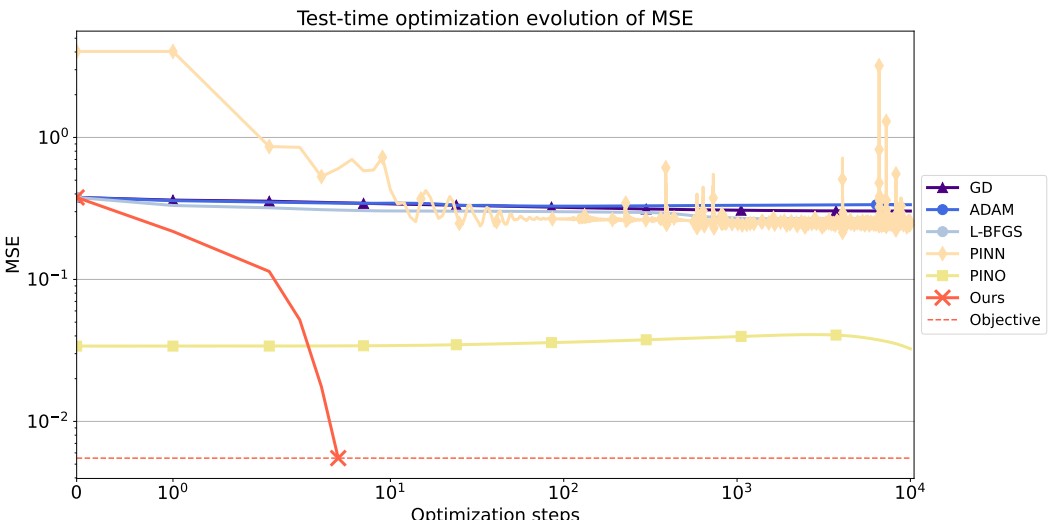

Figure 31: Test-time optimization based on the physical residual loss $\mathcal{L}_{\mathrm{PDE}}$ on *Darcy*.

## F.5 HEAT

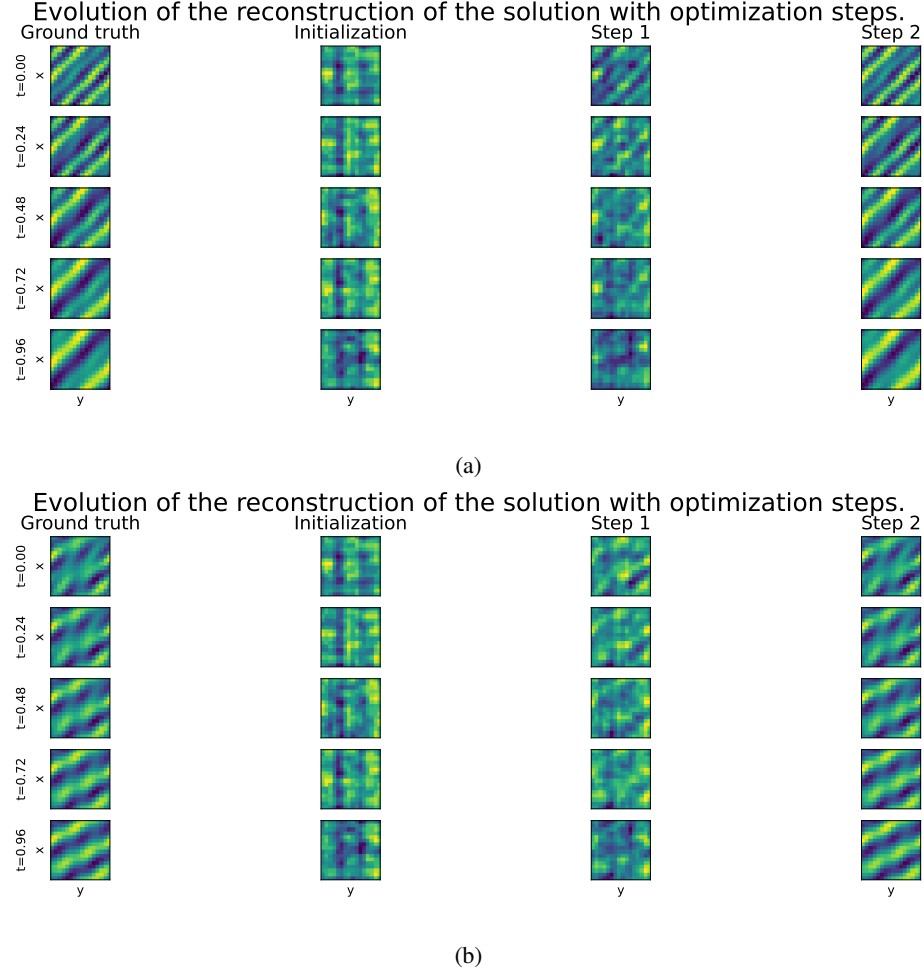

Figure 32: Reconstruction of the solution using our optimizer on the Heat dataset.

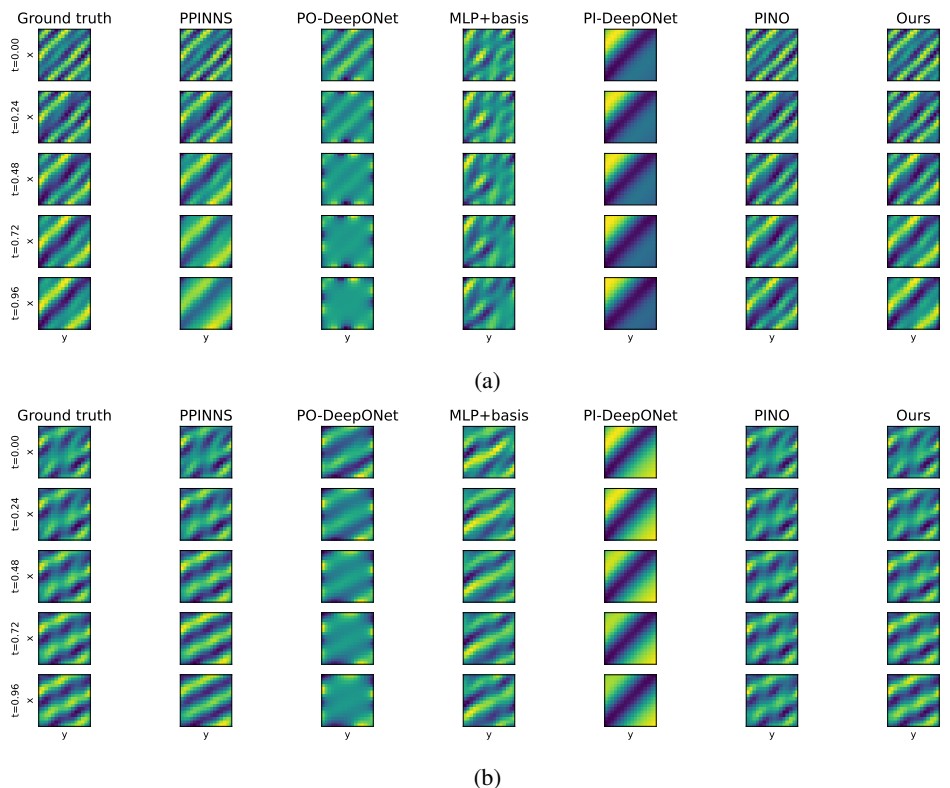

Figure 33: Visual comparison of the solutions for the Heat equation.

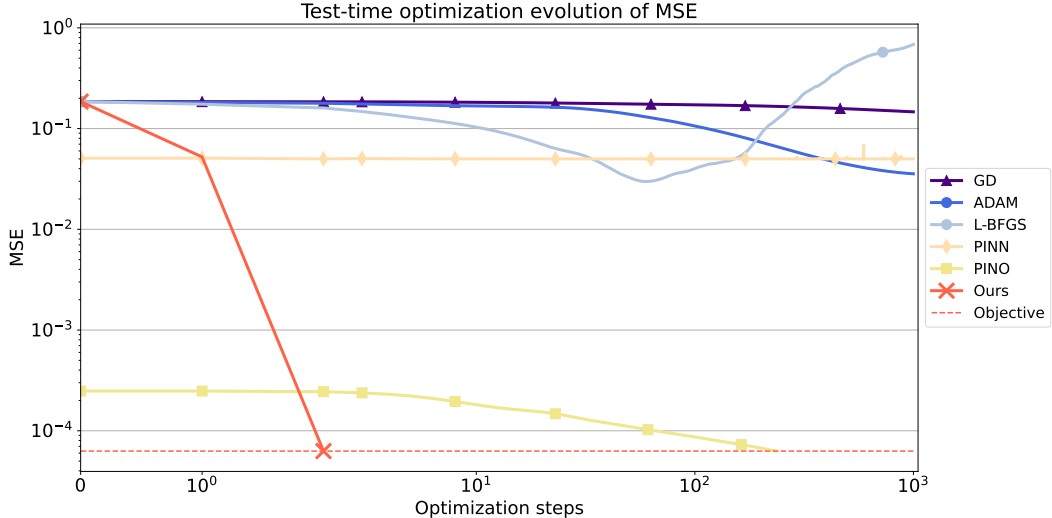

Figure 34: Test-time optimization based on the physical residual loss $\mathcal{L}_{\text{PDE}}$ on *Heat*. For computational reasons, this experiment has been conducted on $1,000$ steps only.

## F.6 ADVECTION

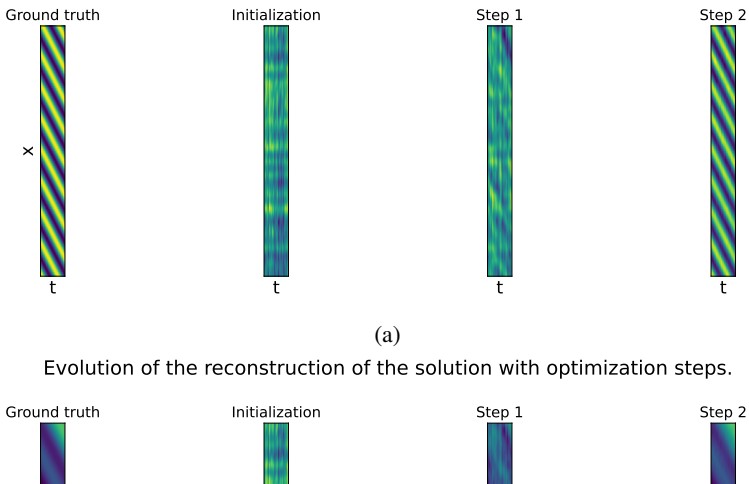

(a)

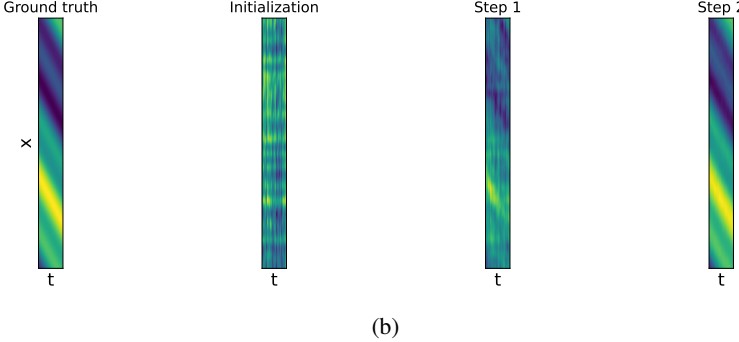

(b)

Figure 35: Reconstruction of the solution using our optimizer on the Advection dataset.

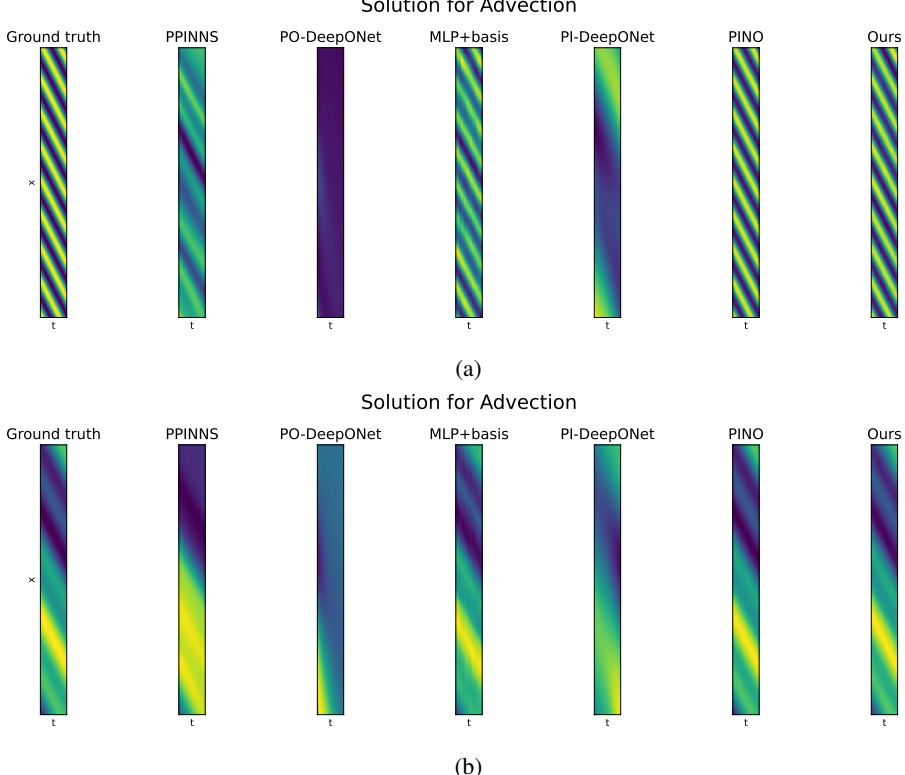

Figure 36: Visual comparison of the solutions for the Advection equation.

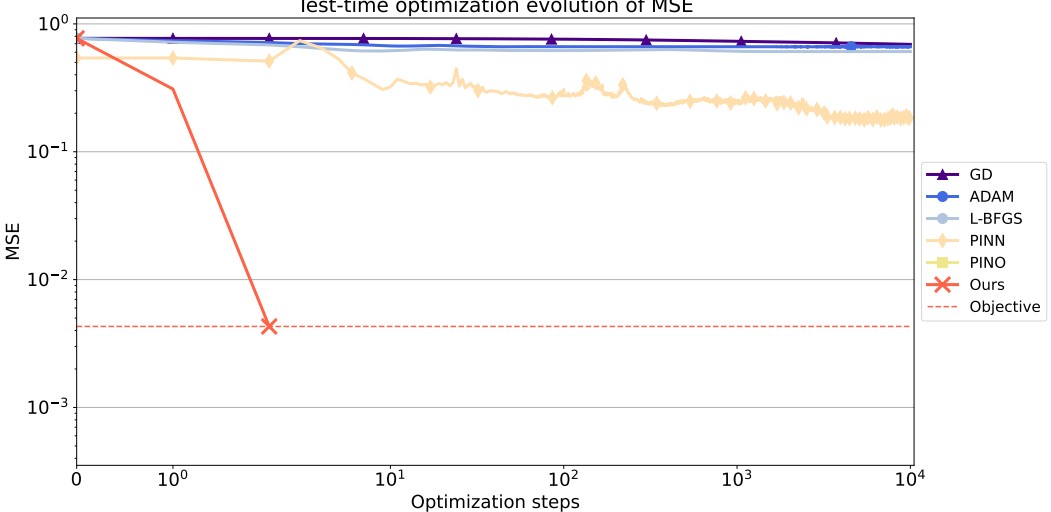

Figure 37: Test-time optimization based on the physical residual loss $\mathcal{L}_{\text{PDE}}$ on *Advection*.

## F.7 NON-LINEAR REACTION-DIFFUSION WITH INITIAL CONDITIONS

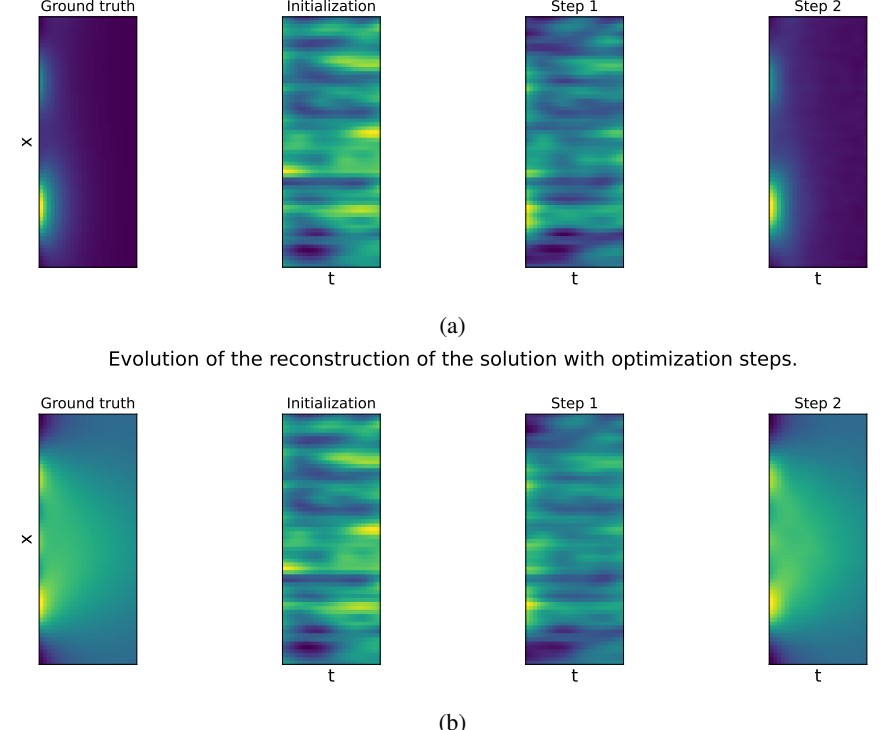

(a)

(b)

Figure 38: Reconstruction of the solution using our optimizer on the NLRDIC dataset.

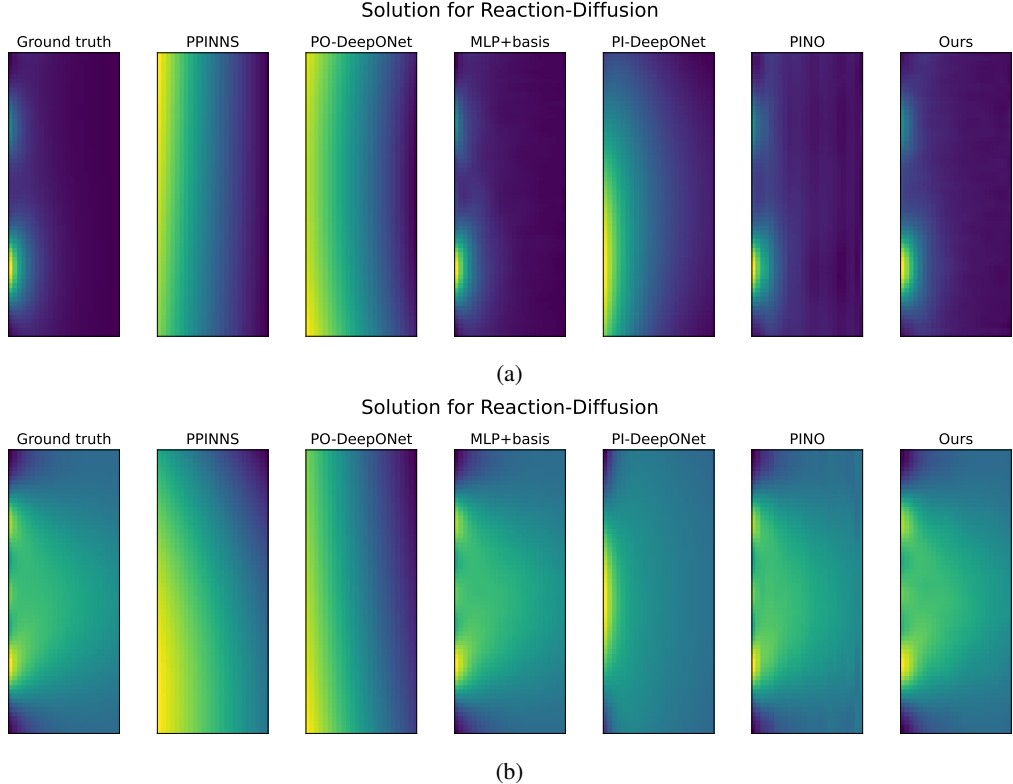

Figure 39: Visual comparison of the solutions for the NLRD with varying IC equation.

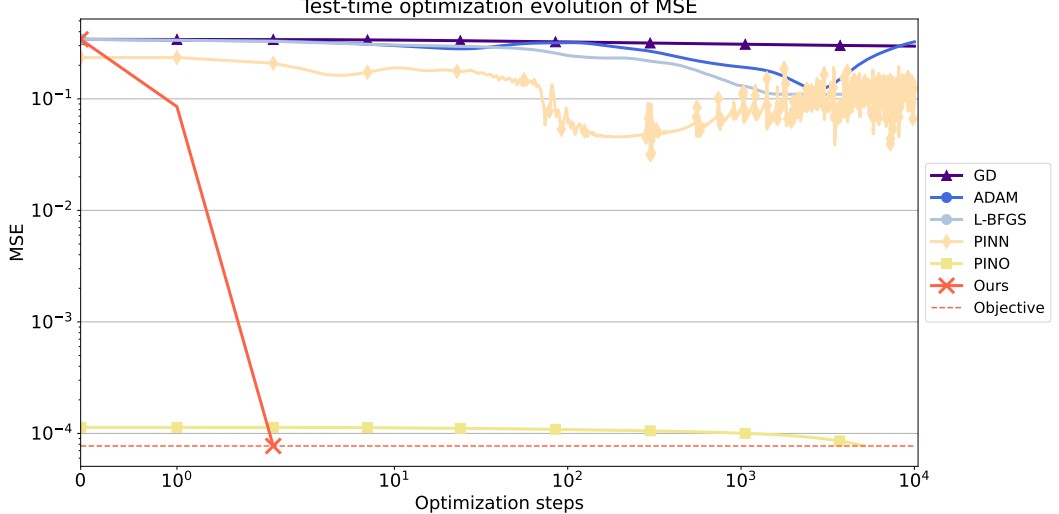

Figure 40: Test-time optimization based on the physical residual loss $\mathcal{L}_{\text{PDE}}$ on *NLRD* with varying IC.

