# OpenReview forum: "Learning a Neural Solver for Parametric PDEs to Enhance Physics-Informed Methods"
_ICLR.cc/2025/Conference — ICLR 2025 Poster_

### Official Review · Reviewer_TPuR · 2024-10-16

**Soundness:** 1
**Presentation:** 2
**Contribution:** 1
**Rating:** 3
**Confidence:** 5

**Summary:**

The paper propose a method that seems to be a PINO method to tackle learning of PDEs with the presents supervision from both data and equations.

**Strengths:**

The authors tackle and important problem in AI and science.

**Weaknesses:**

The writing and position of the paper is very limiting.

A few basic comments before we get to the main ones.
-1 In the intro:
Please note that when a PDE equation is given, that is complete supervision. Please avoid calling it unsupervised. PDE equation is absolute supervision, sufficient to solve the problem completely.
-2 In the intro:
Please pay more attention; there is a difference between neural networks and neural operators. One is for learning on the finite-dimensional setting; one is for functional data, that is your case. I only found out on page 8 that the authors talk about Fourier layers, and then I had to go through the referenced paper to realize that the authors use a neural operator for their solver. Even there, please explain you are doing operator learning and using neural operators. Please also define your neural operator architecture. FNO is not even defined or abbreviated in your work. Please bring the abbreviation and explain in your paper that your work is neural operator-based.

-3 In 3.3
The authors need to define varrho in the equations in this section. I guessed what that was after reading page 8. There is no varrho in the equations in 3.3.

-4 In 3.3
Why choose linear span? We know such a thing is fine for easy problems and not sufficient for complex problems. That's why even the first paper proposing the PINO paradigm also deals with complex problems and doesn't use linear span simplification.

-5 In 3.3
 What does Solve even mean in alg2? I only guessed its meaning.

-6 In 3.3. Please be mindful of term solve and term prediction for F. The F predicts something and doesn't solve anything in the sense that solvers in the field of applied math solve.

-7 In 3.3
Reading line 304, it feels that the authors first do PINO with only physics and then PINO with data. However, the intro implied that it is the other way. Which is done here?

-8 In 3.3 what is trajectory in line 308?


The main comment is the presentation. Reading the paper, it seems the paper is aiming to advance PINO. However, reading abstract, intro, and other sections up to page 8, the paper does not even talk about PINO. Everything the paper promotes and sugest they are doing is already propsed and establisehd in PINO paper. Please read the PINO paper carefully and possition your paper according to that. Also, in both abstract and intro, mantion what is new in your work. I could not find anuthing different than prior methods. And I am sure there are many things new and novel in this paper, but I just couldn't parse it when abstract and intro are all talking about basics of PINO.

**Questions:**

mentioned above.

---

> ### Author Response · Authors · 2024-11-25
>
> ## W1 Unsupervised
> There might be some misunderstanding on this point. What is meant by unsupervised in the context of neural PDE solvers is that **no labeled data is required (except for some I/B conditions)**, i.e. the neural network is optimized **through the PDE loss only**, which is the essence of PINNs. This terminology is commonly used by many authors. We provide below references including some key authors on this topic together with an indication on where to find this wording in the manuscript.
> * Sebastian Schaffer, Thomas Schrefl, Harald Oezelt, Alexander Kovacs, Leoni Breth, Norbert J. Mauser, Dieter Suess, Lukas Exl, Physics-informed machine learning and stray field computation with application to micromagnetic energy minimization, 2023: _"PINNs are inherently mesh-free and unsupervised learning models."_ in the abstract.
> * Wuyang Chen, Jialin Song, Pu Ren, Shashank Subramanian, Dmitriy Morozov, Michael Mahoney, Data-Efficient Operator Learning via Unsupervised Pretraining and In-Context Learning, NeurIPS 2024 _"Unsupervised Pretraining: no simulated solution" (Figure 1) ... "our unlabeled PDE data include physical parameters(aij ,bi,c), forcing functions (f), and coordinates (grids of the discrete physical space)."_,  Section 3.1
> * Yuyao Chen, Lu Lu, George Em Karniadakis, and Luca Dal Negro, Physics-informed neural networks for inverse problems in nano-optics and metamaterials _"Moreover, PINNs do not require any data on the inverse parameters it predicts, and thus it belongs to unsupervised learning."_ (section 2)
> * Cuomo, S., Di Cola, V.S., Giampaolo, F. et al. Scientific Machine Learning Through Physics–Informed Neural Networks: Where we are and What’s Next. J Sci Comput 92, 88 (2022). https://doi.org/10.1007/s10915-022-01939-z _"The basic concept behind PINN training is that it can be thought of as an unsupervised strategy that does not require labelled data"_ (section 1)
> * Edgar Torres, Mathias Niepert, Survey: Adaptive Physics-informed Neural Networks, 2024, _"Physics-informed neural networks (PINNs) present a promising solution for solving partial differential equations (PDEs) using neural networks, especially indata-scarce scenarios due to their unsupervised learning abilities"_, (abstract).
> * Klapa Antonion, Xiao Wang, Maziar Raissi, Laurn Joshie,, Machine Learning Through Physics–Informed Neural Networks: Progress and Challenges, _"The training process of PINNs functions asan unsupervised strategy, eschewing the need for labeled data stemming from prior simulations or experiments."_ (page 47)
>
> ## W2 Neural Operator
> - The distinction between operators and mappings between finite-dimensional spaces is not relevant in our context. Here, the FNO is used to learn a mapping between finite-dimensional spaces and not for its operator propperties. Said otherwise, any other network architecture could be employed as well, whether inspired by a neural operator or not (see ablation in Appendix E1, Table 12 of the updated manuscript). FNO was chosen for its robustness across different PDEs, as demonstrated in the experiments.
> - Thank you for pointing this out; we have added the definition of the FNO abbreviation (see the updated version of the paper, line 420).
>
> ## W3 varrho
> This was already addressed in the paper in section 3.2 on line 234 _"we first transform the gradient using a neural network_  $F_{\varrho}$ _with parameters_ $\varrho$".
>
> ## W4 Linear span
> The proposed method leverages a linear combination of non linear functions. Its approximation properties are similar to the ones of NNs (the same goes for Fourier basis or different polynomial bases for example used for many surrogate models for PDEs). Note that the neural operator "DeepONet" ([1]) makes use of a similar linear expansion of basis function. This is a choice of implementation and not a simplification. Besides, as detailed in footnote 4 of the paper, our preliminary experiments have shown that using a more complex combination of the basis functions, e.g. through a neural network did not improve the performance and simply made the training more complex. This last point is illustrated in an ablation in Appendix E1, table 11, where we compare a linear and a non linear combination of the basis functions.
>
> ## W5 Solve
> Following your suggestion, we replaced "solve" in algorithm 2 by "Estimate" in order to avoid any confusion. Please see the updated version of the manuscript.
>
> ## W6 Solve and predict
> We agree, F itself is not used as a solver, instead it is used to update the gradient direction within the solver. However, this shorthand  greatly simplifies the description. We will make this clear in the final version (footnote 4 of the updated version). Thanks for the suggestion.

---

> ### Author Response · Authors · 2024-11-25
>
> ## W7 PINO
> We apologize, but there seems to be a significant misunderstanding here. We regret if this was not made clear enough in the paper and will clarify the distinction between the objectives and components of the two frameworks. Let us attempt to summarize this distinction below. We hope this will help clear up the confusion.
>
> The PINO paper proposes a method to use neural operators by leveraging both physics-based and data-driven losses. At inference time, the model is fine-tuned using only the physical loss (with an optimizer, such as Adam). In this approach, data are used when available to learn the solution operator, complementing the information provided by the PDE loss.
>
> Our objective is fundamentally different. In our setting, the neural network (NN) is not used to train an ansatz, as in PINO. Instead, it is designed to optimize the convergence of the gradient-based optimizer itself. Specifically, the NN is employed at test time to accelerate the optimization procedure used to solve the PDE. The role of the data is to train this optimizer, guiding the solver rather than the ansatz operator.
>
> Once the solver optimizer is trained, it can be applied directly to new instances of the parametric equation without requiring additional retraining or fine-tuning, as is necessary in PINO (referred to as test-time optimization in the PINO paper). In other words, our work addresses a “learning to solve” problem. This distinction is explicitly stated in lines 80–82 of our manuscript:
> _"With respect to unsupervised approaches, our model implements an improved optimization algorithm, tailored to the parametric PDE problem at hand instead of using a hand-defined method such as stochastic gradient descent (SGD) or Adam."_
>
> ## W8 Trajectory
> Trajectory is a spatiotemporal sequence in the observation space of the phenomenon, corresponding to a simulation of a PDE.
>
> ## Contribution
> Again, we apologize if we were not clear enough. We hope that we have highlighted the significant differences between our framework and objectives and those of PINO above, and that this clarification resolves the misunderstanding. Please let us know if you have any remaining questions, which we will be happy to address.
>
>
> ## References
> [1] Lu Lu, Pengzhan Jin, Guofei Pang, Zhongqiang Zhang, and George Em Karniadakis. Learning
> nonlinear operators via deeponet based on the universal approximation theorem of operators.

---

> > ### Comment · Reviewer_TPuR · 2024-11-28
> > **I appreciate the response.**
> >
> > Thank you for the response. W7 was clarifying things.
> >
> > W1. Thanks for the reply. It is an incorrect terminology. I will leave it to the AC. However, authors are encouraged to consult with ML experts with respect to ML terminologies. The problem, but definition, comes with absolute supervision.
> >
> > W2. I got it after reading w7. Thank you.
> >
> > W3. Thanks
> >
> > W4. Next time, please use advanced methods, maybe in consultation with ML experts. Using bases learning is very outdated and hasn't seen great results for decades. Why use that? I see the Deeponet you mentioned used the same.
> >
> > W5. Thank you.
> >
> > W6. Thanks.
> >
> > W7. Thanks, that makes sense.
> > Why are the results of using FNO for the PINO setting different from the results for Darcy reported in the PINO paper? It seems one order is off.
> >
> > Why not use advanced neural operator architectures or advanced datasets? Like those in PINO paper?

---

> > > ### Author Response · Authors · 2024-12-01
> > >
> > > Thanks a lot for your careful consideration of our responses.
> > >
> > > ## W1.
> > >
> > > We are ML specialists and we used this denomination purposefully because of its general adoption in the context of this topic. We however understand that it could introduce some confusion and we will take care to clearly explain its context in the manuscript in order to avoid any misunderstanding. Thank you for pointing this out
> > >
> > > ## W4.
> > >
> > > We respectfully disagree (at least partially) with you on this point. There are examples of recent and effective ML models that rely on basis function expansion without requiring complex or large NN architecture. This is the case for example with Fourier Networks which leverage linear expansions in Fourier spaces ([1] theorem 1 demonstrates that this simple expansion performs as well as more complex implicit neural networks architectures).
> > >
> > > Note that DeepONet [2], which also relies on linear expansions of basis functions, is a seminal work in the field of neural operators, along with the Fourier Neural Operator used for PINO. It has been used for a large number of applications, see e.g. [3, 4, 5] for recent references.
> > >
> > > However, we acknowledge that our main objective was to evaluate the relevance of our learned optimizer, rather than to develop a model achieving state-of-the-art performance for this set of tasks. We found it easier and more interpretable to work with basis functions rather than more sophisticated models. We agree that experimenting with different families of models, as you suggest, would be interesting. This was mentioned in the perspectives.
> > >
> > > We would also like to highlight that our method already shows interesting performances and  reaches **similar or better performances than baselines** that make use of non lienar combinations of functions.
> > >
> > > ## W7.
> > >
> > > The two settings are very similar but for our experiments, we used 3 layers for consistency across the datasets instead of 4 in the PINO paper.
> > > We ran the PINO baseline, with 4 layers and reached performance (please see below) similar to those presented in the PINO paper, thus validating our results.
> > >
> > > Table: MSE Relative when training PINO with different layer sizes.
> > > | Layers      | Relative MSE |
> > > |-------------|--------------|
> > > | 3 layers    | 1.01e-1      |
> > > | 4 layers    | 6.81e-2      |
> > >
> > >
> > > Please note that, even with this additional layer, the performances of the PINO model are below those of our method.
> > >
> > > Regarding the datasets, we understand the reviewer's point. However, we would like to emphasize that we have already used a variety of representative equations, including problems that are challenging for PINNs—for example, e.g., NLRD [6, 7], FNO [8] (Darcy), and PDEbench [9] (advection). Our benchmarks include non-linear datasets (NLRD), 2D as well as 2D+t datasets (Darcy and Wave), and high-frequency problems (with pulsation reaching up to $50$ in the Helmholtz equation), which already represent significant challenges.
> > > These datasets are widely considered advanced in the context of physics-based methods due to the ill-conditioning problems they address.
> > >
> > > ## References
> > >
> > > [1] Fathony, R., Sahu, A. K., Willmott, D., \& Kolter, J. Z. (2021). Multiplicative Filter Networks. ICLR, 1–10.
> > >
> > > [2] Lu Lu, Pengzhan Jin, Guofei Pang, Zhongqiang Zhang, and George Em Karniadakis. Learning nonlinear operators via deeponet based on the universal approximation theorem of operators.
> > >
> > > [3] Ehsan Haghighat, Umair bin Waheed, George Karniadakis, En-DeepONet: An enrichment approach for enhancing the expressivity of neural operators with applications to seismology, 2024
> > >
> > > [4]P an Huang, Yifei Leng, Cheng Lian, Honglai Liu, Porous-DeepONet: Learning the Solution Operators of Parametric Reactive Transport Equations in Porous Media, 2024.
> > >
> > > [5] Jinho Jeong, Eunji Kwak, Jun-hyeong Kim, Ki-Yong O, Prediction of thermal runaway for a lithium-ion battery through multiphysics-informed DeepONet with virtual data, 2024.
> > >
> > > [6] Aditi S. Krishnapriyan, Amir Gholami, Shandian Zhe, Robert M. Kirby, and Michael W. Mahoney.
> > > Characterizing possible failure modes in physics-informed neural networks. 2021.
> > >
> > > [7] Maryam Toloubidokhti, Yubo Ye, Ryan Missel, Xiajun Jiang, Nilesh Kumar, Ruby Shrestha, and
> > > Linwei Wang. DATS: Difficulty-aware task sampler for meta-learning physics-informed neural
> > > networks, 2024.
> > >
> > > [8] Zongyi Li, Nikola Kovachki, Kamyar Azizzadenesheli, Burigede Liu, Kaushik Bhattacharya, Andrew
> > > Stuart, and Anima Anandkumar. Fourier neural operator for parametric partial differential equations,
> > > 2020.
> > >
> > > [9] Makoto Takamoto, Timothy Praditia, Raphael Leiteritz, Dan MacKinlay, Francesco Alesiani, Dirk
> > > Pflüger, and Mathias Niepert. Pdebench: An extensive benchmark for scientific machine learning,
> > > 2023

---

> ### Author Response · Authors · 2024-12-02
>
> As the rebuttal deadline approaches, please let us know if we have adequately addressed your concerns. If you have any remaining questions or comments about our submission, we would be happy to address them.

---

### Official Review · Reviewer_2vqB · 2024-10-30

**Soundness:** 3
**Presentation:** 3
**Contribution:** 3
**Rating:** 8
**Confidence:** 3

**Summary:**

This paper addresses the problem of optimising Physics informed deep neural networks for solving PDE. Specifically, the paper addresses the issue of ill-conditioned optimization problems.

Using a Fourier representation, the authors clearly lay out the problem with ill conditioning of general PINNs and give a numerical example to illustrate this.

The paper describes how the problem can be transformed using a neural solver into one that can be optimised using gradient descent more quickly than the original problem.

**Strengths:**

Overall the paper does a good job of explaining the problem and describing the proposed solution. However there are a number of issues that need to be expanded upon in order to improve the thoroughness and reliability of the findings.

In general I think the paper represents a useful contribution to the literature and I am leaning towards an acceptance. However, I do think that it could be significantly improved if some of the findings detailed in the appendices could be brought into the main body of the paper as described in this review. Although the paper is fully 10 pages long, some of the descriptive text, especially in the early part of the paper could be made more concise to make space for these valuable aspects of the findings to be included in the main body of the paper.

**Weaknesses:**

The parameters $\Theta$ are updated using gradient descent. This is nested within a additional optimisation process for tuning the parameters $\varrho$. Some results should be included in the main body of the paper which show the CPU time, including for the optimisation of $\mathcal{F}_\varrho$. There are some results presented in Appendix F but these should be discussed in the main paper as they are important for evaluating the usefulness of the method presented.

The paper describes the problem in terms of the condition number $\kappa(A)$. A sketch proof is included in Appendix B to show that convergence is guaranteed and its speed is enhanced using this structure. However the wording around this sketch proof is quite confusing. Section B.2 there states "The introduction of $P$ as a pre-conditioner \textit{often} results in $\kappa(PA)<\kappa(A)$." and "the condition number is improved under \textit{some} optimality conditions. However in Sections B.3 and B.4 convergence and optimality is described as \textit{optimal} and \textit{guaranteed}. This inconsistency should be addressed due to the importance of this in supporting the empirical findings within the main part of the paper.

The results in the paper show very good improvements in the optimisation time and the MSE for the example PDEs used. However a range of parameters for each PDE is quoted in table 1. In table 2 the MSE results for the different equations are shown, however there is no indication of how these errors vary within the parameter ranges. There are some results presented  for this in Appendix E, however these should be brought into the main part of the paper as it is important to show the robustness of the method.

In general the paper is well written and formatted but  In line 436 and 437, the sentence reads ``Therefore our method should be primarily compared toe these baselines."

**Questions:**

My suggestions are included in the "Weaknesses" section and I would be very pleased to hear the authors' response to these suggestions.

---

> ### Author Response · Authors · 2024-11-25
>
> We appreciate your supporting positive review and thank you for all the comments and suggestions. We answer the concerns and questions raised in the following.
>
>
> ## W1 Training time
> We fully agree with the reviewer that training and inference times are important factors in assessing the relevance of this method, as well as for Physics-Informed methods in general. As you mentioned, we have provided a detailed computational time comparison for our method and the baselines in Appendix E3 for certain datasets, along with an analysis comparing our method to vanilla PINNs. While the table is too large to include in the main text, we have, following your suggestion, updated the main text to highlight the key insights from the table (section 4.4, updated version of the paper).
>
> ## W2 Proof
> Thanks for the comment, we agree that the initial version of the proof was somewhat confusing.
>
> In the updated version,  we have thoroughly revised this proof to better highlight the main results, their logical progression, and the connections between them (appendix B). We think that this solves the inconsistency issue mentioned in your review. Please tell us if there are still questions remaining.
> In particular, we have shown that the neural net optimizer leads to a condition number $\kappa(PA)$ equal to 1, which is optimal. If we consider the Poisson example, the condition number $\kappa(A)$ of the original PINNs problem being $K^4$, for $K>1$, the new condition number is greatly reduced. Note that this also directly implies that the number of steps $N^{\prime}(\epsilon)$ of our method is \textbf{significantly} reduced wrt to PINNs, i.e. $N^{\prime}(\epsilon)\ll N(\epsilon)$.
>
> ## W3 Parameter ranges
> The results presented in table 2 are averaged on several PDEs, i.e. with different parameters sampled from the ranges shown in table 1. We solved $200$ instances of PDE  with varying parameters with our method, and averaged the final MSE over the solution. These PDE instances are **unseen** PDE during training (in terms of PDE parameters, forcing terms and/or boundary/initial conditions). Following your suggestion, we have added in appendix E1 on figure 14, an evaluation of the performance evolution of the different methods, with respect to the PDE parameters on _Helmholtz_ equation. The results show that our method has consistent performance across the parameter distribution and is then quite robust to this variation when the baselines could only solve the PDE for only a narrow range of parameters.
>
>
> If there are any remaining issues or specific questions, we’re ready to address them during the discussion period. Please let us know if there’s anything else you’d like us to clarify to convince you about our submission.

---

> > ### Author Response · Authors · 2024-12-02
> >
> > As the rebuttal deadline approaches, please let us know if we have addressed your concerns.
> > If you have any remaining questions or comments about our submission, we would be happy to address them.

---

### Official Review · Reviewer_g6hN · 2024-10-30

**Soundness:** 4
**Presentation:** 4
**Contribution:** 3
**Rating:** 8
**Confidence:** 3

**Summary:**

This paper proposes an optimization process that can learn a set of parametric PDEs quickly and precisely while addressing the ill-conditioning issues that traditional PINNs face. The authors first train a physics-informed neural network on a dataset containing different PDE coefficients, initial conditions, and boundary conditions, enabling the network to predict PDE solutions in the initial step. Then, a solution approximation expressed as a linear expansion in a pre-defined basis is optimized to adapt to each PDE instance. This method enables fast learning of different PDE instances and effectively overcomes ill-conditioning issues.

**Strengths:**

A very good paper demonstrating a novel approach to addressing PINN’s limitations, with extensive experimental results to support the claims.

The paper presents strong motivation and a thorough review of related work.

The methods and the overall paper are easy to follow.

**Weaknesses:**

I have some questions about the training of the neural solver.

**Questions:**

My understanding is that the objective function for training the neural solver is data loss + PDE loss, which represents a soft-constrained optimization problem. Have you considered using a hard-constrained method, as in Lu et al. [1]?

[1] Lu, Lu, et al. "Physics-informed neural networks with hard constraints for inverse design." SIAM Journal on Scientific Computing 43.6 (2021): B1105-B1132.

---

> ### Author Response · Authors · 2024-11-25
>
> We appreciate the reviewer's feedback and support for our work, and we address their question below.
>
> ## Q1
> First of all, we would like to emphasize that in our method, there are two nested optimization processes: we train an optimizer (the neural solver) to accelerate the convergence of a baseline gradient (here SGD) in order to accelerate PINNs training. In this nested process, data are used for training the neural solver in the outer loop while the PDE loss is optimized in an inner loop. The use of data is then different from other algorithms like for example PINO or Physics-informed DeepONet (PI-DeepONet) that use them for solving a soft constrained problem between data and physics loss. Within the inner-loop, we did not use hard-constraints. This is a nice suggestion. We could not do the experiments during the rebuttal since this involves retraining many models but but this is an interesting direction for future work! Thank you for this suggestion.

---

> > ### Comment · Reviewer_g6hN · 2024-11-26
> >
> > I see. This could be a good direction. Thank you.

---

### Official Review · Reviewer_N1S1 · 2024-11-02

**Soundness:** 2
**Presentation:** 2
**Contribution:** 2
**Rating:** 6
**Confidence:** 4

**Summary:**

Physics-informed Neural Networks (PINNs) are popular neural network-based approaches for solving partial differential equations and provide a compelling alternative to traditional PDE solvers. Unfortunately, the PINN objective yields a difficult optimization problem to train the network. The difficulty stems from the presence of an ill-conditioned differential operator in the loss, which leads to an ill-conditioned objective. The present paper proposes a framework for solving parametric PDEs based on learning an optimizer tailored to that class of problems. The authors provide a heuristic argument showing that the learned optimizer can be viewed as preconditioning the original loss. Preliminary experiments show the proposed approach yields better performance relative to existing methods.

**Strengths:**

- This is the first time I've seen the use of learned optimizers in SciML. I think this approach has the potential to be very useful.
- The potential connection between learning the optimizer is interesting, though still somewhat tenuous; see weaknesses below.
- The proposed framework outperforms vanilla baselines, which is good.

**Weaknesses:**

**Related work**

 While I have not seen the idea of learning an optimizer applied in the SciML literature, it certainly is not new in the overall ML literature. The authors should explain how their approach fits into this existing body of work in the related work section.
The authors should mention Cho et al.'s recent work, which proposed a very effective method for solving parametric PDEs using PINNs.

**Theory**

The theoretical support for the benefit of learning the optimizer is weak.
The argument provided in Appendix B is heuristic and relies heavily on the assumption that both networks are well-approximated by their first-order Taylor expansions.
This regime only holds under certain strong assumptions, which the paper neglects to mention.
Finally, it is not apparent why it should be the case that $\kappa(PA) \ll \kappa(A)$.
So, the potential connection between the learning optimizer is interesting but very tenuous.

**Experiments**

The experiments in the paper are very weak.
Comparing to the vanilla PINN alone using L-BFGS for training is inadequate.
As is well known, better performance is achieved by using the combination of Adam+L-BFGS.
Plus, more sophisticated optimizers, like Muller et al.'s natural gradient approach or Rathore et al.'s NNCG, can greatly enhance PINN training.
It would be much more compelling to compare to one of these more sophisticated approaches to training PINNs.

The paper mentions the ability to cover parametric PDEs as a contribution, but none of the experiments demonstrate this advantage. Moreover, the method introduced in this paper should be compared to a dedicated PINN method for parametric PDES like that of Cho et al. (2024).

The authors' approach clearly has a higher time cost than the vanilla approach, but they do not provide runtimes for their method relative to the vanilla method. This makes it challenging to develop a concrete idea of how much more expensive their approach is.

**Presentation**

The text has many grammatical errors, which can sometimes make the presentation difficult to follow. I recommend the authors do a careful read-through and make appropriate edits.
This will greatly improve the readability of the paper.

**Overall**

I believe the paper contains some interesting ideas that have the potential to enhance the applicability of PINNs.
Unfortunately, the paper is not ready for publication at ICLR.
The experiments are lacking, so it is difficult to determine if the approach here would improve over more sophisticated variants of PINNs,  which are used in practice.
In addition, the theoretical contribution is too heuristic, and the presentation needs polishing.

**References**

Cho, W., Jo, M., Lim, H., Lee, K., Lee, D., Hong, S. and Park, N., 2024. Parameterized physics-informed neural networks for parameterized PDEs. arXiv preprint arXiv:2408.09446.

Müller, J. and Zeinhofer, M., 2023, July. Achieving high accuracy with PINNs via energy natural gradient descent. In International Conference on Machine Learning (pp. 25471-25485). PMLR.

Rathore, P., Lei, W., Frangella, Z., Lu, L. and Udell, M., 2024. Challenges in training PINNs: A loss landscape perspective. arXiv preprint arXiv:2402.01868.

**Questions:**

1) Why did you not include wall-clock times?

2) Why did you not include a comparison(s) with a more sophisticated variant of PINN(s)?

---

> ### Author Response · Authors · 2024-11-25
>
> We appreciate the reviewer's feedback and recognition of our approach's innovation and we address the raised concerns below.
>
> ## W1 Related work
> - **comparison with ref. [4]**: Firstly, we thank the reviewer for the recent reference [4] which has been added in the revised version. We have also performed new experiments with the model in [4] which appears as a new baseline in the updated table 2 (see P2INNs). As can be seen the performance of the model are close to the ones obtained with the other baselines and well below the ones obtained with our model. While running the experiments, we observed that this model also suffers from the same slow optimization schedule as the other baselines.
> Note that our evaluation setting is different from the one used in their paper [4]. Here we evaluate all the models for in-domain generalization, i.e. the models are trained on a set of PDE instances and tested on new instances of the PDE, with different parameters. In [4] the model is evaluated on new collocation point of the training PDEs.
>
> - As for the **positioning w.r.t. "learning to optimize" literature**, you are right, this has been explored in other fields of ML. **The relation with this literature is discussed in the last paragraph of our related work** section in appendix A under the name of "learning to solve". We briefly introduce the literature and we refer to a thorough survey by [5]. As for its use for physics-aware ML, we found only one very recent reference that leverages meta-learning [7].
>
>
> ## W2 Theory
> We agree that our theoretical analysis relies on a simplifying linearization assumption. A proof of convergence for the actual non linear NN model would require an analysis of the non linear regime which up to our knowledge is not available. This is why we developed the analysis of the simplified linear case. Although this does not solve the non linear problem, we believe that this provides an interesting intuition and motivation for the proposed method.
>
> The proof in the initial version of the paper was only a sketch and probably not clear enough. In the updated version,  we have thoroughly revised the proof to better highlight the main results, their logical progression, and the connections between them (appendix B).
> In particular, we have shown that the neural net optimizer leads to a condition number $\kappa(PA)$ equal to 1, which is optimal. If we consider the Poisson example, the condition number $\kappa(A)$ of the original PINNs problem being $K^4$, for $K>1$, the new condition number is greatly reduced. Note that this also directly implies that the number of steps $N^{\prime}(\epsilon)$ of our method is **significantly** reduced wrt to PINNs, i.e. $N^{\prime}(\epsilon)\ll N(\epsilon)$. This is all carefully explained in the new version of the paper, appendix B.

---

> > ### Author Response · Authors · 2024-11-25
> >
> > ## W3 Experiments
> > We thank the reviewer for the suggestions and we have added several new experiments for addressing this concern including a comparison with the reference work [4].
> >
> > - **Comparison with Adam+L-BFGS training**
> > We have added a comparison using an Adam+L-BFGS optimizer strategy [6] (see updated table 2 and table 1 below PINNs-multi-opt row). These results show that training PINNs using a Adam+L-BFGS optimizer indeed improves the training of PINNs. However, this still requires an extensive computational time to reach good performance, since one training has to be performed for each PDE (please see table 17).
> > We detail the experimental protocol, hyper parameters and computational times in the updated version of the manuscript, Appendix D3, tables 4 and 6).
> >
> > - **Parametric PDEs**
> > Our experiments directly target parametric PDEs. The models are trained on a set of PDE instances sampled from their parameter distribution (the parameters include PDE coefficients, boundary conditions and forcing terms). They are evaluated on new unseen PDE instances with **new parameter values** as indicated in table 1.
> > The model is compared with Operator baselines that also handle parametric PDEs [1, 2, 3].  Following your suggestion, we have also added a comparison with the dedicated method [4] as discussed above and included this new baseline in table 2 and in the table 1 below (see P2INNs row). We thank the reviewer for this suggestion, which will strengthen our comparison. We observe that this baseline behaves better than standard PINNs baselines. However, it still suffers from the PDE loss ill-conditioning, making it hard and slow to optimize and leading to performance well below the ones of our proposed approach.
> >
> > - **Runtime vs baselines** In Appendix E3 (Tables 16 and 17), we provide detailed information about the training and inference times for the experiments presented. Additionally, this section includes a dedicated discussion comparing our method with vanilla PINNs in terms of computational performance. See below (table 2 and 3) an excerpt of table 16 and 17 on the Helmholtz PDE.
> > The tables demonstrate that while our method incurs a slightly higher training time, it significantly outperforms vanilla PINNs during inference in term of compute effort. For example, in the case of the Helmholtz equations, vanilla PINNs using Adam+L-BFGS require 15 seconds to optimize a new PDE, whereas our method achieves this in just 0.2 seconds. This efficiency during inference is highlighted in the updated Table 17.
> >
> > We thanks the reviewer for his comments and suggestions on the experimental part of our paper. We hope that the added elements will convinced you about our submission and its capabilities.
> >
> > ## W4 Presentation
> > We will thoroughly correct the typos.
> >
> > ## Q1
> > We have added computational times in the appendix (E3). We  reference this section in the main part of the updated version of the manuscript, section 4.4.
> >
> > ## Q2
> > We have considered several operator methods as baselines to cover the parametric aspect. Following your suggestion, we have added [4] as additional baseline to strengthen our argumentation (see table 1 below (an excerpt of table 2), the updated version of the manuscript, table 2 and experimental detail in appendix D3).
> > Overall the training and inference times of our model are of the same order as those of the baselines.

---

> > > ### Author Response · Authors · 2024-11-25
> > >
> > > Table 1: Results of trained models - metrics in Relative MSE on the test set. Best performances are highlighted in bold, and second best are underlined.
> > >
> > > | Baseline             | 1d (Helmholtz) | 1d (Poisson) | 1d+time (NLRD) | 2d (Darcy-Flow) | 2d+time (Heat) |
> > > |-----------------------|----------------|--------------|----------------|-----------------|----------------|
> > > | PINNS-multi-opt       | 8.47e-1       | 1.18e-1      | 7.57e-1        | 8.38e-1         | 6.10e-1        |
> > > | P2INNs               | 9.90e-1       | 1.50e-1      | 5.69e-1        | 8.38e-1         | 1.78e-1        |
> > > | **Ours**             | **2.41e-2**   | **5.56e-5**  | _2.91e-4_      | **1.87e-2**     | **2.31e-3**    |
> > >
> > > Table 2: Training time of the experiments shown in Table 2 (main paper). of the paper on a NVIDIA TITAN RTX (25 Go) GPU. d stands for days, h for hours, m for minutes.
> > > | Method            | Helmholtz |
> > > |-------------------|-----------|
> > > | MLP + basis       | 30m       |
> > > | PPINNs            | 15m       |
> > > | P2INNs            | 2h        |
> > > | PODON             | 10m       |
> > > | PIDON             | 10m       |
> > > | PINO              | 15m       |
> > > | Ours              | 30m       |
> > >
> > >
> > > Table 3: Inference time averaged on the test set. All experiment are conducted on a single NVIDIA RTX A6000 (48Go). We report the mean time computed on the test set to evaluate the baselines as performed on Table 2 in the paper (i.e. with $10$ test-time optimization steps when applicable and $20$ steps on the Heat dataset). We consider as inference the solving of a PDE given its parameters and/or initial/boundary conditions.
> > > | Method            | Helmholtz |
> > > |-------------------|-----------|
> > > | MLP + basis       | 1.12e-2   |
> > > | PINNs+L-BFGS      | 274       |
> > > | PINNs-multi-opt   | 15.5      |
> > > | PPINNs            | 3.09e-1   |
> > > | P2INNs            | 2.84e-1   |
> > > | PODON             | 3.27e-1   |
> > > | PIDON             | 3.32e-1   |
> > > | PINO              | 3.14e-1   |
> > > | Ours              | 2.58e-1   |
> > >
> > >
> > > ## References
> > >
> > > [1] Sifan Wang, Hanwen Wang, and Paris Perdikaris. Learning the solution operator of parametric partial
> > > differential equations with physics-informed deeponets, 2021.
> > >
> > > [2] Somdatta Goswami, Aniruddha Bora, Yue Yu, and George Em Karniadakis. Physics-informed deep
> > > neural operator networks, 2022.
> > >
> > > [3] Zongyi Li, Hongkai Zheng, Nikola Kovachki, David Jin, Haoxuan Chen, Burigede Liu, Kamyar
> > > Azizzadenesheli, and Anima Anandkumar. Physics-informed neural operator for learning partial
> > > differential equations, 2023
> > >
> > > [4] Cho, W., Jo, M., Lim, H., Lee, K., Lee, D., Hong, S. and Park, N., 2024. Parameterized physics-informed neural networks for parameterized PDEs. arXiv preprint arXiv:2408.09446.
> > >
> > > [5] Tianlong Chen, Xiaohan Chen, Wuyang Chen, Howard Heaton, Jialin Liu, Zhangyang Wang, and
> > > Wotao Yin. Learning to optimize: A primer and a benchmark, 2021.
> > >
> > > [6] Rathore, P., Lei, W., Frangella, Z., Lu, L. and Udell, M., 2024. Challenges in training PINNs: A loss landscape perspective. arXiv preprint arXiv:2402.01868.
> > >
> > > [7] Alex Bihlo. Improving physics-informed neural networks with meta-learned optimization, 2024.

---

### Official Review · Reviewer_dXw8 · 2024-11-03

**Soundness:** 2
**Presentation:** 2
**Contribution:** 2
**Rating:** 3
**Confidence:** 4

**Summary:**

This paper proposes a method to overcome the challenging optimization of PDE-loss in physics-informed methods by learning a modification on the gradient (w.r.t. model parameters), which intuitively serves as a preconditioner. The experiment displays a noticeable acceleration in convergence.

**Strengths:**

-	The method is full of imaginations!

-	The empirical result in the test cases investigated are surprising. The proposed method help significantly accelerate the convergence of pde-loss optimization.

**Weaknesses:**

-	The theoretical analysis (main text and appendix B) is superficial and should not be considered as ‘proof’. In L814, the only important part of convergence analysis is circumvented by saying ‘the introduction of P as a preconditioner often results in κ(PA)<κ(A)’ instead of showing it. Thus the author actually didn’t prove anything. In fact, the effeteness of the method is highly related to whether it can help mitigate the condition number, which might not hold true, not supported by either theoretical or empirical evidence.
Also, L790-791 simplify the problem so much that the derivation could no longer provide insight to understanding the proposed method, unless the authors demonstrate the approximation to $Pv$ is reasonable in some sense.

-	To train the model $\mathcal{F}_{\rho}$, one has to first fix a set of grid point. This is neither flexible nor suitable to leverage the advantageous of PINN. In many PINN papers, the pde residual are estimated in a Monte-Carlo manner by drawing random points and computing its pointwise residual, so as to eliminate the heave computational and memory cost. Moreover, since the paper are target at solving a family of PDEs, it is always the case that one has to use finer grids / higher resolution to adequately capture small-scale effects for some harder instances in the large PDE class.

-	Regarding the method itself, in my opinion, the performance significantly depends on how well $\mathcal{F}_{\rho}$

generalizes to unseen or even out-of-distribution data. Intuitively it seems like it would be harder for $\mathcal{F}_{\rho}$

to generalize than the surrogate model (as in PINO or physics-DeepONet) itself, since the mapping $\mathcal{F}_{\rho}$ try

to approximate is more complicate. A systematic study on how well $\mathcal{F}_{\rho}$ needs to be learnt so as to ensure reasonable improvement on pde-loss convergence is necessary to make this method practical.

-	As mentioned in the limitations, the method will possibly suffer from a memory issue and is slow to train since it needs to back-propagate through several optimization iterations. This  hinders the practicability of the method.

-	The PDE considered are either 1D or linear, which are all quite toyish. The paper could benefit from studying their methods in some equations will optimization indeed brings severe challenges for physics-informed methods, for instance Navier-Stokes with relatively high Reynolds numbers. Otherwise, the experiments are not convincing.

-	Many details important for evaluating this paper are only mentioned in the appendix, e.g. the training and inference time cost. I am aware of the page limit, but the content should be briefly summarized in the main text, otherwise few readers notice they are there.

To conclude, although the paper is rich in novelty and has impressive performance in the experiments shown in the paper, and has the potential to provide new insight to the community, it is still not fully convincing- more understanding, discussion, and evaluation are necessary. Therefore, though I appreciate the author’s effort, I vote for rejection.

**Questions:**

-	Do you have any intuition on what the model $\mathcal{F}_{\rho}$ is learning? The paper interprets it as a preconditioner, but this answer is very vague and lack of information.

Think this way: given infinite data and a perfect training, what is the resulting mapping of $\mathcal{F}_{\rho}$?

Then consider: what is the effect of $\mathcal{F}_{\rho}$? Does it necessarily (or, ideally) help reduce the condition number?

-	Are the baseline hybrid methods trained with exactly the same dataset (and same amount of data) as the proposed method?

-	L308, isn’t $m$ the number of grid points instead of trajectories?

---

> ### Author Response · Authors · 2024-11-25
>
> We appreciate the reviewer's feedback and recognition of our approach's innovation and we address the raised concerns below.
>
> ## W1 Proof
>
>  - **Simplifying, linear assumption used in the theoretical analysis.** A proof of convergence for the actual non linear NN model would require an analysis of the non linear regime which up to our knowledge is not available. This is why we developed the analysis of the simplified linear case. Although this does not solve the non linear problem, we believe that this provides an interesting intuition and motivation for the proposed method.
>
> - Under these assumptions, the theoretical results demonstrate that $\kappa(PA)<\kappa(A)$.
> The proof in the initial version of the paper was only a sketch and probably not clear enough. In the updated version,  we have thoroughly revised the proof to better highlight the main results, their logical progression, and the connections between them (appendix B).
> In particular, we have shown that the neural net optimizer leads to a condition number $\kappa(PA)$ equal to 1, which is optimal. If we consider the Poisson example, the condition number $\kappa(A)$ of the original PINNs problem being $K^4$, for $K>1$, the new condition number is greatly reduced. Note that this also directly implies that the number of steps $N^{\prime}(\epsilon)$ of our method is **significantly** reduced wrt to PINNs, i.e. $N^{\prime}(\epsilon)\ll N(\epsilon)$. This is all carefully explained in the new version of the paper, appendix B.
>
>
> ## W2 Grids
> The proposed method could handle as well irregular grids and different resolutions since the reconstruction loss is computed point-wise through the basis.
> To illustrate this, we have added a new experiment for which the training points are sampled on a irregular grid (Appendix, section E1, table 13). This experiment highlights that our approach effectively handles non-linear grids while maintaining superior performance compared to most baselines. Below is an excerpt from table 13.
>
> Comparison of the performances when training our solver using regular or irregular and different grids for each PDE. Metrics on the test set.
> | Grid       | Relative MSE |
> |------------|--------------|
> | regular    | 2.41e-2      |
> | irregular  | 3.38e-1      |
>
>
> ## W3 Training and generalization of $\mathcal{F}_{\varrho}$
> - **Unseen PDEs**
> The paper targets solving parametric PDE, on a distribution of the PDE parameters including PDE coefficients, forcing terms and boundary conditions. The evaluations are performed on in-distribution, unseen equations with new parameter samples. Details for the sampling of the parameter values are provided in table 1.
> Our results show that the proposed method generalizes to these in-distribution unseen PDEs (Table 2, main text and Figure 14, appendix E1).
>
> - **Training of $\mathcal{F}_{\varrho}$**
> Following your suggestion, we have added training and testing loss plots as a function of the number of epochs for training on the Helmholtz equation with our method. These results are presented in section E5 Figures 15 and 16 of the updated manuscript and show that our method can exhibits interesting performances on the test set with only $200$ epochs of training (MSE $\approx 0.01$).
> Moreover, we show in appendix E1, figures 12a and 12b, a study of the performances of our method when reducing the number of available data. This experiment suggests that our model has better performance that the supervised method, even when fewer PDE solutions are available.
>
>
>  ## W4 Memory issue
>  We agree as indicated in the limitations that the proposed method requires a more complex training.
>  However, this approach provides a considerable acceleration at test time compared to classical PINN optimization schemes. This acceleration is achieved through two key factors: (i) improved convergence speed at test time, and (ii) the ability to handle parametric equations, whereas vanilla PINNs, for instance, require separate training for each set of parameter values. Note that all our experiments were conducted on a **single** GPU with a maximum capacity of 48 GB, including training, even for 2D+time equations.
>
>
>  ## W5 Datasets choices
>  We acknowledge the reviewer's point about the need for competitive benchmarking. We would like to stress that we have been using a variety of representative equations, including problems challenging for PINNs - see e.g. [1, 2] - (NLRD), FNO [3] (Darcy), PDEbench [4] (advection). Our benchmarks also include non-linear datasets (NLRD), 2D as well as 2D+t datasets (Darcy and Wave) and high frequencies (pulsation going until $50$ in the Helmholtz equation) which already represent interesting benchmarks.
>
> ## W6 Training and inference time cost
> Following your suggestion, we have updated the main text to emphasize the key insights regarding computational cost (Section 4.4, updated version of the paper).

---

> > ### Author Response · Authors · 2024-11-25
> >
> > ## Q1.
> >  We agree that the relation with the condition number is demonstrated (please see the updated version of the results/ proof in appendix B) for the linear approximation. In the proof, we make the assumption of perfect training in the sense that the basis can perfectly reconstruct the data. In the general case, we observe that the number of iterations is considerably reduced compared to baseline optimizers (see e.g. Fig 2) and that the optimizer learns also in this case more favorable gradient directions. In the new version, we have also added comparisons with more sophisticated gradient strategy combining  ADAM and LBFGS as suggested by reviewer N1S1 and we observe the same improvement compared to this strategy.
> >
> >  ## Q2.
> >  We confirm that the baselines are trained with the **exact same amount of data** (PDE as well as collocation points) as our method. For evaluation, we consider **new** PDEs by sampling PDE parameters and/or boundary/initial conditions **in** the training distribution. For Physics-based-only methods, we do not make use of the labeled solution data.
> >
> >  ## Q3.
> >  In the paper we denoted by $m$ the number of collocation points used for each PDE and by $M$ the number of PDEs used in the dataset.
> >
> >  ## References
> >
> > [1] Aditi S. Krishnapriyan, Amir Gholami, Shandian Zhe, Robert M. Kirby, and Michael W. Mahoney.
> > Characterizing possible failure modes in physics-informed neural networks. 2021.
> >
> > [2] Maryam Toloubidokhti, Yubo Ye, Ryan Missel, Xiajun Jiang, Nilesh Kumar, Ruby Shrestha, and
> > Linwei Wang. DATS: Difficulty-aware task sampler for meta-learning physics-informed neural
> > networks, 2024.
> >
> > [3] Zongyi Li, Nikola Kovachki, Kamyar Azizzadenesheli, Burigede Liu, Kaushik Bhattacharya, Andrew
> > Stuart, and Anima Anandkumar. Fourier neural operator for parametric partial differential equations,
> > 2020.
> >
> > [4] Makoto Takamoto, Timothy Praditia, Raphael Leiteritz, Dan MacKinlay, Francesco Alesiani, Dirk
> > Pflüger, and Mathias Niepert. Pdebench: An extensive benchmark for scientific machine learning,
> > 2023

---

> > > ### Comment · Reviewer_dXw8 · 2024-11-25
> > >
> > > Thanks for your reply!
> > >
> > > I didn't clearly see how your response is related to Q 1. I am not talking about the proof in that question. I am only asking two conceptual questions whose answers will help me better assess your draft.

---

> ### Author Response · Authors · 2024-12-01
>
> Thanks for the answer. Below, we provide additional insights to explain our "intuition" about the model's behavior.
>
> - **How F modifies the gradient path in the descent algorithm**
>
> Our core idea is related to the principle of "learning to solve." The optimizer F learns to adapt the gradient path dynamically for each instance of the equation. Specifically, F modifies the direction and magnitude of the stochastic gradient descent (SGD) updates in the physical loss landscape. It achieves this by utilizing solution values at various sample points to adjust the gradient, steering it toward the corresponding minimum in the mean squared error (MSE) landscape. While the descent remains within the residual physics error space, the learned optimizer provides an improved gradient direction, enhancing convergence efficiency.
>
> - **Approximation and optimal training (infinite data, optimal solution)**
>
> To further build intuition, we draw an analogy with solving linear systems. Assuming a linear approximation near the solution and ideal training conditions (i.e., infinite data and an optimal solution), the learned optimization effectively achieves ideal conditioning, where $\kappa=1$. While this linear analogy does not directly resolve nonlinear problems, it holds in the solution's vicinity and provides valuable insights into the model's local behavior.
>
> - **Illustration of convergence acceleration achieved by our method**
>
> Although we lack a general explanation for all cases, we illustrate the model's behavior using a comparison between baseline SGD and the learned optimizer in solving a Helmholtz equation. As shown in Figure 17 in the appendix, this new experiment highlights the slow convergence of PINNs compared to the significant acceleration achieved by our method. Notably, increasing the learning rate in PINNs led to divergence, further demonstrating the robustness and advantages of our approach.

---

> ### Author Response · Authors · 2024-12-02
>
> As the rebuttal deadline approaches, please let us know if we have adequately addressed your concerns. If you have any remaining questions or comments about our submission, we would be happy to address them.

---

### Author Response · Authors · 2024-11-25

We thank all the reviewers for their insightful comments and their help to improve the paper.
We have tried to address the suggestions and comments as best as possible. Moreover, we have added  new experiments and modifications to the paper to illustrate our answers.

- **Theoretical analysis (Reviewers dXw8, N1S1, 2vqB)**: We have thoroughly re-formulated the analysis to make the claims, proofs and conclusions clearer (Appendix B). We have also detailed the intermediate steps, while trying to keep the whole proof as clear as possible.  In particular, we formally show that under our linearization assumptions our approach is guaranteed to converge with optimality properties and provide bounds on the convergence rate.

- **Additional Baselines (Reviewers N1S1)**: Following your suggestions, we have added the method from [1] as a baseline for a parametric PINNs model and the sophisticated training scheme for PINNs introduced in [2].

- **Computational cost (Reviewers dXw8, N1S1 and 2vqB)**: We have added a summary of key findings on computational time in the main paper.

- **Additional ablations and experiments (Reviewers dXw8, 2vqB and N1S1)** are provided in appendix E: test on irregular grids (Reviewer dXw8), evaluation of different network architectures (FNO, ResNet, MLP), robustness of the models with respect to the PDE parameter range (Reviewer 2vqB)). We have updated all the tables (computations cost, performance) with the new baselines. We have also added train and test loss and reconstruction quality w.r.t. the training epoch (Reviewers N1S1 and 2vqB)).

- **Effect of our model (Reviewer dXw8)**: We proposed additional visualization tools to illustrate the effect of our method.

[1] Cho, W., Jo, M., Lim, H., Lee, K., Lee, D., Hong, S. and Park, N., 2024. Parameterized physics-informed neural networks for parameterized PDEs. arXiv preprint arXiv:2408.09446.

[2] Rathore, P., Lei, W., Frangella, Z., Lu, L. and Udell, M., 2024. Challenges in training PINNs: A loss landscape perspective. arXiv preprint arXiv:2402.01868.

---

### Meta-Review · Area_Chair_7vQj · 2024-12-20

**Metareview:**

The work proposes to use a neural network to learn a pre-conditioner for the PINN loss depending on some parameter of the PDE problem. Numerics on various problems show that converges speed is significantly increased and a proof of this in the linear setting is provided.

**Additional Comments On Reviewer Discussion:**

The reviewer raised some concerns about the theory as well as the presentation of the work. I believe the authors have addressed both issues well while the reviewers have not raised their scores. In particular, the reviewers TPuR and dXw8 have failed to respond to the authors' rebuttal and have not adjusted their scores, while the rebuttal has clearly addressed all major concerns. The proposed method makes a step towards addressing the very important problem of incorporating physical equations into supervised learning algorithms. The presented theory and numerics are significant and novel.

---

### Decision · Program_Chairs · 2025-01-22

Accept (Poster)